# DISTRIBUTIONAL VALUE GRADIENTS FOR STOCHASTIC ENVIRONMENTS

**Baptiste Debes**
PSI
KU Leuven
Leuven, Belgium
baptiste.debes@kuleuven.be

**Tinne Tuytelaars**
PSI
KU Leuven
Leuven, Belgium
tinne.tuytelaars@kuleuven.be

## ABSTRACT

Gradient-regularized value learning methods improve sample efficiency by leveraging learned models of transition dynamics and rewards to estimate return gradients. However, existing approaches, such as MAGE, struggle in stochastic or noisy environments, limiting their applicability. In this work, we address these limitations by extending distributional reinforcement learning on continuous state-action spaces to model not only the distribution over scalar state-action value functions but also over their gradients. We refer to this approach as Distributional Sobolev Training. Inspired by Stochastic Value Gradients (SVG), our method utilizes a one-step world model of reward and transition distributions implemented via a conditional Variational Autoencoder (cVAE). The proposed framework is sample-based and employs Max-sliced Maximum Mean Discrepancy (MSMMD) to instantiate the distributional Bellman operator. We prove that the Sobolev-augmented Bellman operator is a contraction with a unique fixed point, and highlight a fundamental smoothness trade-off underlying contraction in gradient-aware RL. To validate our method, we first showcase its effectiveness on a simple stochastic reinforcement-learning toy problem, then benchmark its performance on several MuJoCo environments.[1]

## 1 INTRODUCTION

Reinforcement learning (RL) tackles sequential decision-making by training agents to maximize cumulative rewards. Off-policy actor-critic algorithms pair an actor, generating the control policy, with a critic, estimating expected returns (i.e., the Q-function). This mapping from state-action pairs to expected returns, known as credit assignment, is typically learned via temporal-difference (TD) methods (Sutton, 1988) and is critical for policy optimization in continuous-action settings. *This paper is motivated by two lines of work aimed at improving credit assignment:*

- **The use of action-gradients:** In value-based continuous control, the critic's value is not used directly to select actions but to provide action-gradients for policy optimization (Lillicrap et al., 2016; Fujimoto et al., 2018; Haarnoja et al., 2018; D'Oro & Jaskowski, 2020). Conventional TD learning implicitly learns these gradients via value prediction, but this relies on smoothness assumptions of the true value function that can degrade performance. To address this limitation, D'Oro & Jaskowski (2020); Garibbo et al. (2024) incorporate gradient information (Czarnecki et al., 2017) into critic training by learning a transition–reward world model (i.e. a differentiable proxy for the environment) and backpropagating through it (Heess et al., 2015).

- **Distributional RL:** Many environments exhibit irreducible uncertainty in transitions and rewards. Distributional RL (Morimura et al., 2010; Bellemare et al., 2017; 2023) captures this by modeling the return distribution rather than just its expectation. Categorical (Barth-Maron et al., 2018) and quantile-based (Dabney et al., 2018b;a) approaches have provided rich and stable learning signals, yielding performance gains in a variety of tasks (Barth-Maron et al., 2018; Dabney et al., 2018a; Hessel et al., 2017).

---

[1]The JAX Bradbury et al. (2018) implementation is available at https://github.com/BaptisteDebes/Distributional-value-gradients.

We argue that randomness also affects action gradients of returns, which can have a detrimental effect, especially in high-dimensional action spaces. As our experiments show (Section 6), existing methods that use gradient information deterministically (Czarnecki et al., 2017; D'Oro & Jaskowski, 2020) may struggle once the gradient to model becomes noisy or stochastic, losing some of the sample-efficiency benefits of gradient modeling.

**Paper contributions**  We extend distributional modeling to capture both returns and their gradients, coining the framework *Distributional Sobolev Reinforcement Learning*. At its core is a novel Sobolev Bellman operator that bootstraps both return and gradient distributions. By combining gradient-based training with uncertainty modeling, we aim to boost policy and value learning. This necessitates a generative model that supports differentiation of outputs and their input gradients, hence we introduce *Distributional Sobolev Training* and detail its implementation. Since most environments are non-differentiable, we employ a conditional VAE (cVAE) (Sohn et al., 2015) to model transitions and rewards. This enriches SVG (Heess et al., 2015) with a more expressive neural architecture. Finally, we extend previous works on value gradient and introduce the framework of *Sobolev Temporal Difference*. We provide the first contraction proofs in this scheme. In this context, we introduce the maximum-sliced MMD metric as a practical divergence that induces contraction and is tractable to approximate.

## 2 BACKGROUND

### 2.1 NOTATION AND RL OBJECTIVE

We consider a Markov Decision Process (MDP) with continuous state and action spaces, $\mathcal{S}$ and $\mathcal{A}$, transition kernel $P\colon \mathcal{S} \times \mathcal{A} \to \mathcal{P}(\mathcal{S})$, reward law $R\colon \mathcal{S} \times \mathcal{A} \to \mathcal{P}(\mathbb{R})$, and initial distribution $\mu \in \mathcal{P}(\mathcal{S})$. A deterministic policy $\pi_\theta\colon \mathcal{S} \to \mathcal{A}$ induces the $\gamma$-discounted occupancy $d_\mu^{\pi_\theta} = (1-\gamma)\sum_{t=0}^\infty \gamma^t \operatorname{Law}(s_t \mid \pi_\theta, \mu)$ (Silver et al., 2014; D'Oro & Jaskowski, 2020). The Q-function $Q^\pi(s,a)$ is the expected future return starting from state $s$ and action $a$, i.e., $Q^\pi(s,a) = \mathbb{E}\left[\sum_{t=0}^\infty \gamma^t r(s_t, a_t) \mid s_0 = s, a_0 = a\right]$. It yields the objective

$$J(\theta) = \mathbb{E}_{s\sim\mu}\left[Q^{\pi_\theta}(s, \pi_\theta(s))\right]. \tag{1}$$

Under mild conditions (Silver et al., 2014), the Deterministic Policy Gradient theorem gives

$$\nabla_\theta J(\theta) = \frac{1}{1-\gamma}\,\mathbb{E}_{s\sim d_\mu^{\pi_\theta}}\left[\nabla_\theta \pi_\theta(s)\,\nabla_a Q^{\pi_\theta}(s,a)\right]_{a=\pi_\theta(s)} \tag{2}$$

### 2.2 TEMPORAL-DIFFERENCE AS AN AFFINE OPERATOR

In practice, the true $Q^\pi$ is unknown and is approximated by a parameterized critic $Q_\phi$. More generally, any value-like mapping $V\colon \mathcal{S} \times \mathcal{A} \to \mathcal{Y}$ (where $\mathcal{Y} = \mathbb{R}$ or a space of probability distributions Bellemare et al. (2017)) admits a temporal-difference update written as a single affine operator:

$$\left(\mathcal{T}_\pi V\right)(s,a) = b(s,a) + \mathcal{L}[V](s,a). \tag{3}$$

Here $b(s,a)$ injects the immediate-reward term and $\mathcal{L}$ linearly transforms the successor estimate. The following recovers the Bellman expectation operator where $Q$ is the state-action value function

$$b(s,a) = \mathbb{E}\left[R(s,a)\right], \qquad \mathcal{L}^{\mathrm{Exp}}[Q](s,a) = \gamma\,\mathbb{E}\left[Q(s',\pi(s')) \mid s,a\right] \tag{4}$$

In distributional RL Bellemare et al. (2017), let $Z^\pi(s,a)$ be the random return with distribution $\eta^\pi(s,a)$. One gets

$$b(s,a) = \operatorname{Law}\left[R(s,a)\right], \qquad \mathcal{L}^{\mathrm{Dist}}[\eta](s,a) = \mathbb{E}_{s'\sim P(\cdot\mid s,a)}\left[(x \mapsto \gamma x)_\# \,\eta\left(s',\pi(s')\right)\right], \tag{5}$$

where $(x \mapsto \gamma x)_\#\eta$ is the *law of $\gamma X$* when $X \sim \eta$, which yields

$$\left(\mathcal{T}_\pi^{\mathrm{Dist}}\eta\right)(s,a) = \operatorname{Law}\left[R(s,a) + \gamma\,Z(s',\pi(s'))\right], \quad \text{where} \quad s' \sim P(\cdot \mid s,a). \tag{6}$$

Using off-policy samples $(s,a,r,s') \sim B$ from a replay buffer (Mnih et al., 2013) together with delayed target networks $\theta', \phi'$ (Lillicrap et al., 2016; Fujimoto et al., 2018; Haarnoja et al., 2018), we define the one-step targets

$$\delta_{\mathrm{tgt}}(s,a,s') = r + \gamma\,Q_{\phi'}\left(s',\pi_{\theta'}(s')\right), \qquad \eta_{\mathrm{tgt}}(s,a) = \operatorname{Law}\left[r + \gamma\,Z_{\phi'}(s',\pi_{\theta'}(s'))\right]. \tag{7}$$

The critic $V_\phi$ (scalar $Q_\phi$ or distribution $Z_\phi$) is then trained by minimizing

$$\mathcal{L}(\phi) = \mathbb{E}_{(s,a,r,s')\sim B}\big[d\big(V_\phi(s,a),\, T(s,a)\big)\big], \tag{8}$$

where $T = \delta_{\text{tgt}}$ in the expected-value case or $T = \eta_{\text{tgt}}$ in the distributional case, and $d$ is either a regression loss (e.g. squared error) or a distributional metric such as the Wasserstein distance (Bellemare et al., 2017; Sun et al., 2024).

## 3 A NEW BELLMAN OPERATOR

### 3.1 LEARNING A USEFUL CRITIC

Many value-based methods (Lillicrap et al., 2016; Fujimoto et al., 2018; Haarnoja et al., 2018) rely on a learned critic to provide the actor's training signal, implying that "an actor can only be as good as allowed by its critic" (D'Oro & Jaskowski, 2020). Unfortunately, typical critics — predicting only mean returns — cannot capture inherent return uncertainty. Distributional RL addresses this by modeling the return distribution. However, as noted in D'Oro & Jaskowski (2020), another fundamental issue is that minimizing TD-error does not guarantee the critic will be effective at steering policy optimization. We therefore propose a more principled approach to distributional temporal-difference learning, which explicitly incorporates the critic's action-gradient into its training objective. This aligns critic optimization with policy improvement rather than just fitting returns or their distribution.

**Proposition 1.** *Let $\pi$ be an $L_{\pi,\theta}$-Lipschitz continuous policy, and let $\mathrm{Law}[\nabla_a Z^\pi(s,a)\,|_{a=\pi(s)}]$ and $\mathrm{Law}[\nabla_a \hat{Z}(s,a)\,|_{a=\pi(s)}]$ denote the true and estimated distributions of the action-gradients at $a = \pi(s)$, respectively. Define the $p$–Wasserstein distance between two probability measures $\mu, \nu$ by*

$$W_p(\mu,\nu) = \left(\inf_{\zeta \in \Pi(\mu,\nu)} \mathbb{E}_{(X,Y)\sim\zeta}\big[\|X - Y\|^p\big]\right)^{1/p}.$$

*Then, specializing to $p = 1$, the error between the true policy gradient $\nabla_\theta J(\theta)$ and its estimate $\nabla_\theta \hat{J}(\theta)$ satisfies*

$$\big\|\nabla_\theta J(\theta) - \nabla_\theta \hat{J}(\theta)\big\| \leq$$
$$\frac{L_{\pi,\theta}}{1-\gamma}\, \mathbb{E}_{s\sim d_\mu^\pi}\Big[W_1\big(\mathrm{Law}[\nabla_a Z^\pi(s,a)\,|_{a=\pi(s)}],\, \mathrm{Law}[\nabla_a \hat{Z}(s,a)\,|_{a=\pi(s)}]\big)\Big].$$

The proof is in Appendix D. This result generalizes Proposition 3.1 from D'Oro & Jaskowski (2020) to a distributional setting. The Lipschitz continuity of $\pi$ typically holds when using neural-network function approximation. We discuss why this assumption is reasonable in practice in Appendix C.2.

Following D'Oro & Jaskowski (2020), we induce a critic optimization objective from Proposition 1, showing that we can approximate the true policy gradient by matching the action gradients in the *distributional sense*. Using bootstrapping to approximate the true distribution leads to the optimization problem

$$\hat{Z} \in \arg\min_{\hat{Z}\in\mathcal{Z}}\ \mathbb{E}_{s\sim d_\mu^{\pi_\theta}}\Big[W_1\big(\mathrm{Law}\big[\nabla_a\hat{Z}(s,a)\big|_{a=\pi_\theta(s)}\big],\, \mathrm{Law}\big[\nabla_a\hat{Z}_{\text{tgt}}(s,a)\big|_{a=\pi_\theta(s)}\big]\big)\Big]. \tag{9}$$

$$\hat{Z}_{\text{tgt}}(s,a) := r(s,a) + \gamma\, \hat{Z}\big(s', \pi_\theta(s')\big), \qquad (s',r) \sim p(\cdot \mid s,a). \tag{10}$$

We note that *Equations 9–10 assume a known and differentiable dynamics model $p$.* We maintain this assumption for the time being and will relax it in Section 5. *Mirroring previous work (D'Oro & Jaskowski, 2020; Garibbo et al., 2024), we introduce this assumption upfront and then lift the constraint.* In the next section, we formalize the notions necessary to instantiate a working implementation of this optimization problem.

### 3.2 DISTRIBUTIONAL SOBOLEV TRAINING

In this section, we introduce a novel Bellman operator for learning the joint distribution of the discounted cumulative reward and its action-gradient, and then express it in the affine-transform form presented earlier.

**Random action Sobolev return**     We extend the random return $Z(s, a)$ to a *joint random variable* that captures both the return *and* its action-gradient. Formally, the *random action Sobolev return* is

$$Z^{S_a}(s, a) = \Big[\sum_{t=0}^{\infty} \gamma^t\, r(s_t, a_t);\ \nabla_a \sum_{t=0}^{\infty} \gamma^t\, r(s_t, a_t)\Big], \quad s_0 = s,\ a_0 = a. \tag{11}$$

**Sobolev distributional temporal difference**     Next, we define the Sobolev distributional Bellman operator $T_\pi^{S_a}$ over these $(|\mathcal{A}| + 1)$-dimensional random variables. Let $\eta^{S_a}(s, a) = \mathrm{Law}[Z^{S_a}(s, a)]$. We borrow notation from Zhang et al. (2021); Rowland et al. (2019) and extend the classical distributional operator as follows. Under policy $\pi$, sample

$$s' \sim P(\cdot \mid s, a), \quad a' = \pi(s'), \quad r \sim R(\cdot \mid s, a), \quad X' \sim \eta^{S_a}(s', a').$$

Define the full affine transform

$$\mathbf{f}^{S_a}\big(x\,;\, r, s', \gamma\big) = \big[f^{\mathrm{return}}(x);\ f^{\mathrm{action}}(x)\big]. \tag{12}$$

For readability we hereafter write $\mathbf{f}^{S_a}(x)$, implicitly carrying the dependence on $(r, s', \gamma)$. We define $T_\pi^{S_a}$ as the operator that, at each $(s, a)$, pushes the next-step law $\eta^{S_a}(s', a')$ forward through this pointwise affine map:

$$\big(T_\pi^{S_a}\, \eta^{S_a}\big)(s, a) \;:=\; \mathrm{Law}\big[\mathbf{f}^{S_a}(X')\big]. \tag{13}$$

Its components are

$$f^{\mathrm{return}}(x) = r + \gamma\, x^{\mathrm{return}}, \tag{14}$$

$$f^{\mathrm{action}}(x) = \frac{\partial r}{\partial a}(s, a)\ +\ \gamma\, \Big(\frac{\partial f}{\partial a}(s, a)\Big)^T \big[\partial_s x^{\mathrm{return}} + (\partial_s \pi(s'))^T x^{\mathrm{action}}\big]. \tag{15}$$

This action-gradient component is novel: it arises by differentiating the Bellman target, capturing how the return's gradient transforms under $P(s', r \mid s, a)$. The derivation of Eqs. 14–15 appears in Appendix C. Notably, since $f$, $r$, and $\pi$ are differentiable, these updates are implemented automatically via backpropagation through the reparameterized simulator (Baydin et al., 2018; Paszke et al., 2019; Bradbury et al., 2018).

**Affine form of the Sobolev Bellman backup**     As shown in Appendix C, the Sobolev Bellman backup can be written in a single affine-operator form:

$$Z^{S_a}(s, a) \;=\; b(s, a)\ +\ \mathcal{L}^{\mathrm{Sob}}(s, a)\big[\, Z^{S_a}(s', a')\big], \tag{16}$$

where $b(s, a) = \big(r(s, a), \partial_a r(s, a)\big) \in \mathbb{R}^{1+|\mathcal{A}|}$ collects the immediate reward and its action-gradient, and $\mathcal{L}^{\mathrm{Sob}}(s, a)$ is a state-action-dependent *linear operator* (not merely a matrix) encapsulating the Jacobian blocks of the transition $f$ and policy $\pi$. Importantly, Equation 16 represents a clear departure from recent works that use critic gradient information (D'Oro & Jaskowski, 2020; Garibbo et al., 2024). Whereas these methods use the action-gradient only as an auxiliary regularization signal, our formulation incorporates it directly into the quantity we perform temporal difference over, jointly with the scalar return. This joint TD structure is what enables the contraction analysis presented in the next section.

**Complete Sobolev TD.**     It is worth noting that exactly the same chain-rule derivation used in Eq. 15 extends to also bootstrap the *state-gradient* (not just the action-gradient), yielding the *complete* Sobolev Bellman operator. We discuss this operator in more detail in Appendix C. In the main text, we focus on the *incomplete* version for clarity and computational tractability. Although using both action- and state-gradients would provide more information, handling them together is substantially more computationally expensive and is known to be non-trivial in practice (Garibbo et al., 2024). The complete operator is therefore provided as a generalization for readers interested in the full theoretical framework, while the incomplete version captures the core ideas and is sufficient for our algorithmic development.

## 4 THEORETICAL RESULTS

The most natural distance for distributional RL is the Wasserstein metric, and we therefore begin by establishing contraction under this choice. We state the contraction results here, and defer the explicit contraction factors and the smoothness assumptions they rely on to Appendix D. The appendix

also contains results of the same type for the complete operator. The main takeaway is that, under appropriate and practically reasonable assumptions, we can show contraction.

**Theorem 1** (Action-gradient Sobolev contraction). *Assume bounded Jacobians for the transition $f$ and policy $\pi$, and a Lipschitz coupling relating state- to return-gradients. Then*

$$\bar{W}_p\big(T_\pi^{S_a}\eta_1,\ T_\pi^{S_a}\eta_2\big)\ \leq\ \gamma\,\kappa\,\bar{W}_p\big(\eta_1,\eta_2\big),$$

*where $\bar{W}_p(\eta_1,\eta_2) = \sup_{(s,a)} W_p(\eta_1(s,a),\eta_2(s,a))$ and $\kappa$ depends on the smoothness assumptions. If $\gamma\,\kappa < 1$, $T_\pi^{S_a}$ is a strict contraction.*

**Towards a tractable metric** Both the classical and *Sobolev* distributional Bellman operators (Eqs. 6, 13) involve intractable push-forward integrals. While the supremum–$p$–Wasserstein distance $\bar{W}_p$ is a natural theoretical choice (Bellemare et al., 2017), exact multivariate optimal transport costs $O(m^3\log m)$ with $m$ samples and, more broadly, Wasserstein distances are difficult to estimate and to use directly for training (Appendix E). This motivates a shift to metrics that remain faithful to distributional structure yet are easier to compute in practice. One such candidate is the Maximum Mean Discrepancy (MMD) (Gretton et al., 2012), a kernel-based divergence that is tractable, sample-based, and already explored in distributional RL (Nguyen et al., 2020; Killingberg & Langseth, 2023; Wiltzer et al., 2024). For laws $P, Q \subset \mathbb{R}^{d'}$, the squared Maximum Mean Discrepancy between $P$ and $Q$ is

$$\mathrm{MMD}^2(P,Q) = \mathbb{E}_{x,x'\sim P}\big[k(x,x')\big] + \mathbb{E}_{y,y'\sim Q}\big[k(y,y')\big] - 2\,\mathbb{E}_{x\sim P,\,y\sim Q}\big[k(x,y)\big], \tag{17}$$

where $k$ denotes the kernel function. Further properties and empirical estimators are discussed in Appendix F.2.

**Max–Sliced MMD.** To obtain *provable contraction* for our Sobolev operator, we lift MMD via the max–sliced divergence framework (Deshpande et al., 2019; Nadjahi et al., 2020). For $\theta \in \mathbb{S}^{d'-1}$ and $P_\theta(x) = \langle\theta, x\rangle$, we define

$$\mathbf{MS}\mathrm{MMD}(\mu,\nu) = \sup_{\theta\in\mathbb{S}^{d'-1}} \mathrm{MMD}\big((P_\theta)_{\#}\mu,\ (P_\theta)_{\#}\nu\big).$$

*Approximation.* $\mathbf{MS}\mathrm{MMD}$ can be approximated by gradient-based optimization of the direction $\theta$ on the unit sphere; see Algorithm 1.

**Contraction under Max–Sliced MMD.**

**Theorem 2** (Action–gradient Sobolev contraction under $\mathbf{MS}\mathrm{MMD}$). *Assume the conditions of Theorem 1 and the mild additions in Theorem 6. Then*

$$\overline{\mathbf{MS}\mathrm{MMD}}\Big(T_\pi^{S_a}\eta_1,\ T_\pi^{S_a}\eta_2\Big)\ \leq\ \gamma\,\kappa\,\overline{\mathbf{MS}\mathrm{MMD}}(\eta_1,\eta_2),$$

*where $\overline{\mathbf{MS}\mathrm{MMD}}(\eta_1,\eta_2) = \sup_{(s,a)\in\mathcal{S}\times\mathcal{A}} \mathbf{MS}\mathrm{MMD}(\eta_1(s,a),\eta_2(s,a))$ and $\kappa$ depends on the Jacobian bounds; see Appendix G. If $\gamma\,\kappa < 1$, $T_\pi^{S_a}$ is a strict contraction with a unique fixed point.*

**Trade-off interpretation** The condition $\gamma\,\kappa < 1$ makes the trade-off explicit: either enforce smoothness (reduce $\kappa$ via bounded Jacobians and Lipschitz couplings) or shorten the effective horizon (reduce $\gamma$). A similar quantity $\kappa$ governs our contraction results for both Wasserstein (Theorem 1) and max–sliced MMD (Theorem 2), so the trade-off arises in each case. Importantly, $\kappa$ is dictated by the environment's dynamics and policy sensitivities; when the underlying physics yields large gradients, this effect cannot be eliminated and the only remedy is to lower $\gamma$. *This observation is a core contribution of our work and is enabled by the Sobolev Temporal Difference framework we introduce.*

## 5 APPROACH

To turn our theoretical Sobolev-distributional Bellman operator into a practical algorithm, we need (i) a critic that produces joint return–gradient samples, (ii) Bellman backups without intractable

integrals, (iii) a tractable metric for comparing predicted and target distributions, and (iv) to relax the assumption that our environment is differentiable. We address each of these below.

**Sobolev inductive bias.** We coin the term *Sobolev inductive bias* for the simple idea of modeling a gradient by a gradient: using the gradient of our approximator to stand in for the gradient of the true function. Concretely, let $F: \mathbb{R}^a \to \mathbb{R}^b$ be a differentiable target and $f_\varphi$ a neural network parameterized by $\varphi$. Sobolev training is one concrete instance of this principle (e.g. (Czarnecki et al., 2017; D'Oro & Jaskowski, 2020)), which implements $\mathcal{L}^S(\varphi; x) = \|F(x) - f_\varphi(x)\|^2 + \lambda_S \|\nabla_x F(x) - \nabla_x f_\varphi(x)\|^2$. Further mathematical motivation for this inductive bias can be found in Appendix B.1.

**Reparameterized Sobolev critic.** We model the joint return-and-gradient law $\eta_\pi^{S_a}(s, a)$ by a generator

$$Z_\phi^{S_a} : (s, a, \xi) \longmapsto \left( Z_\phi(s, a, \xi),\ \nabla_a Z_\phi(s, a, \xi) \right), \qquad \xi \sim \mathcal{N}(0, I).$$

such that $\mathrm{Law}\left[ Z_\phi^{S_a}(s, a) \right] = \eta_\phi^{S_a}(s, a) \approx \eta_\pi^{S_a}(s, a)$. This purely sample-based critic sidesteps intractable likelihoods and scales naturally to high-dimensional actions. It is similar in spirit to Singh et al. (2022); Freirich et al. (2019), and is structured as a generative model that deterministically maps noise to samples (Li et al., 2015; Goodfellow et al., 2020). We discuss this choice further and why most alternative parametrizations would not fit our scenario in Appendix B.2.

**Overestimation bias.** Value estimates routinely exhibit overestimation (Hasselt, 2010; van Hasselt et al., 2015), and even gradient-regularized critics inherit this issue (D'Oro & Jaskowski, 2020; Garibbo et al., 2024). It biases the policy toward overvalued actions, undermining its performance. TD3 addresses it by training two critics and setting the target in Equation 8 to the minimum of the two (Fujimoto et al., 2018). In our sample-based distributional setting, we follow TQC (Kuznetsov et al., 2020): train two distributional critics, draw $N$ samples from each, discard the top $p\%$ by magnitude, and concatenate the rest to form the target distribution.

**One-step world model.** Now we relax the assumption that the environment is differentiable. Unlike (D'Oro & Jaskowski, 2020; Garibbo et al., 2024), which fit only the conditional expectation $(\hat{s}', \hat{r}) = \mathbb{E}[s', r \mid s, a]$, we learn a stochastic, differentiable simulator $g$ whose push-forward law approximates the true transition–reward distribution $P(s', r \mid s, a)$. Concretely, we posit

$$(\hat{s}', \hat{r}) = g(s, a, \varepsilon), \quad \varepsilon \sim \rho_w(\varepsilon), \qquad \mathrm{Law}\left[ g(s, a) \right] \approx \mathrm{Law}\left[ s', r \mid s, a \right].$$

Using $g$ in place of the true environment, our Sobolev Bellman update from Equation 12 becomes

$$Z^{S_a}(s, a) = \mathbf{f}^{S_a}\left( Z^{S_a}(\hat{s}', \hat{a}'); \hat{r}, s', \gamma \right), \quad \hat{a}' = \pi(\hat{s}').$$

We implement $g$ via a conditional VAE (Sohn et al., 2015), which has proven effective for modeling dynamics in RL (Ha & Schmidhuber, 2018; Zhu et al., 2024), with learned prior $p_\upsilon(\varepsilon \mid s, a)$, encoder $q_\zeta(\varepsilon \mid s, a, s', r)$, and decoder $p_\psi(s', r \mid s, a, \varepsilon)$.

Because Sobolev TD requires drawing many model transitions per update and differentiating each sample with respect to $(s, a)$, the world model must support both *cheap sampling* and *cheap reparameterized gradients*. Several generative model classes satisfy these requirements, such as normalizing flows (Dinh et al., 2016) or GANs (Goodfellow et al., 2020). Diffusion models (Ho et al., 2020) have shown strong empirical performance in high-dimensional generative tasks, but they are not well suited to our setting: computing the Jacobian of a generated sample with respect to its conditioning variables requires backpropagating through the full denoising chain, which is prohibitively expensive for the repeated one-step model queries whose gradients Sobolev TD evaluates.

**Algorithm summary.** We replace the critic in standard off-policy actor–critic (Fujimoto et al., 2018; Barth-Maron et al., 2018) with our Sobolev distributional critic and train it and the cVAE world model $g(s, a, \varepsilon)$ jointly; this yields the **Distributional Sobolev Deterministic Policy Gradient (DSDPG) algorithm**. At each step we sample Sobolev-returns and $(\hat{s}', \hat{r})$, form bootstrapped Sobolev-Bellman targets, minimize the MSMMD loss on value–gradient pairs, and update the actor by ascending the gradient of the critic's estimated expected return. *Even though plain MMD is not shown to be contractive, we also instantiate our framework using it, thus proposing two metrics to train the critics.* Fig. 1 (left) illustrates the overall DSDPG workflow, while the right panel shows how Sobolev-return distributions are sampled and the MMD loss is estimated. The full pseudo-code can be found in Appendix I.

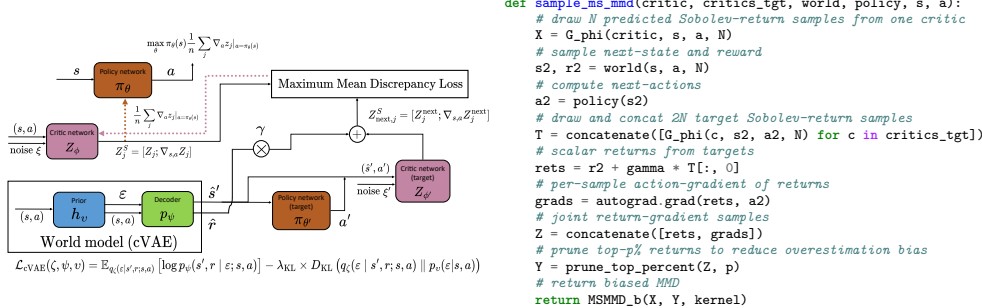

```python
def sample_ms_mmd(critic, critics_tgt, world, policy, s, a):
    # draw N predicted Sobolev-return samples from one critic
    X = G_phi(critic, s, a, N)
    # sample next-state and reward
    s2, r2 = world(s, a, N)
    # compute next-actions
    a2 = policy(s2)
    # draw and concat 2N target Sobolev-return samples
    T = concatenate([G_phi(c, s2, a2, N) for c in critics_tgt])
    # scalar returns from targets
    rets = r2 + gamma * T[:, 0]
    # per-sample action-gradient of returns
    grads = autograd.grad(rets, a2)
    # joint return-gradient samples
    Z = concatenate([rets, grads])
    # prune top-p% returns to reduce overestimation bias
    Y = prune_top_percent(Z, p)
    # return biased MMD
    return MSMMD_b(X, Y, kernel)
```

Figure 1: Left: block diagram of our DSDPG algorithm, where the critic $Z_\phi$ maps noise $\xi$ and $(s,a)$ to Sobolev-return samples, a cVAE world model generates next-state–reward samples $(\hat{s}', \hat{r})$, MSMMD or MMD compares predicted and bootstrapped Sobolev-return distributions, and the policy is updated via the critic's mean (gradient flows shown as dashed arrows; inspired by Singh et al. (2022)). Right: pseudocode for estimating the biased MSMMD between predicted and bootstrapped Sobolev returns.

## 6 RESULTS

### 6.1 TOY REINFORCEMENT LEARNING

We introduce a simple continuous-state, continuous-action 2D point-mass task within a square bounding box (Figure 2a). At each time step, the agent controls the 2D acceleration of the mass. At the start of each episode, we randomly draw one of $N$ possible bonus locations arranged uniformly around the center. Reaching this location ends the episode with a terminal reward. The agent does not know which location is correct. Instead, each time it visits a location, a binary memory flag is set. This partial observability makes the task sparse and forces exploration. By varying $N$, we tune the number of modes in the return distribution: small $N$ yields near-deterministic returns, large $N$ yields multimodal, high-variance returns. This toy task offers a clean, adjustable benchmark for gradient-aware distributional critics under controlled multimodal uncertainty.

As shown in Figure 2b, Distributional Sobolev (MSMMD Sobolev in pink and MMD Sobolev, orange) consistently outperforms all baselines as we increase the number of bonus locations. This indicates that our method effectively leverages gradient information and remains robust to the growing multimodality of the return distribution. By contrast, deterministic Sobolev (Huber Sobolev, red) offers no clear advantage over gradient-agnostic critics. Interestingly, the provably contractive MSMMD has a mild advantage over plain MMD. For brevity, we defer the full experimental details and baseline descriptions to the next subsection, while exact empirical settings are in Appendix K.

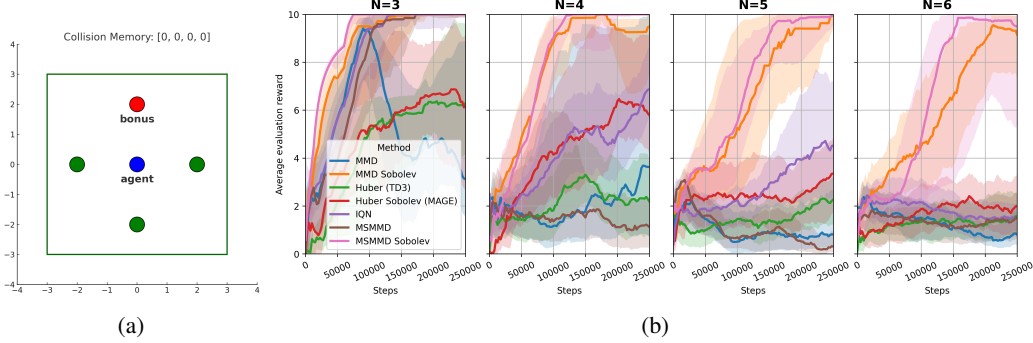

Figure 2: (a) Illustration of the 2D point-mass environment with $N$ possible bonus locations. (b) Evaluation curves (10 means the agent reached the bonus) for our Distributional Sobolev (MMD Sobolev and MSMMD Sobolev, in orange and pink resp.), deterministic Sobolev (MAGE D'Oro & Jaskowski (2020), in red), and other baselines as $N$ varies (median over 5 seeds and 25%-75% IQR).

## 6.2 REINFORCEMENT LEARNING

In this section, we evaluate the complete solution, including the learned world model, on several standard MuJoCo environments (Todorov et al., 2012) from the Gymnasium library Towers et al. (2024). Our deep learning framework is JAX (Bradbury et al., 2018). Similarly to MAGE (D'Oro & Jaskowski, 2020), for every exploration step, 10 critic updates are taken (UTD ratio = 10). All experiments are run in a *Dyna* setting (Deisenroth & Rasmussen, 2011) where new samples are drawn from the world model for every critic update. Thus, we evaluate the ability of the Distributional Sobolev training framework against deterministic Sobolev (Czarnecki et al., 2017; D'Oro & Jaskowski, 2020; Garibbo et al., 2024). *Hence, the results should not be understood as looking for state-of-the-art performance in the given environments but rather as a showcase that in some difficult settings, distributional methods can perform better.*

We compare against four baselines: (i) a TD3 variant trained with a Huber loss (Fujimoto et al., 2018; D'Oro & Jaskowski, 2020); (ii) deterministic Sobolev (MAGE) trained on both return and action-gradient via Huber loss (D'Oro & Jaskowski, 2020); and (iii-iv) distributional baselines, IQN (Dabney et al., 2018a) and standard MMD training (Nguyen et al., 2020). All variants involving MMD use a multiquadric kernel with $h = 100$. We use an identical base architecture for every method, with the distributional approaches sampling by reparameterization (Section 5). Each distributional variant generates 10 samples per transition. Further architectural and hyperparameter details are in Appendix L.

To evaluate robustness under stochastic dynamics, we use two complementary perturbations in MuJoCo. **(i) Multiplicative observation noise:** at the start of each episode, we sample $n \sim \mathcal{U}[0.8, 1.2]$ and scale the observation $s \mapsto n\,s$, following Khraishi & Okhrati (2023); this induces partial observability and substantially increases task difficulty. **(ii) Additive Gaussian dynamics noise:** we inject zero-mean Gaussian noise directly into positions and velocities, adopting the standard deviations reported by Khraishi & Okhrati (2023). This second perturbation makes control harder while preserving the qualitative structure of the original tasks. Results are summarized in Figure 3, where the left panels report the final evaluation performance after 150,000 iterations (250,000 for Humanoid-v2), and the right panels show the normalized area under the evaluation curve (nAUC). All results are reported as medians with bootstrapped 95% confidence intervals. The full learning curves are provided in Figure 10.

In the noise-free setting, DSDPG (MSMMD Sobolev and MMD Sobolev) match the performance of all baselines across six MuJoCo tasks. Under multiplicative observation noise, it outperforms every competitor in three of the six environments, most notably Ant-v2 and Humanoid-v2, while deterministic Sobolev (MAGE) suffers severe drops on Walker2d-v2 and Humanoid-v2 and shows larger variance in the simple InvertedDoublePendulum-v2. Under Gaussian noise, DSDPG again outperforms the baselines in three of the six environments. Notably, on Ant-v2, the noise scale borrowed from Khraishi & Okhrati (2023) makes the task substantially more difficult than previously reported.

Additional experiments are provided in Appendix M, including sensitivity to the noise scale, kernel bandwidth, number of Sobolev samples, and world-model capacity. We also report an ablation that removes the overestimation-bias correction. For DSDPG, this corresponds to disabling the TQC truncation (Kuznetsov et al., 2020), and for deterministic baselines, to removing the double estimation (Fujimoto et al., 2018). This ablation shows that overestimation bias correction plays a crucial role in stabilizing learning and achieving high performance. A wall-clock runtime comparison for Humanoid-v2 is provided in Table 3.

**World-model ablation.** To verify that our method does not depend on the particular choice of cVAE as world model, we performed an ablation in which the cVAE was replaced with a lightweight normalizing-flow model (Dinh et al., 2016). The flow-based model integrates into the Sobolev TD framework without modification, as the algorithm only requires that the world model provide cheap sampling and differentiable one-step transitions with respect to $(s, a)$. Across all tested MuJoCo tasks, the qualitative behaviour of the method remains unchanged, indicating that the benefits of Distributional Sobolev TD arise from its ability to exploit gradient information in stochastic environments rather than from the specific generative architecture used. A brief presentation of normalizing flows is provided in Appendix H.2, and the full ablation results, including architecture, training setup, and figures, are reported in Appendix M.1.

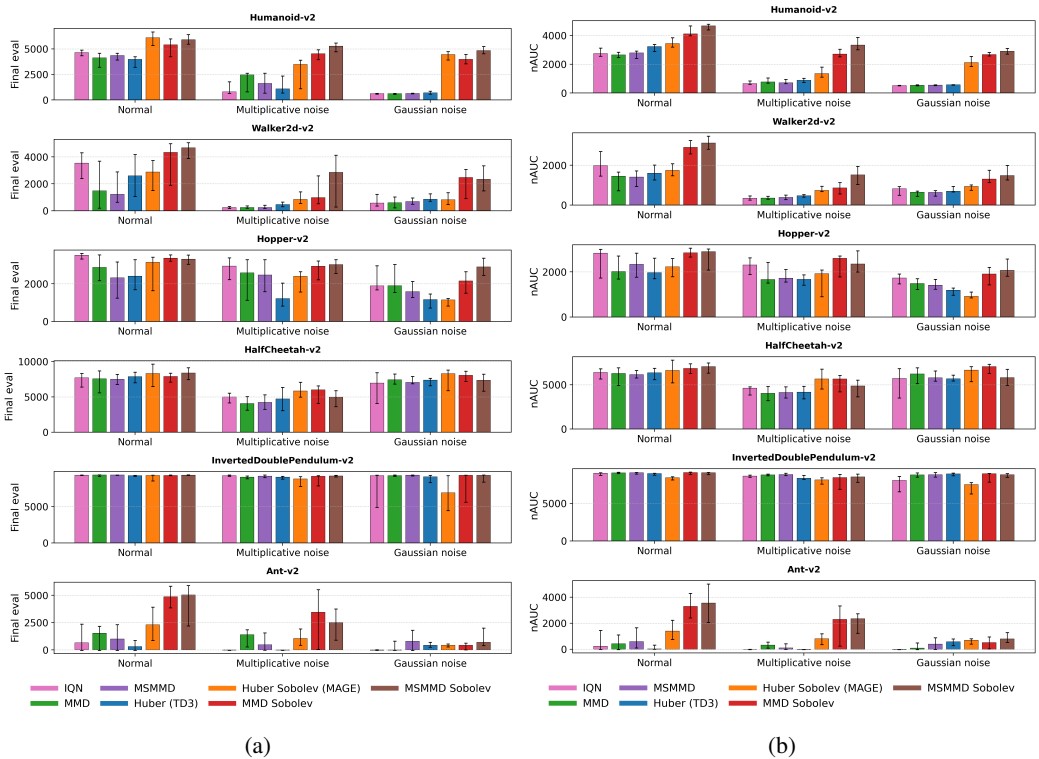

(a)                                                                           (b)

Figure 3: Evaluation of DSDPG (MSMMD/MMD Sobolev), deterministic Sobolev/MAGE D'Oro & Jaskowski (2020), TD3-Huber Fujimoto et al. (2018), IQN Dabney et al. (2018a), standard MMD Nguyen et al. (2020); Killingberg & Langseth (2023) on six MuJoCo tasks. Results are reported over 10 random seeds with median and 95% bootstrap confidence intervals. We compare three settings: normal environment, multiplicative observation noise, and Gaussian dynamics noise (Luo et al., 2021). (a) Final evaluation performance. (b) Normalized AUC over the entire training curve. Our DSDPG variants (MMD Sobolev and MSMMD Sobolev, red and brown) are on par or better than competing methods, shining especially on harder tasks and under noisy environments.

## 7 RELATED WORKS

**Gradient informed training** Our work explicitly models gradients in a stochastic manner using neural networks. As such, it can be seen as a distributional extension of Sobolev training Czarnecki et al. (2017) which was already adapted to reinforcement learning in D'Oro & Jaskowski (2020); Garibbo et al. (2024). Gradient modeling in value-based RL dates back to Fairbank (2008). Additionally, our approach shares connections with Physics Informed Neural Networks (PINNs) (Raissi et al., 2017), which approximate physical processes via differential constraints. In particular, uncertainty-aware PINNs (Yang et al., 2020; Zhong & Meidani, 2023) treat random processes and their derivatives as random variables, akin to our approach of leveraging Sobolev inductive bias and generative modeling (through reparameterization) while enforcing consistency via a tractable distributional distance.

**Distributional RL** We extend distributional RL (Bellemare et al., 2017; 2023) by modeling return gradients for deterministic policies in continuous action spaces, building on distributional DDPG (Barth-Maron et al., 2018; Lillicrap et al., 2016). Because these gradients are multi-dimensional, our method aligns with distributional multivariate returns (Zhang et al., 2021; Freirich et al., 2019; Wiltzer et al., 2024), highlighting the need for tractable discrepancy measures over multi-dimensional distributions. We treat our distributional critic as a generative model capable of producing actual samples of the underlying distribution (Freirich et al., 2019; Singh et al., 2022; Doan et al., 2018), in contrast to approaches that generate only pseudo-samples (Zhang et al., 2021; Nguyen et al., 2020) or rely on summary statistics (Bellemare et al., 2017; Barth-Maron et al., 2018; Dab-

ney et al., 2018b;a). We measure distributional discrepancy via the Maximum Mean Discrepancy (MMD) (Gretton et al., 2012; Li et al., 2015; Bińkowski et al., 2021; Oskarsson, 2020), which has proven effective in distributional RL (Nguyen et al., 2020; Killingberg & Langseth, 2023; Zhang et al., 2021).

**Model-based RL** Finally, because environment dynamics and rewards are unknown and non-differentiable, we adopt a world model akin to SVG(1) (Heess et al., 2015), instantiated as a cVAE (Kingma, 2013; Sohn et al., 2015). Generating new data through this model places our approach within model-based RL (Chua et al., 2018; Feinberg et al., 2018), particularly the *Dyna* family (Sutton, 1991). More specifically, our method is related to approaches that rely on variational techniques (Ha & Schmidhuber, 2018; Hafner et al., 2020; Zhu et al., 2024) or backpropagate through world models (Hafner et al., 2019; Clavera et al., 2020; Amos et al., 2021; Henaff et al., 2017; 2019; Byravan et al., 2020).

## 8 CONCLUSION, LIMITATIONS AND FUTURE WORK

In this work, we introduced Distributional Sobolev Deterministic Policy Gradient (DSDPG). Our main contributions involve modeling a distribution over both the output and the gradient of a random function, and deriving a tractable computational scheme to achieve this using Maximum Mean Discrepancy (MMD) and its max-sliced variant MSMMD. We demonstrated the effectiveness of the method in leveraging gradient information. We further extended the approach to reinforcement learning by leveraging a differentiable world model of the environment that infers gradients from observations. Building on this, we proposed the framework of Sobolev Temporal Difference, allowing us to provide the first contraction results in gradient-aware RL. We established contraction for both Wasserstein and the tractable MSMMD divergences, and highlighted smoothness assumptions required for contractive gradient-aware training. Finally, we showed that distributional gradient modeling improves stability in a controlled toy problem and enhances robustness to noise in MuJoCo environments, with stronger benefits in high-dimensional tasks.

While our approach shows promise, there remain challenges to address. A primary consideration is the high computational cost, as both policy evaluation and improvement require multiple samples from the distributional critic and their input gradients. Future work should explore more efficient inductive biases.

A second avenue for future work concerns the different variants of the Sobolev Temporal Difference operator. In the main text, we focus on the action-gradient variant for clarity and tractability, whereas the complete variant also includes state-gradients and provides additional information about how returns vary with the dynamics. Using both gradients jointly is computationally demanding and remains non-trivial in practice, but addressing these challenges could further extend the applicability of Sobolev-based distributional critics.

Finally, we believe that the ideas introduced in this work could benefit other fields where aleatoric uncertainty in gradient modeling is important, such as Physics-Informed Neural Networks (Yang & Perdikaris, 2019), Neural Volume Rendering (Lindell et al., 2021) and more generally applications of double backpropagation (Drucker & Le Cun, 1991).

## ACKNOWLEDGEMENTS

This project has received funding from the European Research Council (ERC) under the European Union's Horizon 2020 research and innovation programme (grant agreement n° 101021347).

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

APPENDIX CONTENTS

## A  DERIVATIVES OF REPARAMETERIZABLE RANDOM VARIABLES

In several places, such as Section 3, we need to talk about "the gradient of a random variable with respect to its conditioning variable." To make this precise and intuitive, we rely on the common *reparameterization trick*. We explain that, under appropriate continuity and measurability assumptions, this construction yields a well-defined random variable.

**Reparameterization view.** Suppose $Y \mid x$ is a conditional random variable, where $x \in \mathcal{X} \subset \mathbb{R}^n$ and $\mathcal{Y} \subset \mathbb{R}^m$. We assume there exists a measurable "noise" variable $\Xi \sim p(\xi)$ on a space $\mathcal{Z}$ and a deterministic map

$$g \; : \; \mathcal{X} \times \mathcal{Z} \; \longrightarrow \; \mathcal{Y}$$

that is $C^1$ in its first argument, such that

$$Y \; = \; g(x, \Xi) \quad \text{has the same law as } Y \mid x.$$

In other words, for each sample $\xi$, the realization

$$y(x; \xi) \; = \; g(x, \xi)$$

is a deterministic function of $x$.

**Pathwise derivative.** Because $g(\cdot, \xi)$ is differentiable in $x$, we define the *pathwise* or *reparameterization* derivative by

$$\frac{\partial}{\partial x} Y \; \stackrel{\text{def}}{=} \; \frac{\partial}{\partial x} g(x, \Xi),$$

that is, for each fixed $\xi$, $\; \partial_x y(x; \xi) \; = \; \partial_x g(x, \xi)$. We differentiate the pathwise realization $x \mapsto g(x, \xi)$, then regard $\partial_x g(x, \Xi)$ as a random variable taking values in $\mathbb{R}^{m \times n}$.

**Measurability.** Because $g \in C^1$ in its first argument, for each fixed $\xi$ the map

$$x \; \longmapsto \; \partial_x g(x, \xi)$$

is continuous in $x$. Moreover, by assumption $\xi \mapsto g(x, \xi)$ is measurable for each $x$. It follows from Lemma 4.51 from Aliprantis & Border (2006) ("Carathéodory functions are jointly measurable") that

$$(x, \xi) \; \longmapsto \; \partial_x g(x, \xi)$$

is jointly Borel–measurable on $\mathcal{X} \times \mathcal{Z}$. Consequently, when $\Xi \sim p(\xi)$ the composition $\partial_x g(x, \Xi)$ is a well-defined $\mathbb{R}^{m \times n}$–valued random variable.

**Practical computation.** In modern autodiff frameworks, once we formulate the generative model as $\; Y = g(x, \xi)$, calling $\mathrm{grad}(Y, x)$ automatically returns the sample-wise Jacobian $\partial_x g(x, \xi)$. Hence, all "random variable gradients" in our Sobolev Bellman operator are tractable to sample using standard reparameterization and automatic differentiation techniques.

## B  SOBOLEV INDUCTIVE BIAS AND DISTRIBUTION PARAMETRIZATION

### B.1  SOBOLEV INDUCTIVE BIAS

We discussed the Sobolev inductive bias in Section 5. Recall that given a differentiable target $F\colon \mathbb{R}^a \to \mathbb{R}^b$, Sobolev training (Czarnecki et al., 2017) (originally introduced as "double back-propagation" by Drucker & Le Cun (1991)) minimizes both value- and gradient-mismatch:

$$\mathcal{L}^S(\varphi; x) = \|F(x) - f_\varphi(x)\|^2 \; + \; \lambda_S\|\nabla_x F(x) - \nabla_x f_\varphi(x)\|^2.$$

The first term aligns network outputs to $F(x)$, while the second enforces that the input-derivatives of $f_\varphi$ match those of $F$. Below, we build on this bias to derive conditions on valid gradient fields and their relation to Jacobian symmetry.

Using the gradient of a neural network to predict gradients is not the only way to do so. Indeed, Jaderberg et al. (2017) proposed to generate *synthetic gradients* directly via the outputs of a neural network or a single linear layer. However, Czarnecki et al. (2017); Miyato et al. (2017) observed that predicting synthetic gradients as gradients of a neural network directly improved performance on decoupled trainings experiments from Jaderberg et al. (2017). A partial explanation for this observation is given by Proposition 4 in Czarnecki et al. (2017), which provides a necessary condition for a function $f_\varphi(x) : \mathbb{R}^n \to \mathbb{R}^n$ to produce a valid gradient vector field. It states that if $f_\varphi(x)$ produces a valid gradient vector field of some potential function $\phi(x)$, then the Jacobian matrix $\frac{\partial f_\varphi}{\partial x}(x) : \mathbb{R}^n \to \mathbb{R}^n \times \mathbb{R}^n$ must be symmetric.

This can be readily seen from the observation that if $\frac{\partial \phi}{\partial x}$ is the gradient vector field to be approximated by $f_\varphi(x)$ then $\frac{\partial^2 \phi}{\partial x^2}(x)$ is the Hessian matrix that is known to be symmetric.

A more in-depth discussion is provided in Chaudhari et al. (2024), where Lemma 1 gives a **necessary and sufficient** condition: a differentiable function $f : \mathbb{R}^n \to \mathbb{R}^n$ has a scalar-valued anti-derivative if and only if its Jacobian is symmetric everywhere, meaning that for all $x \in \mathbb{R}^n$, $\frac{\partial f}{\partial x}(x) = \left(\frac{\partial f}{\partial x}(x)\right)^T$. **This suggests a good inductive bias should be to restrict $f_\varphi(x)$ to the class of functions with symmetric Jacobian.** This condition is directly satisfied via the Sobolev inductive bias as well as gradient networks proposed in Chaudhari et al. (2024). As pointed out in Czarnecki et al. (2017), this is a very unlikely condition to be met by a regular feedforward neural network that would predict the gradient directly as its output.

### B.2  DISTRIBUTION PARAMETRIZATION

The Sobolev inductive bias introduced in Section 5 and B.1 requires a generative model where both the output and its input-gradient are treated as random variables. While the reparameterization trick (Kingma, 2013) could be used with a conditional Gaussian distribution, determining how to distribute the gradient of the samples with respect to the conditioners is less straightforward. To address this, we employ a method that relies solely on sampled data and does not assume tractable density estimation of Sobolev random variables.

Moreover, we found that the most common distribution parametrizations for distributional reinforcement learning, namely discrete categorical (Bellemare et al., 2017; Barth-Maron et al., 2018) and quantile-based (Dabney et al., 2018b;a), do not scale tractably to higher dimensions, specifically in the way they instantiate Equation 8. These considerations further motivated us to adopt a sample-based approach for the distributional Sobolev critic, similar in spirit to Singh et al. (2022); Freirich et al. (2019). As a result, our distributional critic is structured as a generative model that deterministically maps noise to samples (Li et al., 2015; Goodfellow et al., 2020). This parametrization lets us model a distribution over both outputs and their input-gradients via the Sobolev inductive bias: differentiating each generated sample with respect to the conditioner yields a reparameterized, sample-wise gradient, which aligns with the intuition from Appendix A.

## C  Sobolev Bellman operator

We first specify our smoothness and boundedness assumptions on the return, transition, reward, and policy functions in Section C.1. For completeness, we summarize mild conditions under which the neural policy and critic used in our method are Lipschitz in Section C.2. We then derive how differentiating the distributional Bellman equation yields the action-gradient update in Section C.3. Next, we show how to bundle the return and its gradient into a single affine update rule in Section C.4. We follow and give both the pathwise and integral forms of the full operator in Section C.5. Finally, we explain how to include state-gradients for a full Sobolev backup in Section C.6.

### C.1  Preamble and assumptions

Unlike traditional Bellman operators acting on scalar- or fixed-dimensional vector-valued functions, our operator acts on the space of continuously differentiable, bounded functions with bounded first derivatives over the compact domain $\mathcal{S} \times \mathcal{A}$. Concretely, we assume the return-distribution admits a reparameterization

$$Z\big(s, a; \varepsilon_Z\big) \ \in \ C_b^1\big(\mathcal{S} \times \mathcal{A}\big),$$

where $\varepsilon_Z \sim p(\varepsilon_Z)$ is exogenous noise, so that each sample $Z(s, a; \varepsilon_Z)$ and its gradients $\nabla_{s,a} Z(s, a; \varepsilon_Z)$ are bounded and continuous on $\mathcal{S} \times \mathcal{A}$.

Let $\varepsilon_f \sim p(\varepsilon_f)$, $\varepsilon_r \sim p(\varepsilon_r)$, and $\varepsilon_\pi \sim p(\varepsilon_\pi)$ be independent noise variables on $\mathcal{Z}$, and let

$$
\begin{aligned}
f &: \mathcal{S} \times \mathcal{A} \times \mathcal{Z} \to \mathcal{S}, && \text{(reparameterized transition)} \\
r &: \mathcal{S} \times \mathcal{A} \times \mathcal{Z} \to \mathbb{R}, && \text{(reparameterized reward)} \\
\pi &: \mathcal{S} \times \mathcal{Z} \to \mathcal{A}, && \text{(reparameterized policy)}
\end{aligned}
$$

be $C^1$ maps such that, for each draw $\varepsilon_f, \varepsilon_r, \varepsilon_\pi, \varepsilon_Z$,

$$s' = f(s, a; \varepsilon_f), \quad r = r(s, a; \varepsilon_r), \quad a' = \pi(s'; \varepsilon_\pi), \quad Z = Z(s, a; \varepsilon_Z).$$

**Importantly, we assume that the gradients of all these $C^1$ mappings ($Z$, $f$, $r$, and $\pi$) are tractable to compute in practice, for instance using automatic differentiation.**

We further assume these maps have uniformly bounded Jacobians: there exist constants $L_{f,s}, L_{f,a}, L_r, L_\pi, L_Z < \infty$ such that

$$\sup_{(s,a,\varepsilon_f)} \left\| \tfrac{\partial f}{\partial s}(s, a; \varepsilon_f) \right\| \le L_{f,s}, \qquad \sup_{(s,a,\varepsilon_f)} \left\| \tfrac{\partial f}{\partial a}(s, a; \varepsilon_f) \right\| \le L_{f,a},$$

$$\sup_{(s,a,\varepsilon_r)} \left\| \tfrac{\partial r}{\partial a}(s, a; \varepsilon_r) \right\| \le L_r, \qquad \sup_{(s,\varepsilon_\pi)} \left\| \tfrac{\partial \pi}{\partial s}(s; \varepsilon_\pi) \right\| \le L_\pi,$$

$$\sup_{(s,a,\varepsilon_Z)} \left\| \nabla_{s,a} Z(s, a; \varepsilon_Z) \right\| \le L_Z.$$

### C.2  Lipschitzness of neural policies and critics

For completeness, we summarize the mild conditions under which the neural policy and critic used in our method are Lipschitz. Similar assumptions are standard in analyses of gradient-aware RL methods (D'Oro & Jaskowski, 2020).

Let $F_\theta : \mathbb{R}^{d_{\text{in}}} \to \mathbb{R}^{d_{\text{out}}}$ denote either the deterministic policy $\pi_\theta(s)$ or the critic $Q_\theta(s, a)$ (deterministic or distributional). We assume a standard MLP

$$h_0(x) = x, \qquad h_{\ell+1}(x) = \sigma_\ell(W_\ell h_\ell(x) + b_\ell), \qquad \ell = 0, \ldots, L - 1,$$

with final output

$$F_\theta(x) = \begin{cases} \psi(W_L h_L(x) + b_L), & \text{(policy)}, \\ w^\top h_L(x) + c, & \text{(critic)}. \end{cases}$$

We require two standard conditions: (i) Each activation $\sigma_\ell$ and the policy's final activation $\psi$ is Lipschitz, with constants $L_{\sigma_\ell}$ and $L_\psi$. This includes common choices such as ReLU (Nair & Hinton,

2010), tanh, sigmoid, and SiLU (Elfwing et al., 2018). (ii) Each weight matrix has finite operator norm $\|W_\ell\|_2 \leq M_\ell < \infty$, which rules out parameter divergence. In practice, weight explosion would lead to immediately visible instabilities such as exploding Q-values, divergent TD-errors, and erratic actions, none of which appear in our experiments.

Under these conditions, each layer $x \mapsto \sigma_\ell(W_\ell x + b_\ell)$ is Lipschitz with constant $L_{\sigma_\ell} M_\ell$. Since compositions of Lipschitz functions produce Lipschitz maps whose constants multiply, it follows that the full network satisfies

$$\|F_\theta(x_1) - F_\theta(x_2)\| \leq K_F \|x_1 - x_2\|,$$

with

$$K_F = \begin{cases} L_\psi \prod_{\ell=0}^{L-1} (L_{\sigma_\ell} M_\ell), & \text{(policy)}, \\ \|w\|_2 \prod_{\ell=0}^{L-1} (L_{\sigma_\ell} M_\ell), & \text{(critic)}. \end{cases}$$

**Parameter-Lipschitzness.** If the input domain is bounded, $\|x\| \leq R$, and the network parameters remain bounded, then the hidden activations $h_\ell(x)$ are uniformly bounded on $\{x : \|x\| \leq R\}$. Consequently, the parameter Jacobian is uniformly bounded, so there exists $L_{F,\theta} < \infty$ such that

$$\sup_{\|x\| \leq R} \|\nabla_\theta F_\theta(x)\| \leq L_{F,\theta}.$$

In particular, for the policy we assume

$$\sup_s \|\nabla_\theta \pi_\theta(s)\| \leq L_{\pi,\theta}.$$

### C.3 DERIVATION OF THE ACTION-GRADIENT TERM

Starting from the distributional Bellman equation for each noise draw $(\varepsilon_f, \varepsilon_r, \varepsilon_Z, \varepsilon_\pi)$:

$$Z(s, a; \varepsilon_Z) = r(s, a; \varepsilon_r) + \gamma Z(s', a'; \varepsilon_Z'),$$

where $s' = f(s, a; \varepsilon_f)$, $a' = \pi(s'; \varepsilon_\pi)$, and $\varepsilon_Z' \sim p(\varepsilon_Z)$ independently. Differentiate w.r.t. $a$:

$$\frac{\partial}{\partial a} Z(s, a; \varepsilon_Z) = \frac{\partial}{\partial a} r(s, a; \varepsilon_r) + \gamma \frac{d}{da}\big[Z(s', a'; \varepsilon_Z')\big].$$

By the chain-rule,

$$\frac{d}{da} Z(s', a'; \varepsilon_Z') = \underbrace{\big(f_a(s, a; \varepsilon_f)\big)^T \frac{\partial}{\partial s} Z(s', a'; \varepsilon_Z')}_{\partial s'/\partial a} \ + $$
$$\underbrace{\big(f_a(s, a; \varepsilon_f)\big)^T \big(\pi_s(s'; \varepsilon_\pi)\big)^T \frac{\partial}{\partial a} Z(s', a'; \varepsilon_Z')}_{\partial a'/\partial a},$$

where we have introduced the shorthand

$$f_a(s, a; \varepsilon_f) = \frac{\partial f}{\partial a}(s, a; \varepsilon_f), \qquad \pi_s(s'; \varepsilon_\pi) = \frac{\partial \pi}{\partial s}(s'; \varepsilon_\pi).$$

Plugging back (and evaluating at $s' = f(s, a; \varepsilon_f)$, $a' = \pi(s'; \varepsilon_\pi)$) yields

$$\frac{\partial}{\partial a} Z(s, a; \varepsilon_Z) = \frac{\partial}{\partial a} r(s, a; \varepsilon_r) \ + $$
$$\gamma \big(f_a(s, a; \varepsilon_f)\big)^T \Big[\frac{\partial}{\partial s} Z(s', a'; \varepsilon_Z') + \big(\pi_s(s'; \varepsilon_\pi)\big)^T \frac{\partial}{\partial a} Z(s', a'; \varepsilon_Z')\Big],$$

which becomes the action-gradient component in our subsequent block-affine formulation.

## C.4  DISTRIBUTIONAL SOBOLEV BELLMAN UPDATE

In Section 3.2 we introduced a Bellman-style update that propagates both the random return and its action-gradient. We now rephrase it in a block-affine form.

Define the stacked random-vector, which by definition coincides with the joint Sobolev return introduced in the main text:

$$
Z^{S_a}(s,a;\varepsilon_Z) \;:=\; \begin{pmatrix} Z(s,a;\varepsilon_Z) \\ \dfrac{\partial}{\partial a} Z(s,a;\varepsilon_Z) \end{pmatrix} \;=\; H(s,a;\varepsilon_Z) \;\in\; \mathbb{R}^{1+m}.
$$

where $\varepsilon_Z \sim p(\varepsilon_Z)$, and let $\varepsilon_f, \varepsilon_r, \varepsilon_\pi$ be independent noise variables for transition, reward, and policy. For each draw $(\varepsilon_f, \varepsilon_r, \varepsilon_\pi, \varepsilon_Z)$ define

$$
s' = f(s,a;\varepsilon_f), \quad r = r(s,a;\varepsilon_r), \quad a' = \pi(s';\varepsilon_\pi).
$$

Then the sample-wise update is

$$
H(s,a;\varepsilon_Z) = \underbrace{\begin{pmatrix} r(s,a;\varepsilon_r) \\ \dfrac{\partial}{\partial a} r(s,a;\varepsilon_r) \end{pmatrix}}_{b(s,a;\varepsilon_r)} + \underbrace{\begin{pmatrix} \gamma & 0_{1\times m} \\ 0_{m\times 1} & \gamma \left(\frac{\partial f}{\partial a}(s,a;\varepsilon_f)\right)^T \left(\frac{\partial \pi}{\partial s}(s';\varepsilon_\pi)\right)^T \end{pmatrix}}_{A(s,a;\varepsilon_f,\varepsilon_\pi)} H\big(s',a';\varepsilon'_Z\big)
$$

$$
+ \underbrace{\begin{pmatrix} 0_{1\times n} \\ \gamma \left(\frac{\partial f}{\partial a}(s,a;\varepsilon_f)\right)^T \end{pmatrix}}_{N(s,a;\varepsilon_f)} \frac{\partial}{\partial s} Z\big(s',a';\varepsilon'_Z\big),
$$

(18)

where for the next-step noise we write $(\varepsilon_f, \varepsilon_\pi, \varepsilon'_Z) \sim p(\varepsilon_f)\, p(\varepsilon_\pi)\, p(\varepsilon_Z)$ independently. We abbreviate

$$
b(s,a;\varepsilon_r) = \begin{pmatrix} r(s,a;\varepsilon_r) \\ \partial_a r(s,a;\varepsilon_r) \end{pmatrix}
$$

as above.

Next, introduce the *state-gradient operator*

$$
\mathcal{D}_s : H \;\mapsto\; \frac{\partial}{\partial s}\big[e_1^T H\big] \;=\; \frac{\partial Z}{\partial s}.
$$

where $e_1^T = (1,0,\ldots,0)$ selects the return component. For each $(s,a;\varepsilon_f)$, define the combined linear map

$$
\boxed{\mathcal{L}\big(s,a;\varepsilon_f,\varepsilon_\pi\big) = A\big(s,a;\varepsilon_f,\varepsilon_\pi\big) + N\big(s,a;\varepsilon_f\big)\, \mathcal{D}_s\big|_{(s',a';\varepsilon'_Z)}}
$$

Then the distributional Sobolev Bellman update can be written as the single *(pseudo) affine* transform

$$
\boxed{H\big(s,a;\varepsilon_Z\big) = b\big(s,a;\varepsilon_r\big) \;+\; \mathcal{L}\big(s,a;\varepsilon_f,\varepsilon_\pi\big)\big[\,H\big(s',a';\varepsilon'_Z\big)\big].}
$$

(19)

**Remark (i).** In this form, the next-state gradient $\frac{\partial}{\partial s} Z(s',a';\varepsilon'_Z)$ is computed within the update and is accounted for by the linear operator $\mathcal{L}(s,a;\varepsilon_f,\varepsilon_\pi)$ via $\mathcal{D}_s$. Thus *computing the state-gradient of the distributional return* is an integral part of the Bellman operator itself.

**Remark (ii).** Although we write the update in a single "block-affine" line, it is not *self-contained* in the 2-vector $(Z,\partial_a Z)$. Indeed, the right-hand side still involves the next-step state-gradient term $N\,\partial_s Z$, which is not part of $H$. Thus we refer to this update as the *incomplete* Sobolev Bellman backup. We therefore call the update affine only in a loose, not literal, sense; the genuine finite-dimensional affine form appears once we lift to the full Sobolev vector $(Z,\partial_a Z,\partial_s Z)$ in Section C.6.

**Measurability**. By the same argument used in Appendix A, the map $(s,a,\varepsilon) \mapsto \partial_s Z(s,a;\varepsilon)$ is jointly Borel-measurable.

## C.5 DISTRIBUTIONAL SOBOLEV BELLMAN OPERATOR

Let $\eta : \mathcal{S} \times \mathcal{A} \longrightarrow \mathcal{P}(\mathbb{R}^{1+m})$ be the law of $H(s,a;\varepsilon_Z) = \big[ Z(s,a;\varepsilon_Z), \; \partial_a Z(s,a;\varepsilon_Z) \big]$, and assume three independent noise variables $\varepsilon_f \sim p_f, \varepsilon_r \sim p_r, \varepsilon_\pi \sim p_\pi$ generate

$$ s' = f(s,a;\varepsilon_f), \quad r = r(s,a;\varepsilon_r), \quad a' = \pi(s';\varepsilon_\pi). $$

We define the (incomplete) distributional Sobolev Bellman operator $T_\pi^{S_a}$ in two equivalent ways:

**Pathwise (law-equality) form:**

$$ \big(T_\pi^{S_a} \eta\big)(s,a) = \mathrm{Law}\Big[ b\big(s,a;\varepsilon_r\big) + \mathcal{L}\big(s,a;\varepsilon_f,\varepsilon_\pi\big)\big[X'\big]\Big], \tag{20} $$

where $X' \sim \eta(s',a')$ after sampling $\varepsilon_f, \varepsilon_r, \varepsilon_\pi$ and setting $s' = f(s,a;\varepsilon_f), a' = \pi(s';\varepsilon_\pi)$.

**Explicit-noise integral (pushforward) form:**

$$ \big(T_\pi^{S_a} \eta\big)(s,a) = \int_{\varepsilon_f} \int_{\varepsilon_\pi} \int_{\varepsilon_r} \big(b(s,a;\varepsilon_r) + \mathcal{L}(s,a;\varepsilon_f,\varepsilon_\pi)\big)_\# \eta\big(f(s,a;\varepsilon_f), \pi(f(s,a;\varepsilon_f);\varepsilon_\pi)\big) $$
$$ \times \, p_r(d\varepsilon_r) \times p_\pi(d\varepsilon_\pi) \times p_f(d\varepsilon_f). \tag{21} $$

## C.6 COMPLETE DISTRIBUTIONAL SOBOLEV BELLMAN OPERATOR

In Section 3.2 and above we developed the *action-gradient Sobolev Bellman backup*, in which our bootstrapped object is the pair

$$ Z^{S_a}(s,a) \; = \; \big(Z(s,a), \, \partial_a Z(s,a)\big) \; \in \; \mathbb{R}^{1+m}. $$

Because the Bellman operator needs the next-step state-gradient $\partial_s Z(s',a')$, which is not included in $Z^{S_a}$, this update is *incomplete*. We now introduce the *complete* operator, which augments the bootstrapped object with the state-gradient. Equivalently, we have to own the differentiable function $Z$ itself at evaluation time and thus rely on a Sobolev inductive bias as introduced in Appendix B.1.

**Lifting to a full Sobolev return.** Nothing prevents us from treating the state-gradient as a *third* component of our bootstrapped vector. Define

$$ Z^{S_{s,a}}(s,a) \; = \; \begin{pmatrix} Z(s,a) \\ \partial_a Z(s,a) \\ \partial_s Z(s,a) \end{pmatrix} \; \in \; \mathbb{R}^{1+m+n}. $$

Then, just as before, we obtain the affine update

$$ Z^{S_{s,a}}(s,a) = b_{\mathrm{full}}(s,a) \; + \; \mathcal{L}_{\mathrm{full}}(s,a)\,\big[\, Z^{S_{s,a}}(s',a')\big], \tag{22} $$

where $b_{\mathrm{full}}(s,a) \in \mathbb{R}^{1+m+n}$ collects $(r, \; \partial_a r, \; \partial_s r)$ and $\mathcal{L}_{\mathrm{full}}(s,a)$ is a bounded linear operator on $\mathbb{R}^{1+m+n}$. Here $\mathcal{L}_{\mathrm{full}}$ is simply a matrix, and we obtain $b_{\mathrm{full}}$ and $\mathcal{L}_{\mathrm{full}}$ by the same derivations as in the action-gradient case such that

$$ b_{\mathrm{full}}\big(s,a;\varepsilon_r\big) = \begin{pmatrix} r(s,a;\varepsilon_r) \\ \partial_a r(s,a;\varepsilon_r) \\ \partial_s r(s,a;\varepsilon_r) \end{pmatrix} $$

and

$$ \mathcal{L}_{\mathrm{full}}(s,a;\varepsilon_f,\varepsilon_\pi) = \gamma \begin{pmatrix} 1_{1\times 1} & 0_{1\times m} & 0_{1\times n} \\ 0_{m\times 1} & f_a^\top(s,a;\varepsilon_f)\,\pi_s^\top(s';\varepsilon_\pi) & 0_{m\times n} \\ 0_{n\times 1} & f_s^\top(s,a;\varepsilon_f)\,\pi_s^\top(s';\varepsilon_\pi) & f_s^\top(s,a;\varepsilon_f) \end{pmatrix}. $$

With $s' = f(s,a;\varepsilon_f)$ and $a' = \pi(s';\varepsilon_\pi)$ we denote the *complete* Sobolev Bellman operator by

$$ \big(T_\pi^{S_{s,a}}\eta\big)(s,a) = \mathrm{Law}\Big[b_{\mathrm{full}}\big(s,a;\varepsilon_r\big) + \mathcal{L}_{\mathrm{full}}\big(s,a;\varepsilon_f,\varepsilon_\pi\big)\big[X'\big]\Big], \qquad X' \sim \eta(s',a'). \tag{23} $$

**Consequences for implementation.** In this complete form, the second and third blocks of $Z^{S_{s,a}}$ need not be tied to the derivatives of the scalar head. One may simply design a network with three output heads

$$(s, a) \;\mapsto\; \big(Z_\theta(s, a), \; G_\theta^a(s, a), \; G_\theta^s(s, a)\big),$$

and train it against the Sobolev-affine targets in Equation 12. Of course, in practice one can still exploit the Sobolev inductive bias, thus ensuring $G^a = \partial_a Z$ and $G^s = \partial_s Z$.

## D    RESULTS WITH WASSERSTEIN

In this section, we complete the proof of Proposition 3.1 by deriving the upper bound on the policy gradient error in Section D.2. We then establish contraction properties of the Sobolev Bellman operators in the supremum–$p$–Wasserstein metric: Theorem 3 treats the *complete* operator $T_\pi^{S_{s,a}}$, while Theorem 4 gives the analogous statement for the *incomplete* operator that backs up only $(Z, \partial_a Z)$. Finally, Corollary 4.1 applies Banach's fixed-point theorem to conclude that the Sobolev iterates $\eta_{n+1} = T_\pi^S \eta_n$ converge geometrically to the unique fixed point $\eta^\pi$ identified in Lemma 8.

### D.1    PROOFS OF CONTRACTION AND FIXED POINT

**Definition 1** ($p$-Wasserstein distance Villani et al. (2009)). *Let $(X, d)$ be a metric space and let $\mathcal{P}_p(X)$ be the set of all probability measures on $X$ with finite $p$th moment. For any $\alpha, \beta \in \mathcal{P}_p(X)$, the $p$–Wasserstein distance between $\alpha$ and $\beta$ is*

$$W_p(\alpha, \beta) \;=\; \left( \inf_{\pi \in \Pi(\alpha, \beta)} \int_{X \times X} d(x, y)^p \, \mathrm{d}\pi(x, y) \right)^{1/p},$$

*where $\Pi(\alpha, \beta)$ is the set of all couplings of $\alpha$ and $\beta$, i.e. Borel probability measures on $X \times X$ whose marginals are $\alpha$ and $\beta$, respectively.*

**Definition 2** (Supremum-$p$–Wasserstein distance). *Let $(X, d)$ be a metric space, let $S$ be the set of states and $A$ the set of actions, and let $\mu_1, \mu_2 \colon S \times A \to \mathcal{P}_p(X)$ assign to each state–action pair $(s, a)$ a probability measure on $X$ with finite $p$th moment. The supremum-$p$–Wasserstein distance between $\mu_1$ and $\mu_2$ is*

$$\bar{W}_p(\mu_1, \mu_2) \;=\; \sup_{(s,a) \in S \times A} W_p\big(\mu_1(s, a), \, \mu_2(s, a)\big),$$

*where $W_p$ is the $p$–Wasserstein distance on $(X, d)$ as in Definition 1.*

**Lemma 1** (Push-forward law identity). *Let $Z$ be a random variable with distribution $\mu$, and let $f$ be any measurable function. Then*

$$\boxed{f_\# \mu \;=\; \mathrm{Law}\big(f(Z)\big).}$$

*Proof.* For any Borel set $A$,

$$\Pr\big(f(Z) \in A\big) = \Pr\big(Z \in f^{-1}(A)\big) = \mu\big(f^{-1}(A)\big) = f_\# \mu(A).$$

Since this holds for all $A$, we conclude $f_\# \mu = \mathrm{Law}(f(Z))$. $\qquad\qquad\qquad\square$

**Lemma 2** ((Pseudo-)affine pushforward form of the Sobolev Bellman operator). *Fix $(s, a)$. Draw independent exogenous noise*

$$\varepsilon_r \sim p(\varepsilon_r), \quad \varepsilon_f \sim p(\varepsilon_f), \quad \varepsilon_\pi \sim p(\varepsilon_\pi),$$

*and set*

$$r = r(s, a; \varepsilon_r), \quad s' = f(s, a; \varepsilon_f), \quad a' = \pi(s'; \varepsilon_\pi), \quad X' \sim \eta^{S_a}(s', a').$$

*Define the random (pseudo-)affine map*

$$\Phi_{s,a}\big(x; \varepsilon_r, \varepsilon_f, \varepsilon_\pi\big) \;=\; b\big(s, a; \varepsilon_r\big) \;+\; \mathcal{L}\big(s, a; \varepsilon_f, \varepsilon_\pi\big)[x], \quad \text{where } b = \begin{pmatrix} r \\ \partial_a r \end{pmatrix}, \quad \mathcal{L} = A + N \mathcal{D}_s$$

*as in Equation 19. Then*

$$\boxed{\big(T_\pi^{S_a} \eta^{S_a}\big)(s, a) = \mathrm{Law}\big(\Phi_{s,a}(X')\big).}$$

*In other words, at a fixed $(s, a)$, conditional on the exogenous noise, the Sobolev Bellman update is an affine pushforward of the joint return–gradient.*

*Proof.* Fix a Borel set $A \subset \mathbb{R}^{1+m}$. Writing out the randomness explicitly in terms of $\varepsilon_r, \varepsilon_f, \varepsilon_\pi$, we have

$$\Pr\big(\Phi_{s,a}(X') \in A\big) = \mathbb{E}_{\varepsilon_r, \varepsilon_f, \varepsilon_\pi}\Big[\Pr\big(\Phi_{s,a}(X') \in A \mid \varepsilon_r, \varepsilon_f, \varepsilon_\pi\big)\Big].$$

Conditioning on $(\varepsilon_r, \varepsilon_f, \varepsilon_\pi)$ turns $\Phi_{s,a}$ into the fixed (pseudo-)affine map $x \mapsto b + \mathcal{L}[x]$, so

$$\Pr\big(\Phi_{s,a}(X') \in A \mid \varepsilon_r, \varepsilon_f, \varepsilon_\pi\big) = \Pr\big(b + \mathcal{L}[X'] \in A\big).$$

Since $X' \sim \eta^{S_a}(s', a')$, an application of Lemma 1

$$\big[(b + \mathcal{L})_\# \, \eta^{S_a}(s', a')\big] = \mathrm{Law}\big(b + \mathcal{L}(X')\big)$$

gives

$$\Pr\big(b + \mathcal{L}[X'] \in A\big) = \big[(b + \mathcal{L})_\# \, \eta^{S_a}(s', a')\big](A).$$

Putting these together,

$$\Pr\big(\Phi_{s,a}(X') \in A\big) = \mathbb{E}_{\varepsilon_r, \varepsilon_f, \varepsilon_\pi}\Big[(b + \mathcal{L})_\# \, \eta^{S_a}(s', a')(A)\Big].$$

Finally, by the explicit-noise integral form Equation 21, this expectation is exactly the definition of $\big(T_\pi^{S_a} \eta^{S_a}\big)(s, a)(A)$. Hence

$$\Pr\big(\Phi_{s,a}(X') \in A\big) = \big(T_\pi^{S_a} \eta^{S_a}\big)(s, a)(A),$$

and since this holds for every Borel $A$, we conclude $\mathrm{Law}\big(\Phi_{s,a}(X')\big) = \big(T_\pi^{S_a} \eta^{S_a}\big)(s, a).$ □

**Lemma 3** (Affine push-forward contraction). *Let $(X, d)$ be a normed vector space equipped with the metric $d$ induced by its norm, and let*

$$F \colon X \to X, \qquad F(x) = b + L[x],$$

*where $b \in X$ is fixed and $L \colon X \to X$ is a bounded linear operator. Define its Lipschitz constant*

$$\|L\|_d = \sup_{x \neq y} \frac{d\big(L[x], L[y]\big)}{d(x, y)} < \infty.$$

*Then for any two probability measures $\alpha, \beta$ on $X$ with finite pth moment,*

$$\boxed{W_p\big(F_\# \alpha, \, F_\# \beta\big) \leq \|L\|_d \, W_p(\alpha, \beta)}$$

*In particular, when $L = \gamma I$ this recovers Lemma 3 of Zhang et al. (2021).*

*Proof.* Fix any $\varepsilon > 0$, and choose a coupling

$$\pi \in \Pi(\alpha, \beta)$$

that is $\varepsilon$-*optimal*, meaning that its transport cost is within $\varepsilon$ of the infimum:

$$\Big(\int_{X \times X} d(x, y)^p \, d\pi(x, y)\Big)^{1/p} < W_p(\alpha, \beta) + \varepsilon.$$

Push this coupling forward under $F \times F$ to obtain

$$\pi' = (F \times F)_\# \pi \in \Pi\big(F_\# \alpha, \, F_\# \beta\big).$$

Then

$$
\begin{aligned}
W_p\big(F_\# \alpha, \, F_\# \beta\big)^p &\leq \int_{X \times X} d(u, v)^p \, d\pi'(u, v) && \text{(by definition of } W_p) \\
&= \int_{X \times X} d\big(F(x), F(y)\big)^p \, d\pi(x, y) && (\pi' = (F \times F)_\# \pi) \\
&= \int_{X \times X} d\big(L[x], L[y]\big)^p \, d\pi(x, y) && (F(x) - F(y) = L[x - y]) \\
&\leq \|L\|_d^p \int_{X \times X} d(x, y)^p \, d\pi(x, y) && \big(d(L[x], L[y]) \leq \|L\|_d \, d(x, y)\big) \\
&< \|L\|_d^p \big(W_p(\alpha, \beta) + \varepsilon\big)^p && (\pi \text{ is } \varepsilon\text{-optimal}).
\end{aligned}
$$

Taking the $p$th root and letting $\varepsilon \to 0$ yields

$$W_p\big(F_\# \alpha, \, F_\# \beta\big) \leq \|L\|_d \, W_p(\alpha, \beta).$$

□

**Lemma 4** (Mixture non-expansion (conditional form, Zhang et al. (2021), Lemma 4)). *Let $C$ be a random variable on $(\Omega, \mathcal{F}, \rho)$, and let $Z_1, Z_2$ be $\mathbb{R}^d$-valued random variables. Let $p \geq 1$ and suppose there exists $\delta \geq 0$ such that for every $c \in \Omega$,*

$$W_p\big(\mathrm{Law}(Z_1 \mid C = c), \mathrm{Law}(Z_2 \mid C = c)\big) \leq \delta.$$

*Then the marginal distributions satisfy*

$$W_p\big(\mathrm{Law}(Z_1), \mathrm{Law}(Z_2)\big) \leq \delta.$$

*Equivalently,*

$$
\boxed{
\begin{array}{c}
\sup_{c \in \Omega} W_p\big(\mathrm{Law}(Z_1 \mid C = c), \mathrm{Law}(Z_2 \mid C = c)\big) \leq \delta \\[4pt]
\Longrightarrow \\[4pt]
W_p\big(\mathrm{Law}(Z_1), \mathrm{Law}(Z_2)\big) \leq \delta
\end{array}
}
$$

*In other words, averaging over the conditioning index cannot exceed the supremum of the conditional Wasserstein distances.*

*Proof.* This proof follows Zhang et al. (2021). Fix any $\varepsilon > 0$. By assumption, for every $c \in \Omega$,

$$W_p\big(\mathrm{Law}(Z_1 \mid C = c), \mathrm{Law}(Z_2 \mid C = c)\big) \leq \delta,$$

so there exists a coupling

$$\pi_c \in \Pi\big(\mathrm{Law}(Z_1 \mid C = c), \mathrm{Law}(Z_2 \mid C = c)\big)$$

such that

$$\int_{\mathbb{R}^d \times \mathbb{R}^d} d(x, y)^p \, d\pi_c(x, y) \leq (\delta + \varepsilon)^p.$$

By the law of total probability, the marginals are

$$\mathrm{Law}(Z_1)(\cdot) = \int_\Omega \mathrm{Law}(Z_1 \mid C = c)(\cdot) \, \rho(dc),$$

$$\mathrm{Law}(Z_2)(\cdot) = \int_\Omega \mathrm{Law}(Z_2 \mid C = c)(\cdot) \, \rho(dc).$$

Define a global coupling $\bar{\pi}$ on $\mathbb{R}^d \times \mathbb{R}^d$ by

$$\bar{\pi}(U) = \int_\Omega \pi_c(U) \, \rho(dc), \qquad U \subseteq \mathbb{R}^d \times \mathbb{R}^d.$$

Then for any measurable $A \subset \mathbb{R}^d$,

$$\bar{\pi}(A \times \mathbb{R}^d) = \int_\Omega \pi_c(A \times \mathbb{R}^d) \, \rho(dc) = \int_\Omega \mathrm{Law}(Z_1 \mid C = c)(A) \, \rho(dc) = \mathrm{Law}(Z_1)(A),$$

and similarly $\bar{\pi}(\mathbb{R}^d \times A) = \mathrm{Law}(Z_2)(A)$, so $\bar{\pi} \in \Pi(\mathrm{Law}(Z_1), \mathrm{Law}(Z_2))$, meaning $\bar{\pi}$ is a valid joint law for a pair whose marginals are $\mathrm{Law}(Z_1)$ and $\mathrm{Law}(Z_2)$. Hence $\bar{\pi}$ is an admissible coupling in the definition of $W_p(\mathrm{Law}(Z_1), \mathrm{Law}(Z_2))$.

$$
\begin{aligned}
W_p\big(\mathrm{Law}(Z_1), \mathrm{Law}(Z_2)\big)^p &\leq \int_{\mathbb{R}^d \times \mathbb{R}^d} d(x, y)^p \, d\bar{\pi}(x, y) && \text{(by definition of } W_p) \\
&= \int_\Omega \Big[ \int d(x, y)^p \, d\pi_c(x, y) \Big] \rho(dc) && \text{(by definition of } \bar{\pi}) \\
&\leq \int_\Omega (\delta + \varepsilon)^p \, \rho(dc) && (\pi_c \text{ is } \varepsilon\text{-optimal}) \\
&= (\delta + \varepsilon)^p.
\end{aligned}
$$

Therefore,

$$W_p\big(\mathrm{Law}(Z_1), \mathrm{Law}(Z_2)\big) \leq (\delta + \varepsilon),$$

and since $\varepsilon > 0$ was arbitrary, letting $\varepsilon \to 0$ yields

$$W_p\big(\mathrm{Law}(Z_1), \mathrm{Law}(Z_2)\big) \leq \delta.$$

$\square$

**Lemma 5** (Mixture $p$-convexity for $W_p$). *Let $(X, d)$ be a metric space, $p \in [1, \infty)$, and let $(\Omega, \mathcal{F}, \rho)$ be a probability space. Let $(\mu_c)_{c \in \Omega}, (\nu_c)_{c \in \Omega} \subset \mathcal{P}_p(X)$ be such that the mixtures $\int_\Omega \mu_c \, \rho(dc)$ and $\int_\Omega \nu_c \, \rho(dc)$ are well defined. Then*

$$W_p\Big( \int_\Omega \mu_c \, \rho(dc), \int_\Omega \nu_c \, \rho(dc) \Big) \ \leq \ \left( \int_\Omega W_p(\mu_c, \nu_c)^p \, \rho(dc) \right)^{1/p}.$$

*Proof.*

**Step 1: $\varepsilon$-optimal couplings for each $c$.**
Fix $\varepsilon > 0$. For each $c \in \Omega$, pick an $\varepsilon$-optimal coupling $\pi_c^\varepsilon \in \Pi(\mu_c, \nu_c)$ such that

$$\int_{X \times X} d(x, y)^p \, \pi_c^\varepsilon(dx, dy) \ \leq \ W_p(\mu_c, \nu_c)^p + \varepsilon.$$

**Step 2: Measurable selection and mixed coupling.**
Assume the couplings $\pi_c^\varepsilon$ can be chosen as a measurable function of $c$ so that the following mixed coupling is well defined

$$\bar{\pi}^\varepsilon(U) \ := \ \int_\Omega \pi_c^\varepsilon(U) \, \rho(dc), \qquad U \subseteq X \times X \text{ Borel}.$$

For any measurable $A \subseteq X$,

$$\bar{\pi}^\varepsilon(A \times X) = \int_\Omega \pi_c^\varepsilon(A \times X) \, \rho(dc) = \int_\Omega \mu_c(A) \, \rho(dc) = \Big( \int_\Omega \mu_c \, \rho(dc) \Big)(A),$$

and similarly

$$\bar{\pi}^\varepsilon(X \times A) = \int_\Omega \pi_c^\varepsilon(X \times A) \, \rho(dc) = \int_\Omega \nu_c(A) \, \rho(dc) = \Big( \int_\Omega \nu_c \, \rho(dc) \Big)(A).$$

Hence $\bar{\pi}^\varepsilon$ has the mixed marginals $\int_\Omega \mu_c \, \rho(dc)$ and $\int_\Omega \nu_c \, \rho(dc)$, meaning that

$$\bar{\pi}^\varepsilon \ \in \ \Pi\Big( \int_\Omega \mu_c \, \rho(dc), \int_\Omega \nu_c \, \rho(dc) \Big).$$

**Step 3: Bound the transport cost of the mixed coupling.**
Since $(c, x, y) \mapsto d(x, y)^p$ is nonnegative and measurable, Tonelli's theorem allows us to exchange the order of integration in $(c, x, y)$:

$$
\begin{aligned}
\int_{X \times X} d(x, y)^p \, \bar{\pi}^\varepsilon(dx, dy) &= \int_{X \times X} d(x, y)^p \left( \int_\Omega \pi_c^\varepsilon(dx, dy) \, \rho(dc) \right) \\
&= \int_\Omega \left( \int_{X \times X} d(x, y)^p \, \pi_c^\varepsilon(dx, dy) \right) \rho(dc) \\
&\leq \int_\Omega \left( W_p(\mu_c, \nu_c)^p + \varepsilon \right) \rho(dc) \\
&= \int_\Omega W_p(\mu_c, \nu_c)^p \, \rho(dc) \ + \ \varepsilon.
\end{aligned}
$$

**Step 4: Take the infimum over couplings and pass to the limit.**
By definition of $W_p$,

$$W_p\Big( \int \mu_c \, d\rho, \int \nu_c \, d\rho \Big)^p \ \leq \ \int_{X \times X} d(x, y)^p \, \bar{\pi}^\varepsilon(dx, dy) \ \leq \ \int_\Omega W_p(\mu_c, \nu_c)^p \, \rho(dc) + \varepsilon.$$

Taking $p$th roots and letting $\varepsilon \to 0$ yields

$$W_p\Big( \int_\Omega \mu_c \, \rho(dc), \int_\Omega \nu_c \, \rho(dc) \Big) \ \leq \ \left( \int_\Omega W_p(\mu_c, \nu_c)^p \, \rho(dc) \right)^{1/p}.$$

$\square$

**Lemma 6** (Spectral norm of a block-triangular matrix). *Let $A \in \mathbb{R}^{m \times m}$, $B \in \mathbb{R}^{n \times m}$, and $C \in \mathbb{R}^{n \times n}$. Then*

$$\left\| \begin{pmatrix} 1 & 0 & 0 \\ 0 & A & 0 \\ 0 & B & C \end{pmatrix} \right\|_2 \leq \max\{1, \|A\|_2, \|C\|_2\} + \|B\|_2.$$

*Proof.* Split

$$M = \begin{pmatrix} 1 & 0 & 0 \\ 0 & A & 0 \\ 0 & B & C \end{pmatrix} = \underbrace{\begin{pmatrix} 1 & 0 & 0 \\ 0 & A & 0 \\ 0 & 0 & C \end{pmatrix}}_{D} + \underbrace{\begin{pmatrix} 0 & 0 & 0 \\ 0 & 0 & 0 \\ 0 & B & 0 \end{pmatrix}}_{E}.$$

By the triangle inequality for the spectral norm,

$$\|M\|_2 \leq \|D\|_2 + \|E\|_2.$$

Since $D$ is block-diagonal, $\|D\|_2 = \max\{1, \|A\|_2, \|C\|_2\}$, and $E$ has only the single nonzero block $B$, so $\|E\|_2 = \|B\|_2$. Substitution gives the claimed bound. $\square$

**Theorem 3** (Supremum-$p$–Wasserstein contraction of the **complete** Sobolev Bellman operator). *Let*

$$T_\pi^{S_{s,a}} \colon \left( \mathcal{S} \times \mathcal{A} \to \mathcal{P}(\mathbb{R}^{1+m+n}) \right) \longrightarrow \left( \mathcal{S} \times \mathcal{A} \to \mathcal{P}(\mathbb{R}^{1+m+n}) \right),$$

*be the complete Sobolev Bellman operator bootstrapping the vector $(Z, \partial_a Z, \partial_s Z)$.*

*The distributional Sobolev Bellman operators are defined by the reparameterized policy $\pi(s'; \varepsilon_\pi)$, the transition $f(s, a; \varepsilon_f)$ and reward $r(s, a; \varepsilon_r)$. By assumptions from Section C.1 we write*

$$\|\mathcal{L}\|_d = \sup_{s,a} \sup_{\varepsilon_f, \varepsilon_\pi} \left\| \mathcal{L}(s, a; \varepsilon_f, \varepsilon_\pi) \right\|_2 \leq \gamma \left( \max\{1, L_{f,a} L_\pi, L_{f,s}\} + L_{f,s} L_\pi \right) = \gamma \kappa_{\text{full}},$$

*then for any two Sobolev return-distribution functions $\eta_1, \eta_2$,*

$$\boxed{ \bar{W}_p\left(T_\pi^{S_{s,a}} \eta_1, T_\pi^{S_{s,a}} \eta_2\right) \leq \|\mathcal{L}\|_d \, \bar{W}_p(\eta_1, \eta_2) \leq \gamma \kappa_{\text{full}} \, \bar{W}_p(\eta_1, \eta_2). }$$

*In particular, $T_\pi^{S_{s,a}}$ is a $\|\mathcal{L}\|_d$–contraction whenever $\|\mathcal{L}\|_d < 1$, and a sufficient condition for this is $\gamma \kappa_{\text{full}} < 1$.*

*Proof.* We show that the Sobolev Bellman map $T_\pi^{S_{s,a}}$ is a $\|\mathcal{L}\|_d$–contraction in the supremum–$p$–Wasserstein metric.

By definition we have,

$$\bar{W}_p\left(T_\pi^{S_{s,a}} \eta_1, T_\pi^{S_{s,a}} \eta_2\right) = \sup_{s,a} W_p\left(T_\pi^{S_{s,a}} \eta_1(s, a), T_\pi^{S_{s,a}} \eta_2(s, a)\right).$$

Let's fix an arbitrary pair $(s, a)$. Then we draw the *same* exogenous noises for both updates

$$\varepsilon_r \sim p(\varepsilon_r), \quad \varepsilon_f \sim p(\varepsilon_f), \quad \varepsilon_\pi \sim p(\varepsilon_\pi),$$

set

$$r = r(s, a; \varepsilon_r), \quad s' = f(s, a; \varepsilon_f), \quad a' = \pi(s'; \varepsilon_\pi),$$

and sample

$$X_1' \sim \eta_1^{S_{s,a}}(s', a'), \qquad X_2' \sim \eta_2^{S_{s,a}}(s', a').$$

Define the random affine map

$$\Phi_{s,a}\left(x; \varepsilon_r, \varepsilon_f, \varepsilon_\pi\right) = b\left(s, a; \varepsilon_r\right) + \mathcal{L}\left(s, a; \varepsilon_f, \varepsilon_\pi\right)[x],$$

$$\text{where } b = \begin{pmatrix} r \\ \partial_a r \\ \partial_s r \end{pmatrix}, \quad \mathcal{L} = \gamma \begin{pmatrix} 1 & 0 & 0 \\ 0 & f_a^\top \pi_s^\top & 0 \\ 0 & f_s^\top \pi_s^\top & f_s^\top \end{pmatrix}$$

as in Equation 22.

By Lemma 2, we have

$$T_\pi^{S_{s,a}} \eta_i(s,a) = \mathrm{Law}\big(\Phi_{s,a}(X_i')\big), \qquad i = 1, 2,$$

so

$$W_p\big(T_\pi^{S_{s,a}} \eta_1(s,a),\, T_\pi^{S_{s,a}} \eta_2(s,a)\big) = W_p\big(\mathrm{Law}(\Phi_{s,a}(X_1')),\, \mathrm{Law}(\Phi_{s,a}(X_2'))\big).$$

Since $\mathrm{Law}(X_i') = \eta_i^{S_{s,a}}(s', a')$, for every $(s', a')$,

$$W_p\big(\mathrm{Law}(X_1'),\, \mathrm{Law}(X_2')\big) = W_p\big(\eta_1^{S_{s,a}}(s', a'),\, \eta_2^{S_{s,a}}(s', a')\big) \le \bar{W}_p(\eta_1, \eta_2).$$

Condition on the full noise triple $(\varepsilon_r, \varepsilon_f, \varepsilon_\pi)$, so that $\Phi_{s,a}$ is a deterministic affine map. By Lemma 3,

$$W_p\big(\mathrm{Law}(\Phi_{s,a}(X_1') \mid \varepsilon_r, \varepsilon_f, \varepsilon_\pi),\, \mathrm{Law}(\Phi_{s,a}(X_2') \mid \varepsilon_r, \varepsilon_f, \varepsilon_\pi)\big)$$
$$\le \|\mathcal{L}(s, a; \varepsilon_f, \varepsilon_\pi)\|_d\, W_p\big(\eta_1^{S_{s,a}}(s', a'),\, \eta_2^{S_{s,a}}(s', a')\big)$$
$$\le \|\mathcal{L}(s, a; \varepsilon_f, \varepsilon_\pi)\|_d\, \bar{W}_p(\eta_1, \eta_2).$$

Taking the supremum over $\varepsilon_f, \varepsilon_\pi$ and then applying Lemma 4 yields

$$W_p\big(\mathrm{Law}(\Phi_{s,a}(X_1')),\, \mathrm{Law}(\Phi_{s,a}(X_2'))\big) \le \sup_{\varepsilon_f, \varepsilon_\pi} \|\mathcal{L}(s, a; \varepsilon_f, \varepsilon_\pi)\|_d\, \bar{W}_p(\eta_1, \eta_2).$$

Using $T_\pi^{S_{s,a}} \eta_i(s,a) = \mathrm{Law}(\Phi_{s,a}(X_i'))$, we conclude

$$W_p\big(T_\pi^{S_{s,a}} \eta_1(s,a),\, T_\pi^{S_{s,a}} \eta_2(s,a)\big) = W_p\big(\mathrm{Law}(\Phi_{s,a}(X_1')),\, \mathrm{Law}(\Phi_{s,a}(X_2'))\big)$$
$$\le \Big(\sup_{\varepsilon_f, \varepsilon_\pi} \|\mathcal{L}(s, a; \varepsilon_f, \varepsilon_\pi)\|_d\Big) \bar{W}_p(\eta_1, \eta_2).$$

Finally, taking the supremum over $(s, a)$ yields

$$\bar{W}_p\big(T_\pi^{S_{s,a}} \eta_1,\, T_\pi^{S_{s,a}} \eta_2\big) = \sup_{s,a} W_p\big(T_\pi^{S_{s,a}} \eta_1(s,a),\, T_\pi^{S_{s,a}} \eta_2(s,a)\big)$$
$$\le \sup_{s,a}\Big(\sup_{\varepsilon_f, \varepsilon_\pi} \|\mathcal{L}(s, a; \varepsilon_f, \varepsilon_\pi)\|_d\Big) \bar{W}_p(\eta_1, \eta_2)$$
$$= \|\mathcal{L}\|_d\, \bar{W}_p(\eta_1, \eta_2)$$

so $T_\pi^{S_{s,a}}$ is a contraction with coefficient $\|\mathcal{L}\|_d$.

By Section C.1, we have

$$\|f_a(s, a; \varepsilon_f)\|_2 \le L_{f,a}, \quad \|f_s(s, a; \varepsilon_f)\|_2 \le L_{f,s}, \quad \|\pi_s(s'; \varepsilon_\pi)\|_2 \le L_\pi.$$

Hence for each $(s, a, \varepsilon_f, \varepsilon_\pi)$ and applying Lemma 6 we have

$$\big\|\mathcal{L}(s, a; \varepsilon_f, \varepsilon_\pi)\big\|_2 = \gamma \left\|\begin{pmatrix} 1 & 0 & 0 \\ 0 & f_a^\top \pi_s^\top & 0 \\ 0 & f_s^\top \pi_s^\top & f_s^\top \end{pmatrix}\right\|_2$$
$$\le \gamma\Big(\underbrace{\max\{1,\, \|f_a^\top \pi_s^\top\|_2,\, \|f_s^\top\|_2\}}_{\text{diagonal blocks}} + \underbrace{\|f_s^\top \pi_s^\top\|_2}_{\text{off-diagonal block } B}\Big)$$
$$\le \gamma\Big(\max\{1,\, \|f_a\|_2 \|\pi_s\|_2,\, \|f_s\|_2\} + \|f_s\|_2 \|\pi_s\|_2\Big)$$
$$\le \gamma\Big(\max\{1,\, L_{f,a} L_\pi,\, L_{f,s}\} + L_{f,s} L_\pi\Big) = \gamma\, \kappa_{\text{full}}.$$

Since the last inequality holds for every choice of $(s, a, \varepsilon_f, \varepsilon_\pi)$, taking the supremum gives

$$\|\mathcal{L}\|_d = \sup_{s, a, \varepsilon_f, \varepsilon_\pi} \big\|\mathcal{L}(s, a; \varepsilon_f, \varepsilon_\pi)\big\|_2 \le \gamma\, \kappa_{\text{full}}.$$

$\square$

**Lemma 7** (Fixed-point law of the **complete** Sobolev Bellman operator). *Define the infinite-horizon return and its full action- and state-gradients under policy $\pi$ by*

$$Z(s,a) = \sum_{t=0}^{\infty} \gamma^t r_t, \qquad G^a(s,a) = \partial_a \sum_{t=0}^{\infty} \gamma^t r_t, \qquad G^s(s,a) = \partial_s \sum_{t=0}^{\infty} \gamma^t r_t,$$

*and let*

$$\eta^\pi(s,a) = \mathrm{Law}\big(Z(s,a), G^a(s,a), G^s(s,a)\big).$$

*Then $\eta^\pi$ is a fixed point of the complete Sobolev Bellman operator:*

$$\boxed{T_\pi^{S_{s,a}}\, \eta^\pi \;=\; \eta^\pi.}$$

*Proof.* Recalling the one-step affine update

$$\Phi_{s,a}(x;\, \varepsilon_r, \varepsilon_f, \varepsilon_\pi) = b_{\mathrm{full}}(s,a;\varepsilon_r) + \mathcal{L}_{\mathrm{full}}(s,a;\varepsilon_f,\varepsilon_\pi)[\,x\,],$$

the Bellman recursion and its derivatives combine to

$$\big(Z(s,a),\, G^a(s,a),\, G^s(s,a)\big) = \Phi_{s,a}\big(Z(s',a'),\, G^a(s',a'),\, G^s(s',a')\big).$$

By Lemma 2 we have

$$\mathrm{Law}\big(Z(s,a), G^a(s,a), G^s(s,a)\big) = \mathrm{Law}\Big(\Phi_{s,a}\big(Z(s',a'), G^a(s',a'), G^s(s',a')\big)\Big)$$

$$= \big(T_\pi^{S_{s,a}}\, \eta^\pi\big)(s,a).$$

Since this holds for every $(s,a)$, we conclude $T_\pi^{S_{s,a}} \eta^\pi = \eta^\pi$. $\qquad\square$

**Theorem 4** (Supremum-$p$–Wasserstein contraction of the **incomplete** Sobolev Bellman operator). *Let*

$$T_\pi^{S_a} : \big(\mathcal{S} \times \mathcal{A} \to \mathcal{P}(\mathbb{R}^{1+m})\big) \;\longrightarrow\; \big(\mathcal{S} \times \mathcal{A} \to \mathcal{P}(\mathbb{R}^{1+m})\big)$$

*be the action–gradient Sobolev Bellman operator updating only $H(s,a) = (Z(s,a), \partial_a Z(s,a))$.*

*Let $p \geq 1$. Fix any two return–gradient laws $\eta_1, \eta_2 : \mathcal{S} \times \mathcal{A} \to \mathcal{P}(\mathbb{R}^{1+m})$. Assume*

$$\|f_a\|_2 \leq L_{f,a}, \quad \|\pi_s\|_2 \leq L_\pi.$$

*(**Reparameterized pathwise gradients and coupled lifting**). For each $(s',a')$, assume there exist random variables*

$$(H_1', G_1') \in \mathbb{R}^{1+m} \times \mathbb{R}^n, \qquad (H_2', G_2') \in \mathbb{R}^{1+m} \times \mathbb{R}^n,$$

*defined on a common probability space, such that $H_i' = (Z_i', \partial_a Z_i') \sim \eta_i(s',a')$ and $G_i' = \partial_s Z_i'(s',a')$ is the pathwise (reparameterization) derivative as in Appendix A and is induced by our distribution parametrization (Appendix B.2). Moreover, assume that there exists an* optimal *coupling $\zeta^* \in \Pi(\eta_1(s',a'), \eta_2(s',a'))$ such that $(H_1', H_2') \sim \zeta^*$ and the joint law of $(H_1', G_1', H_2', G_2')$ is compatible with $\zeta^*$, meaning that its $(H_1', H_2')$-marginal equals $\zeta^*$ and it couples the corresponding pathwise derivatives $G_1', G_2'$.*

*Assume there is a constant $L_{\mathcal{D}_s} \geq 0$ such that for every $(s',a')$ and for the above coupling,*

$$\big(\mathbb{E}\|G_1' - G_2'\|_2^p\big)^{1/p} \;\leq\; L_{\mathcal{D}_s} \big(\mathbb{E}\|H_1' - H_2'\|_2^p\big)^{1/p}. \tag{24}$$

*Set*

$$\kappa_{\mathrm{eff}} = \max\{1,\, L_{f,a} L_\pi\} + L_{f,a}\, L_{\mathcal{D}_s}.$$

*Then*

$$\boxed{\bar{W}_p\big(T_\pi^{S_a}\eta_1,\, T_\pi^{S_a}\eta_2\big) \;\leq\; \gamma\, \kappa_{\mathrm{eff}}\, \bar{W}_p(\eta_1, \eta_2).}$$

*If $\gamma\, \kappa_{\mathrm{eff}} < 1$, $T_\pi^{S_a}$ is a strict contraction.*

*Proof.* All vector norms $\|\cdot\|$ are Euclidean norms, and operator norms are the corresponding induced norms. All $W_p$ distances are taken with the metric $d(x, y) = \|x - y\|$.

Fix $(s, a)$ and draw one sample $(\varepsilon_r, \varepsilon_f, \varepsilon_\pi)$. Set $c = (\varepsilon_r, \varepsilon_f, \varepsilon_\pi)$ and write

$$s' = f(s, a; \varepsilon_f), \quad a' = \pi(s'; \varepsilon_\pi),$$

and define the (pseudo-)affine update acting on the extended pair $(H', G')$ by

$$\Phi(H', G') = b(s, a; \varepsilon_r) + A(s, a; \varepsilon_f, \varepsilon_\pi) H' + N(s, a; \varepsilon_f) G'.$$

For this fixed noise draw $c$, let $T_{\pi,c}^{S_a}$ denote the corresponding one-step update, so that $(T_{\pi,c}^{S_a} \eta)(s, a) = \mathrm{Law}\big(\Phi(H', G') \mid c\big)$, where the remaining randomness is only that of the next-step draw $H' \sim \eta(s', a')$ and its attached pathwise gradient $G'$.

By the uniform Jacobian bounds of Section C.1,

$$\sup_{s,a,\varepsilon_f} \|f_a(s, a; \varepsilon_f)\| \leq L_{f,a}, \qquad \sup_{s',\varepsilon_\pi} \|\pi_s(s'; \varepsilon_\pi)\| \leq L_\pi,$$

so

$$\|A(s, a; \varepsilon_f, \varepsilon_\pi)\|_2 = \gamma \max\{1, \|f_a(s, a; \varepsilon_f)\|_2 \|\pi_s(s'; \varepsilon_\pi)\|_2\} \leq \gamma \max\{1, L_{f,a} L_\pi\},$$

and similarly

$$\|N(s, a; \varepsilon_f)\|_2 = \gamma \|f_a(s, a; \varepsilon_f)\|_2 \leq \gamma L_{f,a}.$$

Let $(H_1', G_1', H_2', G_2')$ be coupled as in the theorem assumptions, so that $(H_1', H_2') \sim \zeta^*$ is an optimal coupling of $\eta_1(s', a')$ and $\eta_2(s', a')$. Write $H_i' = X_i' \in \mathbb{R}^{1+m}$. Then

$$
\begin{aligned}
\|\Phi(X_1', G_1') - \Phi(X_2', G_2')\| &= \big\|A(X_1' - X_2') + N(G_1' - G_2')\big\| & \text{(definition of } \Phi) \\
&\leq \|A(X_1' - X_2')\| + \|N(G_1' - G_2')\| & \text{(triangle inequality)} \\
&\leq \|A\| \|X_1' - X_2'\| + \|N\| \|G_1' - G_2'\| & \text{(operator-norm bound)}.
\end{aligned}
$$

Taking $p$th powers, expectations, and $p$th roots, and using Minkowski's inequality yields

$$
\begin{aligned}
\Big(\mathbb{E}&\|\Phi(X_1', G_1') - \Phi(X_2', G_2')\|^p\Big)^{1/p} \\
&\leq \Big(\mathbb{E}\big(\|A\| \|X_1' - X_2'\| + \|N\| \|G_1' - G_2'\|\big)^p\Big)^{1/p} \quad \text{(raise to } p\text{, take } \mathbb{E}\text{, take } p\text{th root)} \\
&\leq \|A\|\Big(\mathbb{E}\|X_1' - X_2'\|^p\Big)^{1/p} + \|N\|\Big(\mathbb{E}\|G_1' - G_2'\|^p\Big)^{1/p} \quad \text{(Minkowski inequality)}.
\end{aligned}
$$

Invoking the derivative–coupling bound Equation 24 gives

$$\Big(\mathbb{E}\|\Phi(X_1', G_1') - \Phi(X_2', G_2')\|^p\Big)^{1/p} \leq \big(\|A\| + \|N\| L_{\mathcal{D}_s}\big)\Big(\mathbb{E}\|X_1' - X_2'\|^p\Big)^{1/p}.$$

Since $(X_1', X_2') \sim \zeta^*$ is optimal,

$$\Big(\mathbb{E}\|X_1' - X_2'\|^p\Big)^{1/p} = W_p\big(\eta_1(s', a'), \eta_2(s', a')\big).$$

Define $(U, V) := \big(\Phi(X_1', G_1'), \Phi(X_2', G_2')\big)$. Then $\mathrm{Law}(U, V)$ is an admissible coupling of the corresponding one-step updated laws at noise draw $c$, hence

$$W_p\big(T_{\pi,c}^{S_a} \eta_1(s, a), T_{\pi,c}^{S_a} \eta_2(s, a)\big) \leq \Big(\mathbb{E}\|U - V\|^p\Big)^{1/p}.$$

Combining the last inequalities and substituting the bounds on $\|A\|$ and $\|N\|$ gives, for this fixed noise draw $c$,

$$
\begin{aligned}
W_p\big(T_{\pi,c}^{S_a} \eta_1(s, a), \, T_{\pi,c}^{S_a} \eta_2(s, a)\big) &\leq \gamma\big(\max\{1, L_{f,a} L_\pi\} + L_{f,a} L_{\mathcal{D}_s}\big) W_p\big(\eta_1(s', a'), \eta_2(s', a')\big) \\
&= \gamma \kappa_{\mathrm{eff}} W_p\big(\eta_1(s', a'), \eta_2(s', a')\big).
\end{aligned}
$$

By Lemma 5 (mixture $p$-convexity), and using that $W_p(\eta_1(s', a'), \eta_2(s', a')) \leq \bar{W}_p(\eta_1, \eta_2)$,

$$W_p^p\big(T_\pi^{S_a}\eta_1(s, a), \, T_\pi^{S_a}\eta_2(s, a)\big) \leq \mathbb{E}_c\Big[W_p^p\big(T_{\pi,c}^{S_a}\eta_1(s, a), \, T_{\pi,c}^{S_a}\eta_2(s, a)\big)\Big]$$

$$\leq (\gamma\,\kappa_{\text{eff}})^p \, \mathbb{E}_c\Big[W_p^p\big(\eta_1(s', a'), \, \eta_2(s', a')\big)\Big]$$

$$\leq (\gamma\,\kappa_{\text{eff}})^p \, \bar{W}_p^p(\eta_1, \eta_2).$$

Taking $p$th roots and then the supremum over $(s, a)$ yields the claim. $\square$

**Lemma 8** (Fixed-point law of the **incomplete** Sobolev Bellman operator). *Define the infinite-horizon return and its full action-gradient under policy $\pi$ by*

$$Z(s, a) = \sum_{t=0}^\infty \gamma^t \, r_t, \qquad G(s, a) = \partial_a \sum_{t=0}^\infty \gamma^t \, r_t,$$

*and let $\eta^\pi(s, a) = \text{Law}\big(Z(s, a), G(s, a)\big)$. Then $\eta^\pi$ is a fixed point of the Sobolev Bellman operator:*

$$\boxed{T_\pi^{S_a}\,\eta^\pi \; = \; \eta^\pi.}$$

*Proof.* Recalling the one-step affine update

$$\Phi_{s,a}(x; \, \varepsilon_r, \varepsilon_f, \varepsilon_\pi) = b(s, a; \varepsilon_r) + \mathcal{L}(s, a; \varepsilon_f, \varepsilon_\pi)[\, x \,],$$

the Bellman recursion and its derivative combine to

$$\big(Z(s, a), \, G(s, a)\big) = \Phi_{s,a}\big(Z(s', a'), \, G(s', a')\big).$$

By Lemma 2 we have

$$\text{Law}\big(Z(s, a), G(s, a)\big) = \text{Law}\big(\Phi_{s,a}(Z(s', a'), G(s', a'))\big) = (T_\pi^{S_a}\,\eta^\pi)(s, a).$$

Since this holds for every $(s, a)$, we conclude $T_\pi^{S_a}\eta^\pi = \eta^\pi$. $\square$

**Corollary 4.1** (Convergence of Sobolev evaluation iterates). Under the conditions of Theorem 3, let $\kappa$ denote the contraction constant with $\kappa = \kappa_{\text{eff}}$ for the incomplete operator or $\kappa = \kappa_{\text{full}}$ for the complete one and suppose $\gamma\,\kappa < 1$. For any initial Sobolev return–distribution function $\eta_0$, define the sequence

$$\eta_{n+1} = T_\pi^S\,\eta_n,$$

where $T_\pi^S$ may be either the incomplete ($S_a$) or complete ($S_{s,a}$) Sobolev operator. Then by Banach's fixed-point theorem the iterates converge to the unique fixed point $\eta^\pi$ (cf. Lemmas 7, 8):

$$\bar{W}_p\big(\eta_n, \, \eta^\pi\big) \; \leq \; (\gamma\,\kappa)^n \, \bar{W}_p\big(\eta_0, \, \eta^\pi\big) \; \xrightarrow[n\to\infty]{} \; 0.$$

In particular, $\eta_n \to \eta^\pi$ in the supremum–$p$–Wasserstein metric.

### D.2  PROOF OF PROPOSITION 3.1

**Lemma 9** (Mean-difference bound via $W_1$). *Let $X, Y$ be $\mathbb{R}^d$-valued random variables with distributions $\mu = \text{Law}(X)$ and $\nu = \text{Law}(Y)$, and assume $\mathbb{E}\|X\| < \infty$, $\mathbb{E}\|Y\| < \infty$. Then*

$$\boxed{\big\|\mathbb{E}[X] - \mathbb{E}[Y]\big\| \; \leq \; W_1(\mu, \nu),}$$

*Here, $\|\cdot\|$ denotes the Euclidean norm on $\mathbb{R}^d$, and $W_1$ is taken with respect to the ground metric $d(x, y) = \|x - y\|$.*

*Proof.* Let

$$m_X = \mathbb{E}[X], \quad m_Y = \mathbb{E}[Y].$$

If $m_X \neq m_Y$, define the unit vector

$$u = \frac{m_X - m_Y}{\|m_X - m_Y\|}.$$

Then the scalar function $f(x) = u^\top x$ satisfies

$$|f(x) - f(y)| = |u^\top(x - y)| \leq \|u\| \, \|x - y\| = \|x - y\|,$$

so $\|f\|_{\mathrm{Lip}} \leq 1$. By Kantorovich–Rubinstein duality (Villani et al., 2009),

$$W_1(\mu, \nu) = \sup_{\|g\|_{\mathrm{Lip}} \leq 1} \left|\mathbb{E}[g(X)] - \mathbb{E}[g(Y)]\right| \geq \left|\mathbb{E}[f(X)] - \mathbb{E}[f(Y)]\right|.$$

But $\mathbb{E}[f(X)] - \mathbb{E}[f(Y)] = u^\top(m_X - m_Y) = \|m_X - m_Y\|$, hence $\|m_X - m_Y\| \leq W_1(\mu, \nu)$. If $m_X = m_Y$, the inequality is trivial. $\qquad\square$

**Proposition 2.** *Let $\pi$ be an $L_{\pi,\theta}$-Lipschitz continuous policy, and let $\mathrm{Law}[\nabla_a Z^\pi(s, a) \,|_{a=\pi(s)}]$ and $\mathrm{Law}[\nabla_a \hat{Z}(s, a) \,|_{a=\pi(s)}]$ denote the true and estimated distributions of the action-gradients at $a = \pi(s)$, respectively. Then the error between the true and estimated policy gradients is bounded by*

$$\boxed{\begin{aligned}
\left\|\nabla_\theta J(\theta) - \nabla_\theta \hat{J}(\theta)\right\| &\leq \\
\frac{L_{\pi,\theta}}{1 - \gamma} \, \mathbb{E}_{s \sim d_\mu^\pi} &\left[ W_1\big(\mathrm{Law}[\nabla_a Z^\pi(s, a) \,|_{a=\pi(s)}], \, \mathrm{Law}[\nabla_a \hat{Z}(s, a) \,|_{a=\pi(s)}]\big)\right].
\end{aligned}}$$

*This proposition is a distributional extension of Proposition 3.1 from D'Oro & Jaskowski (2020).*

*Proof.*

**Step 1: True and estimated policy gradients.**

The true policy gradient is:

$$\nabla_\theta J(\theta) = \frac{1}{1 - \gamma}\mathbb{E}_{s \sim d_\mu^\pi}\left[\mathbb{E}\left[\nabla_a Z^\pi(s, a)\big|_{a=\pi(s)}\right]\nabla_\theta \pi(s)\right],$$

and the estimated policy gradient is:

$$\nabla_\theta \hat{J}(\theta) = \frac{1}{1 - \gamma}\mathbb{E}_{s \sim d_\mu^\pi}\left[\mathbb{E}\left[\nabla_a \hat{Z}(s, a)\big|_{a=\pi(s)}\right]\nabla_\theta \pi(s)\right].$$

**Step 2: Policy gradient error.** The norm of the difference between the true and estimated policy gradients is:

$$\begin{aligned}
\left\|\nabla_\theta J(\theta) - \nabla_\theta \hat{J}(\theta)\right\| = \Bigg\|\frac{1}{1 - \gamma}\mathbb{E}_{s \sim d_\mu^\pi}\Bigg[&\left(\mathbb{E}\left[\nabla_a Z^\pi(s, a)\big|_{a=\pi(s)}\right] - \mathbb{E}\left[\nabla_a \hat{Z}(s, a)\big|_{a=\pi(s)}\right]\right) \\
&\times \nabla_\theta \pi(s)\Bigg]\Bigg\|.
\end{aligned}$$

**Step 3: Applying the Triangle Inequality and Lipschitz Continuity**

Using the triangle inequality and the Lipschitz continuity of the policy ($\|\nabla_\theta \pi(s)\| \leq L_{\pi,\theta}$), we have:

$$\begin{aligned}
\left\|\nabla_\theta J(\theta) - \nabla_\theta \hat{J}(\theta)\right\| &\leq \frac{1}{1 - \gamma}\mathbb{E}_{s \sim d_\mu^\pi}\Bigg[\left\|\mathbb{E}\left[\nabla_a Z^\pi(s, a)\big|_{a=\pi(s)}\right] - \mathbb{E}\left[\nabla_a \hat{Z}(s, a)\big|_{a=\pi(s)}\right]\right\| \\
&\qquad \times \|\nabla_\theta \pi(s)\|\Bigg] \\
&\leq \frac{L_{\pi,\theta}}{1 - \gamma}\mathbb{E}_{s \sim d_\mu^\pi}\left[\left\|\mathbb{E}\left[\nabla_a Z^\pi(s, a)\big|_{a=\pi(s)}\right] - \mathbb{E}\left[\nabla_a \hat{Z}(s, a)\big|_{a=\pi(s)}\right]\right\|\right].
\end{aligned}$$

**Step 4: Bounding the mean difference by $W_1$.**

Let

$$X = \nabla_a Z^\pi(s, a)\big|_{a=\pi(s)}, \qquad Y = \nabla_a \hat{Z}(s, a)\big|_{a=\pi(s)}.$$

By Lemma 9, we have

$$\begin{aligned}
\left\|\mathbb{E}[X] - \mathbb{E}[Y]\right\| &\leq W_1\big(\mathrm{Law}(X), \, \mathrm{Law}(Y)\big) \\
&= W_1\Big(\mathrm{Law}\big(\nabla_a Z^\pi(s, a)\big|_{a=\pi(s)}\big), \, \mathrm{Law}\big(\nabla_a \hat{Z}(s, a)\big|_{a=\pi(s)}\big)\Big).
\end{aligned}$$

**Step 5: Conclusion**

Combining the results from the previous steps, we established that the $L^2$ norm of the difference between the true and estimated policy gradients can be bounded as follows:

$$\left\| \nabla_\theta J(\theta) - \nabla_\theta \hat{J}(\theta) \right\| \leq$$
$$\frac{L_{\pi,\theta}}{1-\gamma} \, \mathbb{E}_{s\sim d_\mu^\pi} \left[ W_1 \big( \mathrm{Law}[\nabla_a Z^\pi(s,a) \mid_{a=\pi(s)}], \, \mathrm{Law}[\nabla_a \hat{Z}(s,a) \mid_{a=\pi(s)}] \big) \right].$$

$\square$

# E  PRACTICAL DIFFICULTIES WITH WASSERSTEIN FOR TRAINING

**Adversarial $W_1$ is a loose proxy for true Wasserstein.**    Some works have directly applied WGAN training to distributional RL, for example Freirich et al. (2019), who cast return distributions as targets for adversarial matching. In practice, WGAN training replaces the exact Kantorovich–Rubinstein dual (Arjovsky et al., 2017; Gulrajani et al., 2017) with a *parametric* discriminator plus approximate Lipschitz control (weight clipping or gradient penalties). This induces three sources of deviation from the true distance: finite-capacity approximation error, imperfect Lipschitz enforcement, and optimization error. Systematic analyses show that the resulting WGAN losses can correlate poorly with the actual Wasserstein metric and need not be meaningful approximations of it, undermining proofs that presume access to the exact $W_1$. See both the empirical and theoretical analysis of Mallasto et al. (2019) and the critique by Stanczuk et al. (2021).

**Computational cost of exact OT in multiple dimensions.**    Even ignoring estimator issues, computing multivariate $W_p$ exactly on mini-batches is costly: building the pairwise cost matrix requires $O(m^2)$ memory, and solving the discrete OT problem takes $O(m^3 \log m)$ time (Genevay et al., 2019). This makes per-update calls prohibitive in distributional RL.

**Sample complexity.**    Beyond runtime, OT also suffers from poor statistical efficiency: the empirical Wasserstein distance converges to its population value at rate $O(n^{-1/d})$ in dimension $d$, compared to $O(n^{-1/2})$ for MMD (Genevay et al., 2019). This slow convergence further limits its practicality in high-dimensional RL.

## F   BACKGROUND ON MMD

**Definition 3** (Maximum Mean Discrepancy as an IPM). *Let $k\colon X \times X \to \mathbb{R}$ be a symmetric, positive-semi-definite reproducing kernel with RKHS $\mathcal{H}$ and feature map*

$$\phi\colon X \to \mathcal{H}, \qquad \phi(x) = k(x, \cdot).$$

*For probability measures $P, Q$ on $X$, define their mean embeddings*

$$\mu_P = \int_X \phi(x)\, dP(x), \quad \mu_Q = \int_X \phi(x)\, dQ(x).$$

*Then the* Maximum Mean Discrepancy *is*

$$\mathrm{MMD}_k(P, Q) \;:=\; \|\mu_P - \mu_Q\|_{\mathcal{H}},$$

*whose square admits the kernel expansion*

$$\begin{aligned}
\mathrm{MMD}_k^2(P, Q) &= \|\mu_P - \mu_Q\|_{\mathcal{H}}^2 \\
&= \iint k(x, x')\, dP(x)\, dP(x') \;+\; \iint k(y, y')\, dQ(y)\, dQ(y') \\
&\quad -\; 2 \iint k(x, y)\, dP(x)\, dQ(y) \\
&= \iint k(x, y)\, d\big(P - Q\big)(x)\, d\big(P - Q\big)(y).
\end{aligned}$$

*Moreover, $\mathrm{MMD}_k$ coincides with the integral probability metric (IPM) over the unit ball of $\mathcal{H}$, namely*

$$\mathrm{MMD}_k(P, Q) \;=\; \sup_{\substack{f \in \mathcal{H} \\ \|f\|_{\mathcal{H}} \leq 1}} \Big\{ \mathbb{E}_{x \sim P}[f(x)] \;-\; \mathbb{E}_{y \sim Q}[f(y)] \Big\} \;=\; \|\mu_P - \mu_Q\|_{\mathcal{H}},$$

*as shown in Gretton et al. (2012).*

**Remark (Euclidean densities).** Working in $\mathbb{R}^d$, if $P$ and $Q$ admit densities $p(x)$ and $q(x)$ with respect to Lebesgue measure $dx$, then

$$dP(x) = p(x)\, dx, \quad dQ(x) = q(x)\, dx, \quad d\big(P - Q\big)(x) = \big(p(x) - q(x)\big)\, dx,$$

and each of the above integrals becomes an ordinary Lebesgue integral in $x, y$.

**Definition 4** (Conditionally positive definite (CPD) kernel). *Let $X$ be a measurable space and let $k : X \times X \to \mathbb{R}$ be symmetric. We say that $k$ is* conditionally positive definite (CPD) *if*

$$\iint_{X \times X} k(x, x')\, d\mu(x)\, d\mu(x') \;\geq\; 0 \qquad \textit{for all finite signed measures } \mu \textit{ on } X \textit{ with } \mu(X) = 0.$$

*If the inequality is strict for every nonzero such $\mu$, then $k$ is* conditionally strictly positive definite *(CSPD).*

**Proposition 3** (Equivalence of $\gamma_k$ and RKHS–MMD for CPD kernels). *Let $k : X \times X \to \mathbb{R}$ be conditionally positive definite (CPD) and define*

$$\rho_k(x, y) \;:=\; k(x, x) + k(y, y) - 2k(x, y).$$

*Fix $z_0 \in X$ and set the* distance–induced *(one–point centered) kernel*

$$k^\circ(x, y) \;:=\; \tfrac{1}{2}\big[\rho_k(x, z_0) + \rho_k(y, z_0) - \rho_k(x, y)\big] \;=\; k(x, y) - k(x, z_0) - k(z_0, y) + k(z_0, z_0).$$

*Then $k^\circ$ is positive definite and admits an RKHS $\mathcal{H}_{k^\circ}$. For any $P, Q$ with finite integrals,*

$$\begin{aligned}
\gamma_k^2(P, Q) &:= \iint k(x, y)\, d(P - Q)(x)\, d(P - Q)(y) \\
&= \iint k^\circ(x, y)\, d(P - Q)(x)\, d(P - Q)(y) \\
&= \big\| \mu_{k^\circ}(P) - \mu_{k^\circ}(Q) \big\|_{\mathcal{H}_{k^\circ}}^2 \\
&= \mathrm{MMD}_{k^\circ}^2(P, Q).
\end{aligned}$$

Justification. *This follows from the distance–induced kernel construction and equivalence results in Sejdinovic et al. (2013).*

**Proposition 4** (MMD as a Metric on $\mathcal{P}(X)$)**.** *Let $k\colon X \times X \to \mathbb{R}$ be a symmetric kernel. We say that $\mathrm{MMD}_k$ defines a metric on $\mathcal{P}(X)$ iff $k$ is* conditionally strictly positive definite *(CSPD), i.e., for every nonzero finite signed Borel measure $\nu$ with $\nu(X) = 0$,*

$$\iint_{X \times X} k(x,y)\, d\nu(x)\, d\nu(y) \; > \; 0.$$

*Then $\mathrm{MMD}_k$ satisfies the metric axioms on $\mathcal{P}(X)$:*

1. Nonnegativity: $\mathrm{MMD}_k(P,Q) \geq 0$.

2. Symmetry: $\mathrm{MMD}_k(P,Q) = \mathrm{MMD}_k(Q,P)$.

3. Identity of indiscernibles: $\mathrm{MMD}_k(P,Q) = 0 \Rightarrow P = Q$.

4. Triangle inequality: *for any $P, Q, R \in \mathcal{P}(X)$,* $\mathrm{MMD}_k(P,Q) \leq \mathrm{MMD}_k(P,R) + \mathrm{MMD}_k(R,Q)$.

Justification. *This is the standard correspondence between negative-type distances, distance-induced kernels, and RKHS MMD metrics as outlined in Sejdinovic et al. (2013).*

**Examples of kernels inducing a metric.**

- *Gaussian RBF kernel:* for any bandwidth $\sigma > 0$,

$$k_\sigma^{\mathrm{RBF}}(x,y) \;=\; \exp\!\big(-\|x-y\|_2^2/(2\sigma^2)\big),$$

which is characteristic on $\mathbb{R}^d$ and hence induces a metric on $\mathcal{P}(\mathbb{R}^d)$ via $\mathrm{MMD}_{k_\sigma^{\mathrm{RBF}}}$.

- *Multiquadric kernel* (Killingberg & Langseth (2023)):

$$k_h^{\mathrm{MQ}}(x,y) \;=\; -\sqrt{1 + h^2\,\|x-y\|_2^2}, \quad h > 0,$$

which is conditionally strictly positive-definite and thus induces a metric on distributions via $\mathrm{MMD}_{k_h^{\mathrm{MQ}}}$.

## F.1 Contraction under MMD

Contraction guarantees under MMD can be established in much the same way as in Appendix D. This first requires defining the notion of the supremum-MMD, which, as with the supremum-Wasserstein distance, is a worst-case bound over the entire state–action space:

$$\mathrm{MMD}_\infty(\eta,\nu) := \sup_{(s,a)\in\mathcal{S}\times\mathcal{A}} \mathrm{MMD}\big(\eta(s,a),\,\nu(s,a)\big).$$

As Nguyen et al. (2020) first introduced MMD-based distributional reinforcement learning, they provided criteria under which a kernel induces a contraction in this sup-MMD metric with the standard distributional Bellman operator. We first recall the univariate distributional Bellman operator Bellemare et al. (2017):

$$\big(\mathcal{T}_\pi^{\mathrm{Dist}}\eta\big)(s,a) = \mathrm{Law}\big[R(s,a) + \gamma\,Z(s',\pi(s'))\big], \quad s' \sim P(\cdot \mid s,a).$$

**Sufficient conditions.** Let $k(x,y) = \sum_{i\in I} c_i\,k_i(x,y)$ be a positive-definite kernel on $\mathbb{R}$. If each component $k_i$ satisfies:

1. *Shift-invariance*: $k_i(x+c, y+c) = k_i(x,y)$ for all $c \in \mathbb{R}$,

2. *Scale-sensitivity* of order $\alpha_i$: $k_i(c\,x, c\,y) = |c|^{\alpha_i} k_i(x,y)$ for all $c \in \mathbb{R}$,

then for any policy $\pi$,

$$\mathrm{MMD}_\infty\big(\mathcal{T}_\pi^{\mathrm{Dist}}\eta,\ \mathcal{T}_\pi^{\mathrm{Dist}}\nu\big) \ \leq \ \gamma^{\alpha_*/2}\,\mathrm{MMD}_\infty(\eta,\nu),$$

where $\alpha_* = \min_{i\in I}\alpha_i$.

**Contraction under the multiquadric kernel.** Another work, whose kernel we use, Killingberg & Langseth (2023) proposed the multiquadric kernel discussed above. They showed a contraction in $\mathrm{MMD}^2$ for that specific kernel. More precisely, this holds for any pair of distributions $\mu,\nu \in \mathcal{P}(\mathbb{R})$, letting $(f_{r,\gamma})_\#\mu$ denote the pushforward of $\mu$ by $z \mapsto r + \gamma z$, they show

$$\mathrm{MMD}^2\big((f_{r,\gamma})_\#\mu,\ (f_{r,\gamma})_\#\nu;\ k_h^{\mathrm{MQ}}\big) \ \leq \ \gamma\,\mathrm{MMD}^2\big(\mu,\nu;\ k_h^{\mathrm{MQ}}\big).$$

Taking square–roots on both sides gives the pointwise MMD bound

$$\mathrm{MMD}\big((f_{r,\gamma})_\#\mu,\ (f_{r,\gamma})_\#\nu;\ k_h^{\mathrm{MQ}}\big) \ \leq \ \sqrt{\gamma}\,\mathrm{MMD}\big(\mu,\nu;\ k_h^{\mathrm{MQ}}\big).$$

Define the supremum–MMD over state–action pairs by

$$\mathrm{MMD}_\infty(\eta,\nu) := \sup_{(s,a)\in\mathcal{S}\times\mathcal{A}} \mathrm{MMD}\big(\eta(s,a),\ \nu(s,a)\big).$$

Then for any two return–distribution mappings $\eta,\nu$,

$$\begin{aligned}
\mathrm{MMD}_\infty\big(\mathcal{T}_\pi^{\mathrm{Dist}}\eta,\ \mathcal{T}_\pi^{\mathrm{Dist}}\nu\big) &= \sup_{(s,a)} \mathrm{MMD}\big((f_{R(s,a),\gamma})_\#\eta(s,a),\ (f_{R(s,a),\gamma})_\#\nu(s,a)\big) \\
&\leq \sup_{(s,a)} \sqrt{\gamma}\,\mathrm{MMD}\big(\eta(s,a),\ \nu(s,a)\big) \\
&= \sqrt{\gamma}\,\mathrm{MMD}_\infty(\eta,\nu).
\end{aligned}$$

Thus the distributional Bellman operator is a $\sqrt{\gamma}$–contraction in $\mathrm{MMD}_\infty$.

**Contraction in the multivariate setting.** To the best of our knowledge, the only MMD-contraction result in a multivariate setting is from Wiltzer et al. (2024). They show that when each sampled return vector $Z \in \mathbb{R}^d$ is pushed through the affine map $z \mapsto \mathbf{R}(s,a) + \gamma z$, the same two requirements, shift-invariance and homogeneity of the kernel, guarantee a $\gamma^{\alpha/2}$ contraction in the supremum-MMD metric. In other words, by treating each component of the return vector uniformly and applying the identical homogeneity-based argument from the univariate case, one obtains exactly the same geometric shrinkage factor.

This, however, falls short of the setting we require for the distributional Sobolev Bellman operator in Appendix C, where the pushforward is the more general (pseudo)-affine map

$$x \ \mapsto \ \Phi_{s,a}(x) = b(s,a) + \mathcal{L}(s,a)[\,x\,],$$

and $\mathcal{L}(s,a)$ need not be a simple diagonal scaling. **Characterizing the precise conditions on both the kernel and the (pseudo-)linear operator $\mathcal{L}(s,a)$ under which this general $\Phi_{s,a}$ yields a contraction in supremum-MMD remains an open problem.**

### F.2 EMPIRICAL ESTIMATORS OF MMD

In practice, the expectations in Equation 17 cannot be computed exactly and must be approximated from samples. Given two sets of $m$ samples $\{x_i\}_{i=1}^m \sim P$ and $\{y_i\}_{i=1}^m \sim Q$, one commonly used estimator is the *biased* form (Gretton et al., 2012):

$$\widehat{\mathrm{MMD}}_b^2 = \frac{1}{m^2}\sum_{i,j=1}^m k(x_i,x_j) + \frac{1}{m^2}\sum_{i,j=1}^m k(y_i,y_j) - \frac{2}{m^2}\sum_{i,j=1}^m k(x_i,y_j). \tag{25}$$

An alternative is the *unbiased* estimator proposed in (Gretton et al., 2012):

$$\widehat{\mathrm{MMD}}_u^2 = \frac{1}{m(m-1)}\sum_{\substack{i,j=1\\i\neq j}}^m k(x_i,x_j) + \frac{1}{m(m-1)}\sum_{\substack{i,j=1\\i\neq j}}^m k(y_i,y_j) - \frac{2}{m^2}\sum_{i,j=1}^m k(x_i,y_j). \tag{26}$$

The biased estimator given above is commonly preferred over the unbiased one in works involving MMD for distributional RL (Nguyen et al., 2020; Killingberg & Langseth, 2023).

# G    RESULTS WITH MAX SLICED MMD (MSMMD)

In this section, we study max–sliced MMD contractions for Sobolev Bellman updates. Theorem 5 gives a contraction result for the *complete* Sobolev Bellman operator $T_\pi^{S_s,a}$ under the supremum–max–sliced $\mathrm{MMD}_k$ discrepancy, under appropriate conditions on the underlying kernel $k$. Theorem 6 establishes an analogous contraction statement for the *incomplete* action–gradient operator $T_\pi^{S_a}$, under a distributional lifting assumption.

**Lemma 10** (Scale–contraction of $\mathrm{MMD}^2$ under the multiquadric kernel). *Let $h > 0$ and consider the (negative) multiquadric kernel*

$$k_h(x, y) \;=\; -\sqrt{1 + h^2 \|x - y\|^2}, \qquad x, y \in \mathbb{R}^d.$$

*For probability measures $\mu, \nu$ on $\mathbb{R}^d$ with finite second moments, define*

$$\mathrm{MMD}_{k_h}^2(\mu, \nu) \;=\; \mathbb{E}\, k_h(X, X') + \mathbb{E}\, k_h(Y, Y') - 2\,\mathbb{E}\, k_h(X, Y),$$

*for $X, X' \sim \mu$ i.i.d. and $Y, Y' \sim \nu$ i.i.d.*

*For the scaling map $S_s : x \mapsto sx$ with $s \in [0, 1]$, we have*

$$\mathrm{MMD}_{k_h}^2\big((S_s)_\# \mu, (S_s)_\# \nu\big) \;\leq\; s\, \mathrm{MMD}_{k_h}^2(\mu, \nu), \qquad 0 \leq s \leq 1. \tag{27}$$

*Consequently,*

$$\mathrm{MMD}_{k_h}\big((S_s)_\# \mu, (S_s)_\# \nu\big) \;\leq\; \sqrt{s}\, \mathrm{MMD}_{k_h}(\mu, \nu), \qquad 0 \leq s \leq 1. \tag{28}$$

*Proof.* If $s = 0$, the measures collapse to Dirac deltas at the origin, yielding MMD zero on both sides, so the bound holds trivially. We assume $0 < s \leq 1$ henceforth.

For any $u > 0$ we use the identity (see (Schilling et al., 2012, Chapter 15.2, pp. 218–219)):

$$\sqrt{u} = \frac{1}{2\sqrt{\pi}} \int_0^\infty \big(1 - e^{-tu}\big)\, t^{-3/2}\, dt. \tag{29}$$

Multiplying by $-1$ and substituting $u = 1 + h^2 \|x - y\|^2$ yields the representation:

$$k_h(x, y) = \frac{1}{2\sqrt{\pi}} \int_0^\infty \big(e^{-t(1 + h^2 \|x - y\|^2)} - 1\big)\, t^{-3/2}\, dt. \tag{30}$$

Substituting Equation 30 into the definition of $\mathrm{MMD}_{k_h}^2$ leads to an expectation of integrals:

$$\mathrm{MMD}_{k_h}^2(\mu, \nu) = \mathbb{E}\Bigg[\frac{1}{2\sqrt{\pi}} \Bigg( \int_0^\infty (e^{-t(1 + h^2 \|X - X'\|^2)} - 1)\, t^{-3/2}\, dt$$

$$+ \int_0^\infty (e^{-t(1 + h^2 \|Y - Y'\|^2)} - 1)\, t^{-3/2}\, dt$$

$$- 2 \int_0^\infty (e^{-t(1 + h^2 \|X - Y\|^2)} - 1)\, t^{-3/2}\, dt \Bigg)\Bigg].$$

To interchange the expectation $\mathbb{E}$ and the integral $\int_0^\infty dt$, we invoke Fubini's theorem. For this, it suffices to show

$$\int_0^\infty \mathbb{E}\Big[\big|e^{-t(1 + h^2 \|X - X'\|^2)} - 1\big|\, t^{-3/2}\Big]\, dt < \infty,$$

since the two other terms are treated analogously. We split the integration domain as $\int_0^\infty = \int_0^1 + \int_1^\infty$ and bound each part separately:

- **For $0 < t \leq 1$:** using the inequality $|e^{-z} - 1| \leq z$ for $z \geq 0$, we obtain
$$|e^{-tu} - 1| \leq tu,$$
for any $u \geq 0$. Thus the integrand is bounded by a constant multiple of $t^{-1/2}u$. Since $\int_0^1 t^{-1/2}\, dt < \infty$, the integral over $(0, 1]$ is finite whenever $\mathbb{E}[u] < \infty$. In the present setting, for the first term we have $u = 1 + h^2 \|X - X'\|^2$, so
$$\mathbb{E}[u] = 1 + h^2\, \mathbb{E}\|X - X'\|^2 < \infty,$$
which follows from the finite second moments of $\mu$.

- **For** $t > 1$: using $|e^{-z} - 1| \leq 1$ for $z \geq 0$, we have
$$|e^{-tu} - 1| \leq 1,$$
so the integrand is bounded by a constant multiple of $t^{-3/2}$. Since $\int_1^\infty t^{-3/2}\, dt < \infty$, the integral over $[1, \infty)$ is finite.

Hence

$$\int_0^\infty \mathbb{E}\Big[\big|e^{-t(1+h^2\|X-X'\|^2)} - 1\big|\, t^{-3/2}\Big]\, dt = \int_0^1 (\cdots)\, dt + \int_1^\infty (\cdots)\, dt < \infty,$$

so the absolute integrability condition is satisfied and Fubini's theorem justifies the interchange of expectation and integration:

$$\mathrm{MMD}_{k_h}^2(\mu, \nu) = \frac{1}{2\sqrt{\pi}} \int_0^\infty \mathbb{E}\underbrace{\begin{bmatrix} e^{-t(1+h^2\|X-X'\|^2)} - 1 \\ + e^{-t(1+h^2\|Y-Y'\|^2)} - 1 \\ - 2(e^{-t(1+h^2\|X-Y\|^2)} - 1) \end{bmatrix}}_{A(t)} t^{-3/2}\, dt.$$

The constant terms $(-1 - 1 + 2)$ cancel out in $A(t)$, so only the exponential parts remain.

We now introduce the Gaussian kernel

$$k_\gamma^G(x, y) = e^{-\gamma\|x-y\|^2}, \qquad \gamma > 0,$$

and its associated squared MMD

$$D_G^2(\gamma; \mu, \nu) = \mathbb{E}\, k_\gamma^G(X, X') + \mathbb{E}\, k_\gamma^G(Y, Y') - 2\,\mathbb{E}\, k_\gamma^G(X, Y).$$

With this notation, we can rewrite $A(t)$ as

$$A(t) = e^{-t}\Big(\mathbb{E}\, e^{-th^2\|X-X'\|^2} + \mathbb{E}\, e^{-th^2\|Y-Y'\|^2} - 2\,\mathbb{E}\, e^{-th^2\|X-Y\|^2}\Big) = e^{-t} D_G^2(th^2; \mu, \nu).$$

This yields the integral mixture representation:

$$\mathrm{MMD}_{k_h}^2(\mu, \nu) = \frac{1}{2\sqrt{\pi}} \int_0^\infty e^{-t} t^{-3/2} D_G^2(th^2; \mu, \nu)\, dt. \tag{31}$$

Now let $\mu_s = (S_s)_\# \mu$ and $\nu_s = (S_s)_\# \nu$. The MMD under the Gaussian kernel scales as:

$$D_G^2(\gamma; \mu_s, \nu_s) = D_G^2(\gamma s^2; \mu, \nu).$$

We apply Equation 31 to $\mu_s, \nu_s$ and make the change of variables $u = ts^2$. Then $t = u/s^2$, $dt = s^{-2} du$, and the measure transforms as:

$$t^{-3/2} dt = (u/s^2)^{-3/2}(s^{-2} du) = s^3 u^{-3/2} s^{-2} du = s\, u^{-3/2} du.$$

Substituting this yields:

$$\mathrm{MMD}_{k_h}^2(\mu_s, \nu_s) = \frac{s}{2\sqrt{\pi}} \int_0^\infty e^{-u/s^2}\, u^{-3/2} D_G^2(uh^2; \mu, \nu)\, du. \tag{32}$$

For $0 < s \leq 1$, we observe that $1/s^2 \geq 1$, which implies the pointwise bound:

$$e^{-u/s^2} \leq e^{-u}, \qquad \text{for all } u > 0.$$

Since the remaining integrand factor $u^{-3/2} D_G^2(uh^2; \mu, \nu)$ is always non-negative, this inequality is preserved upon integration. Combining this observation with Equation 32 gives

$$\mathrm{MMD}_{k_h}^2(\mu_s, \nu_s) \leq \frac{s}{2\sqrt{\pi}} \int_0^\infty e^{-u}\, u^{-3/2} D_G^2(uh^2; \mu, \nu)\, du. \tag{33}$$

Comparing Equation 33 with Equation 31 yields:

$$\mathrm{MMD}_{k_h}^2(\mu_s, \nu_s) \leq s\, \mathrm{MMD}_{k_h}^2(\mu, \nu).$$

Taking square roots concludes the proof. $\qquad\square$

**Lemma 11** (Mixture $p$–convexity of $\mathrm{MMD}_k$ in an RKHS). *Let $k : X \times X \to \mathbb{R}$ be a symmetric positive–semidefinite reproducing kernel with RKHS $(\mathcal{H}, \langle \cdot, \cdot \rangle)$ and feature map $\phi(x) = k(x, \cdot)$. Let $(\Omega, \mathcal{F}, \rho)$ be a probability space, and let $(\mu_c)_{c \in \Omega}$ and $(\nu_c)_{c \in \Omega}$ be families of probability measures on $X$ such that the mean embeddings $\mu_{\mu_c} := \int_X \phi \, d\mu_c$ and $\mu_{\nu_c} := \int_X \phi \, d\nu_c$ exist in $\mathcal{H}$. Define the mixtures $\bar{\mu} := \int_\Omega \mu_c \, \rho(dc)$ and $\bar{\nu} := \int_\Omega \nu_c \, \rho(dc)$, and assume that $\mu_{\bar{\mu}}$, $\mu_{\bar{\nu}}$, $\int_\Omega \mu_{\mu_c} \, \rho(dc)$, and $\int_\Omega \mu_{\nu_c} \, \rho(dc)$ are well defined in $\mathcal{H}$. Then for every $p \in [1, \infty)$,*

$$\mathrm{MMD}_k(\bar{\mu}, \bar{\nu}) \; \leq \; \left( \int_\Omega \mathrm{MMD}_k(\mu_c, \nu_c)^p \, \rho(dc) \right)^{1/p}.$$

*Proof.* By linearity of mean embeddings,

$$\mu_{\bar{\mu}} = \int_\Omega \mu_{\mu_c} \, \rho(dc), \qquad \mu_{\bar{\nu}} = \int_\Omega \mu_{\nu_c} \, \rho(dc),$$

where $\mu_{\mu_c} = \int_X \phi(x) \, d\mu_c(x)$ and $\mu_{\nu_c} = \int_X \phi(x) \, d\nu_c(x)$ are elements of $\mathcal{H}$. Thus,

$$\mu_{\bar{\mu}} - \mu_{\bar{\nu}} = \int_\Omega v(c) \, \rho(dc), \qquad v(c) := \mu_{\mu_c} - \mu_{\nu_c} \in \mathcal{H}.$$

Hence

$$\mathrm{MMD}_k(\bar{\mu}, \bar{\nu}) = \| \mu_{\bar{\mu}} - \mu_{\bar{\nu}} \|_\mathcal{H} = \left\| \int_\Omega v(c) \, \rho(dc) \right\|_\mathcal{H}$$

$$\leq \int_\Omega \| v(c) \|_\mathcal{H} \, \rho(dc) \qquad \text{(triangle inequality in } \mathcal{H})$$

$$\leq \left( \int_\Omega \| v(c) \|_\mathcal{H}^p \, \rho(dc) \right)^{1/p} \qquad (L^1 \leq L^p \text{ on a probability space}).$$

Finally, $\| v(c) \|_\mathcal{H} = \| \mu_{\mu_c} - \mu_{\nu_c} \|_\mathcal{H} = \mathrm{MMD}_k(\mu_c, \nu_c)$, which gives the claim. $\qquad \square$

**Lemma 12** (Mixture $p$–convexity for CPD kernels via the distance–induced RKHS). *Let $k : X \times X \to \mathbb{R}$ be conditionally positive definite (CPD) and let $k^\circ$ be the associated distance–induced (one–point centered) kernel from Proposition 3, so that for all probabilities $P, Q$ with finite integrals,*

$$\gamma_k(P, Q) = \mathrm{MMD}_{k^\circ}(P, Q).$$

*Let $(\Omega, \mathcal{F}, \rho)$ be a probability space, and let $(\mu_c)_{c \in \Omega}$ and $(\nu_c)_{c \in \Omega}$ be families of probability measures on $X$ with finite embeddings for $k^\circ$. Define the mixtures $\bar{\mu} := \int_\Omega \mu_c \, \rho(dc)$ and $\bar{\nu} := \int_\Omega \nu_c \, \rho(dc)$, and assume that $\bar{\mu}$ and $\bar{\nu}$ also have finite embeddings for $k^\circ$. Then for every $p \in [1, \infty)$,*

$$\gamma_k(\bar{\mu}, \bar{\nu}) \; \leq \; \left( \int_\Omega \gamma_k(\mu_c, \nu_c)^p \, \rho(dc) \right)^{1/p}.$$

*Proof.* By Proposition 3, $\gamma_k = \mathrm{MMD}_{k^\circ}$. Applying Lemma 11 to the PSD kernel $k^\circ$ and the families $(\mu_c), (\nu_c)$ yields

$$\mathrm{MMD}_{k^\circ}(\bar{\mu}, \bar{\nu}) \; \leq \; \left( \int_\Omega \mathrm{MMD}_{k^\circ}(\mu_c, \nu_c)^p \, \rho(dc) \right)^{1/p}.$$

Replacing $\mathrm{MMD}_{k^\circ}$ by $\gamma_k$ via Proposition 3 gives the claim. $\qquad \square$

**Lemma 13** (Max–sliced affine push-forward contraction for $\mathrm{MMD}_k$). *Let $k : \mathbb{R} \times \mathbb{R} \to \mathbb{R}$ be a kernel on $\mathbb{R}$, and let $\mathrm{MMD}_k$ be its associated maximum mean discrepancy on $\mathcal{P}(\mathbb{R})$. Assume that for all $\mu, \nu \in \mathcal{P}(\mathbb{R})$:*

(**T**) *Translation invariance: for every $t \in \mathbb{R}$,*

$$\mathrm{MMD}_k\big( (x \mapsto x + t)_\# \mu, \, (x \mapsto x + t)_\# \nu \big) = \mathrm{MMD}_k(\mu, \nu).$$

(**S**) *Scale–contraction: there exists a nondecreasing $c : [0, 1] \to [0, \infty)$ such that for every $s \in [0, 1]$,*

$$\mathrm{MMD}_k\big( (x \mapsto sx)_\# \mu, \, (x \mapsto sx)_\# \nu \big) \leq c(s) \, \mathrm{MMD}_k(\mu, \nu).$$

*Define the* max–sliced *lift of* $\mathrm{MMD}_k$ *by*

$$\mathbf{MS}\mathrm{MMD}_k(\mu,\nu) := \sup_{\theta\in\mathbb{S}^{d-1}} \mathrm{MMD}_k\big((P_\theta)_\#\mu,\ (P_\theta)_\#\nu\big), \qquad P_\theta(x) = \langle\theta, x\rangle.$$

*Let* $F(x) = Ax + b$ *with an arbitrary matrix* $A \in \mathbb{R}^{d\times d}$ *and* $b \in \mathbb{R}^d$, *and denote* $L := \|A\|_{\mathrm{op}} = \sup_{\|v\|=1}\|Av\|$. *If* $L \in [0,1]$, *then for all* $\mu,\nu \in \mathcal{P}(\mathbb{R}^d)$,

$$\boxed{\mathbf{MS}\mathrm{MMD}_k\big(F_\#\mu, F_\#\nu\big)\ \leq\ c(L)\,\mathbf{MS}\mathrm{MMD}_k(\mu,\nu).}$$

*Proof.* Fix $\theta \in \mathbb{S}^{d-1}$ and set $w_\theta := A^\top\theta$.

**Case 1:** $w_\theta = 0$. Then $(P_\theta \circ F)(x) = \langle\theta, b\rangle$ is constant, hence

$$\mathrm{MMD}_k\big((P_\theta)_\# F_\#\mu,\ (P_\theta)_\# F_\#\nu\big) = 0 \tag{34}$$
$$\leq c(0)\,\mathrm{MMD}_k\big((P_\phi)_\#\mu,\ (P_\phi)_\#\nu\big) \qquad \text{for any unit } \phi, \tag{35}$$

so the desired bound holds trivially.

**Case 2:** $\|w_\theta\| > 0$. Write $r_\theta := \|w_\theta\|$ and $\phi_\theta := w_\theta/r_\theta \in \mathbb{S}^{d-1}$. For any $X \sim \mu$ and $Y \sim \nu$,

$$(P_\theta \circ F)(X) = \langle\theta, AX + b\rangle = \langle\theta, b\rangle + r_\theta\langle\phi_\theta, X\rangle,$$

and similarly for $Y$. By **(T)** and **(S)**, we obtain

$$\mathrm{MMD}_k\big((P_\theta)_\# F_\#\mu,\ (P_\theta)_\# F_\#\nu\big) = \mathrm{MMD}_k\big(\mathrm{Law}(r_\theta\langle\phi_\theta, X\rangle),\ \mathrm{Law}(r_\theta\langle\phi_\theta, Y\rangle)\big) \tag{36}$$
$$\leq c(r_\theta)\,\mathrm{MMD}_k\big((P_{\phi_\theta})_\#\mu,\ (P_{\phi_\theta})_\#\nu\big) \tag{37}$$
$$\leq c(r_\theta) \sup_{\phi\in\mathbb{S}^{d-1}} \mathrm{MMD}_k\big((P_\phi)_\#\mu,\ (P_\phi)_\#\nu\big). \tag{38}$$

Since $r_\theta = \|A^\top\theta\| \leq \|A^\top\|_{\mathrm{op}} = \|A\|_{\mathrm{op}} = L$ and $L \in [0,1]$, we have $r_\theta \in [0,1]$ so that $c(r_\theta)$ is well defined.

**Taking the supremum.** Now take the supremum over $\theta \in \mathbb{S}^{d-1}$:

$$\sup_\theta \mathrm{MMD}_k\big((P_\theta)_\# F_\#\mu,\ (P_\theta)_\# F_\#\nu\big) \leq \sup_\theta c(r_\theta) \sup_\phi \mathrm{MMD}_k\big((P_\phi)_\#\mu,\ (P_\phi)_\#\nu\big). \tag{39}$$

Using $r_\theta \leq L$ and the fact that $c$ is nondecreasing on $[0,1]$, we obtain

$$\sup_\theta \mathrm{MMD}_k\big((P_\theta)_\# F_\#\mu,\ (P_\theta)_\# F_\#\nu\big) \leq c(L)\,\mathbf{MS}\mathrm{MMD}_k(\mu,\nu). \tag{40}$$

The left-hand side is exactly $\mathbf{MS}\mathrm{MMD}_k(F_\#\mu, F_\#\nu)$, which proves the claim. $\qquad\square$

**Lemma 14** (Max–sliced mixture $p$-convexity for $\mathrm{MMD}_k$). *Let* $k : \mathbb{R}\times\mathbb{R} \to \mathbb{R}$ *be a kernel on* $\mathbb{R}$, *and let* $\mathrm{MMD}_k$ *be its associated maximum mean discrepancy on* $\mathcal{P}(\mathbb{R})$. *Assume that* $\mathrm{MMD}_k$ *is mixture $p$-convex for some* $p \in [1,\infty)$: *for every probability space* $(\Omega, \mathcal{F}, \rho)$ *and families* $(\mu_c), (\nu_c) \subset \mathcal{P}(\mathbb{R})$ *for which* $\bar\mu := \int_\Omega \mu_c\,\rho(dc)$ *and* $\bar\nu := \int_\Omega \nu_c\,\rho(dc)$ *are well defined,*

$$\mathrm{MMD}_k\left(\int_\Omega \mu_c\,\rho(dc),\ \int_\Omega \nu_c\,\rho(dc)\right) \leq \left(\int_\Omega \mathrm{MMD}_k(\mu_c,\nu_c)^p\,\rho(dc)\right)^{1/p}.$$

*Define the max–sliced lift on* $\mathcal{P}(\mathbb{R}^d)$ *by*

$$\mathbf{MS}\mathrm{MMD}_k(\mu,\nu) := \sup_{\theta\in\mathbb{S}^{d-1}} \mathrm{MMD}_k\big((P_\theta)_\#\mu,\ (P_\theta)_\#\nu\big), \qquad P_\theta(x) = \langle\theta, x\rangle.$$

*Then* $\mathbf{MS}\mathrm{MMD}_k$ *is also mixture $p$-convex:*

$$\boxed{\mathbf{MS}\mathrm{MMD}_k\left(\int_\Omega \mu_c\,\rho(dc),\ \int_\Omega \nu_c\,\rho(dc)\right) \leq \left(\int_\Omega \mathbf{MS}\mathrm{MMD}_k(\mu_c,\nu_c)^p\,\rho(dc)\right)^{1/p}.}$$

*Proof.* Fix $\theta \in \mathbb{S}^{d-1}$ and set

$$\mu_c^\theta := (P_\theta)_\# \mu_c, \qquad \nu_c^\theta := (P_\theta)_\# \nu_c \quad \in \mathcal{P}(\mathbb{R}).$$

Pushforward commutes with mixtures:

$$(P_\theta)_\# \Big( \int \mu_c \, d\rho \Big) = \int \mu_c^\theta \, d\rho, \qquad (P_\theta)_\# \Big( \int \nu_c \, d\rho \Big) = \int \nu_c^\theta \, d\rho.$$

By mixture $p$-convexity of $\mathrm{MMD}_k$ on $\mathbb{R}$,

$$\mathrm{MMD}_k \big( (P_\theta)_\# \textstyle\int \mu_c \, d\rho, \ (P_\theta)_\# \int \nu_c \, d\rho \big) \ \leq \ \Big( \int \mathrm{MMD}_k(\mu_c^\theta, \nu_c^\theta)^p \, d\rho \Big)^{1/p}. \tag{41}$$

Taking the supremum over $\theta$ on the left-hand side of Equation 41 gives

$$\sup_\theta \ \mathrm{MMD}_k \big( (P_\theta)_\# \textstyle\int \mu_c \, d\rho, \ (P_\theta)_\# \int \nu_c \, d\rho \big) \ \leq \ \sup_\theta \ \Big( \int \mathrm{MMD}_k(\mu_c^\theta, \nu_c^\theta)^p \, d\rho \Big)^{1/p}. \tag{42}$$

Define $f(\theta, c) := \mathrm{MMD}_k(\mu_c^\theta, \nu_c^\theta)$ and $h(c) := \sup_\phi f(\phi, c) = \mathbf{MS}\mathrm{MMD}_k(\mu_c, \nu_c)$. Since $f(\theta, c) \leq h(c)$ pointwise in $c$, we obtain for every $\theta$,

$$\Big( \int f(\theta, c)^p \, d\rho(c) \Big)^{1/p} \ \leq \ \Big( \int h(c)^p \, d\rho(c) \Big)^{1/p}.$$

Taking $\sup_\theta$ yields

$$\sup_\theta \ \Big( \int \mathrm{MMD}_k(\mu_c^\theta, \nu_c^\theta)^p \, d\rho \Big)^{1/p} \ \leq \ \Big( \int \mathbf{MS}\mathrm{MMD}_k(\mu_c, \nu_c)^p \, d\rho \Big)^{1/p}. \tag{43}$$

Combining Equation 42 and Equation 43 shows

$$\mathbf{MS}\mathrm{MMD}_k \Big( \int \mu_c \, d\rho, \ \int \nu_c \, d\rho \Big) \ \leq \ \Big( \int \mathbf{MS}\mathrm{MMD}_k(\mu_c, \nu_c)^p \, \rho(dc) \Big)^{1/p},$$

as claimed. $\qquad\square$

**Theorem 5** (Supremum–max–sliced $\mathrm{MMD}_k$ contraction of the **complete** Sobolev Bellman operator)**.** *Let*

$$T_\pi^{S_{s,a}} \colon \big( \mathcal{S} \times \mathcal{A} \to \mathcal{P}(\mathbb{R}^{1+m+n}) \big) \ \longrightarrow \ \big( \mathcal{S} \times \mathcal{A} \to \mathcal{P}(\mathbb{R}^{1+m+n}) \big),$$

*be the complete Sobolev Bellman operator bootstrapping the vector* $(Z, \partial_a Z, \partial_s Z)$.

*Let* $k : \mathbb{R} \times \mathbb{R} \to \mathbb{R}$ *be a kernel on* $\mathbb{R}$ *and let* $\mathrm{MMD}_k$ *be its associated maximum mean discrepancy on* $\mathcal{P}(\mathbb{R})$. *Define the max–sliced lift on* $\mathcal{P}(\mathbb{R}^{1+m+n})$ *by*

$$\mathbf{MS}\mathrm{MMD}_k(\mu, \nu) := \sup_{\theta \in \mathbb{S}^{m+n}} \mathrm{MMD}_k \big( (P_\theta)_\# \mu, \ (P_\theta)_\# \nu \big), \qquad P_\theta(x) = \langle \theta, x \rangle,$$

*and its supremum version over state–action pairs by*

$$\overline{\mathbf{MS}\mathrm{MMD}}_k(\eta_1, \eta_2) := \sup_{(s,a)} \mathbf{MS}\mathrm{MMD}_k \big( \eta_1(s, a), \eta_2(s, a) \big).$$

*Assume* $\mathrm{MMD}_k$ *satisfies:*

(T) ***Translation invariance:*** *for all* $t \in \mathbb{R}$ *and all* $\mu, \nu \in \mathcal{P}(\mathbb{R})$,

$$\mathrm{MMD}_k \big( (x \mapsto x + t)_\# \mu, \ (x \mapsto x + t)_\# \nu \big) = \mathrm{MMD}_k(\mu, \nu).$$

(S) ***Scale–contraction:*** *there exists a nondecreasing* $c : [0, 1] \to [0, \infty)$ *such that for all* $s \in [0, 1]$ *and all* $\mu, \nu \in \mathcal{P}(\mathbb{R})$,

$$\mathrm{MMD}_k \big( (x \mapsto sx)_\# \mu, \ (x \mapsto sx)_\# \nu \big) \leq c(s) \, \mathrm{MMD}_k(\mu, \nu).$$

**(M$_p$)** *Mixture $p$-convexity: for some $p \in [1, \infty)$, for every probability space $(\Omega, \mathcal{F}, \rho)$ and families $(\mu_c), (\nu_c) \subset \mathcal{P}(\mathbb{R})$ for which the mixtures are well defined,*

$$\mathrm{MMD}_k\left(\int_\Omega \mu_c\, \rho(dc),\ \int_\Omega \nu_c\, \rho(dc)\right) \leq \left(\int_\Omega \mathrm{MMD}_k(\mu_c, \nu_c)^p\, \rho(dc)\right)^{1/p}.$$

*The complete Sobolev Bellman operator is defined by the same reparameterized ingredients as in Theorem 3, namely the policy $\pi(s'; \varepsilon_\pi)$, the transition $f(s, a; \varepsilon_f)$, and the reward $r(s, a; \varepsilon_r)$. In particular, for each $(s, a)$ and noise draw $(\varepsilon_f, \varepsilon_\pi)$ the complete Sobolev update acts as an affine push-forward with linear part $\mathcal{L}(s, a; \varepsilon_f, \varepsilon_\pi)$ (as in Theorem 3). Define the uniform bound*

$$\|\mathcal{L}\|_d = \sup_{s,a}\ \sup_{\varepsilon_f, \varepsilon_\pi} \big\|\mathcal{L}(s, a; \varepsilon_f, \varepsilon_\pi)\big\|_{\mathrm{op}} \ \leq\ \gamma\, \kappa_{\mathrm{full}}.$$

*Assume $\|\mathcal{L}\|_d \in [0, 1]$. Then for any two Sobolev return–distribution functions $\eta_1, \eta_2$,*

$$\boxed{\ \overline{\mathbf{MS}\mathrm{MMD}_k}\big(T_\pi^{S_{s,a}} \eta_1,\ T_\pi^{S_{s,a}} \eta_2\big) \ \leq\ c(\|\mathcal{L}\|_d)\ \overline{\mathbf{MS}\mathrm{MMD}_k}(\eta_1, \eta_2).\ }$$

*In particular, $T_\pi^{S_{s,a}}$ is a strict contraction whenever $c(\|\mathcal{L}\|_d) < 1$.*

*Proof.* Fix $(s, a)$ and condition on $C = (\varepsilon_r, \varepsilon_f, \varepsilon_\pi)$. Set

$$r = r(s, a; \varepsilon_r), \qquad s' = f(s, a; \varepsilon_f), \qquad a' = \pi(s'; \varepsilon_\pi),$$

and let $X_i' \mid C \sim \eta_i(s', a')$. By the push-forward identity (Lemma 1) and the affine update in Equation 22,

$$\big(T_\pi^{S_{s,a}} \eta_i\big)(s, a) = \mathrm{Law}\big(\Phi_{s,a}(X_i')\big).$$

**Affine push-forward at fixed $C$.** Let $L(C) := \|\mathcal{L}(s, a; \varepsilon_f, \varepsilon_\pi)\|_{\mathrm{op}}$. Applying Lemma 13 (for $\mathrm{MMD}_k$), which relies on **(T)** and **(S)**, to the conditional laws of $X_i'$ gives

$$\begin{aligned}
&\mathbf{MS}\mathrm{MMD}_k\big(\mathrm{Law}(\Phi_{s,a}(X_1') \mid C),\ \mathrm{Law}(\Phi_{s,a}(X_2') \mid C)\big) \\
&\leq c\big(L(C)\big)\, \mathbf{MS}\mathrm{MMD}_k\big(\mathrm{Law}(X_1' \mid C),\ \mathrm{Law}(X_2' \mid C)\big) \\
&= c\big(L(C)\big)\, \mathbf{MS}\mathrm{MMD}_k\big(\eta_1(s', a'),\ \eta_2(s', a')\big).
\end{aligned} \tag{44}$$

**Averaging over $C$.** Lemma 14 (for $\mathrm{MMD}_k$), which relies on **(M$_p$)**, together with Equation 44 yields

$$\mathbf{MS}\mathrm{MMD}_k\big(\mathrm{Law}(\Phi_{s,a}(X_1')),\ \mathrm{Law}(\Phi_{s,a}(X_2'))\big)$$

$$\leq \left(\int \mathbf{MS}\mathrm{MMD}_k\big(\mathrm{Law}(\Phi_{s,a}(X_1') \mid C), \mathrm{Law}(\Phi_{s,a}(X_2') \mid C)\big)^p\, \rho(dC)\right)^{1/p} \tag{45}$$

$$\leq \left(\int \big(c(L(C))\, \mathbf{MS}\mathrm{MMD}_k\big(\eta_1(s', a'), \eta_2(s', a')\big)\big)^p\, \rho(dC)\right)^{1/p} \tag{46}$$

$$\leq c(\|\mathcal{L}\|_d) \left(\int \mathbf{MS}\mathrm{MMD}_k\big(\eta_1(s', a'), \eta_2(s', a')\big)^p\, \rho(dC)\right)^{1/p},$$

since $c$ is nondecreasing on $[0, 1]$ and $L(C) \leq \|\mathcal{L}\|_d \leq 1$ for all $C$.

**Supremum bound.** For any realization of $C$, by definition of the supremum metric,

$$\mathbf{MS}\mathrm{MMD}_k\big(\eta_1(s', a'), \eta_2(s', a')\big) \leq \sup_{(u,v)} \mathbf{MS}\mathrm{MMD}_k\big(\eta_1(u, v), \eta_2(u, v)\big)$$

$$= \overline{\mathbf{MS}\mathrm{MMD}_k}(\eta_1, \eta_2).$$

Combining this with the previous inequality,

$$\begin{aligned}
\mathbf{MS}\mathrm{MMD}_k\big(&(T_\pi^{S_{s,a}} \eta_1)(s, a),\ (T_\pi^{S_{s,a}} \eta_2)(s, a)\big) \\
&= \mathbf{MS}\mathrm{MMD}_k\big(\mathrm{Law}(\Phi_{s,a}(X_1')),\ \mathrm{Law}(\Phi_{s,a}(X_2'))\big) \\
&\leq c(\|\mathcal{L}\|_d) \left(\int \overline{\mathbf{MS}\mathrm{MMD}_k}(\eta_1, \eta_2)^p\, \rho(dC)\right)^{1/p} \\
&= c(\|\mathcal{L}\|_d)\, \overline{\mathbf{MS}\mathrm{MMD}_k}(\eta_1, \eta_2).
\end{aligned}$$

since $\overline{\mathbf{MS}}\mathrm{MMD}_k(\eta_1, \eta_2)$ does not depend on $C$. Taking the supremum over $(s, a)$ completes the proof:

$$\overline{\mathbf{MS}}\mathrm{MMD}_k(T_\pi^{S_s,a}\eta_1, T_\pi^{S_s,a}\eta_2) \ \leq \ c(\|\mathcal{L}\|_d) \, \overline{\mathbf{MS}}\mathrm{MMD}_k(\eta_1, \eta_2).$$

$\square$

**Corollary 5.1** (Supremum–max–sliced contraction for MMD with the multiquadric kernel). Let $k_h(x, y) = -\sqrt{1 + h^2\|x - y\|^2}$ with $h > 0$ (the multiquadric kernel). Consider the setting of Theorem 5, and assume in addition that the bound $\|\mathcal{L}\|_d$ satisfies $\|\mathcal{L}\|_d \in [0, 1]$. Then, for all return–distribution functions $\eta_1, \eta_2$,

$$\boxed{\overline{\mathbf{MS}}\mathrm{MMD}_{k_h}\big(T_\pi^{S_s,a}\eta_1, \ T_\pi^{S_s,a}\eta_2\big) \ \leq \ \sqrt{\|\mathcal{L}\|_d} \, \overline{\mathbf{MS}}\mathrm{MMD}_{k_h}(\eta_1, \ \eta_2).}$$

*Proof.* We verify the conditions of Theorem 5 for the MMD associated with the multiquadric kernel.

- **(T) (Translation invariance):** Since $k_h$ is radial, it depends only on the distance $\|x - y\|$, implying that the MMD induced by $k_h$ is invariant under simultaneous translation of its arguments.

- **(S) (Scale–contraction):** By Lemma 10 (specifically Equation 28), for any $s \in [0, 1]$, the unsquared MMD under $k_h$ satisfies scale–contraction with the specific function $c(s) = \sqrt{s}$.

- **($\mathbf{M}_p$) (Mixture $p$–convexity):** The kernel $k_h$ is conditionally positive definite (CPD) (see e.g., Theorem 3.1 of Killingberg & Langseth (2023)). Therefore, by Lemma 12, the MMD associated with $k_h$ satisfies mixture $p$–convexity.

Applying Theorem 5 with $c(s) = \sqrt{s}$ yields the contraction factor $c(\|\mathcal{L}\|_d) = \sqrt{\|\mathcal{L}\|_d}$. $\square$

**Theorem 6** (Supremum–max–sliced MMD contraction of the **incomplete** Sobolev Bellman operator). *Let $T_\pi^{S_a} : (\mathcal{S} \times \mathcal{A} \to \mathcal{P}(\mathbb{R}^{1+m})) \to (\mathcal{S} \times \mathcal{A} \to \mathcal{P}(\mathbb{R}^{1+m}))$ be the action–gradient (incomplete) Sobolev Bellman operator that updates $H = (Z, \partial_a Z)$ as in Equation 18.*

*Assume:*

- **(T)** *Translation invariance:* $\mathrm{MMD}_k((x \mapsto x + t)_{\#}\mu, (x \mapsto x + t)_{\#}\nu) = \mathrm{MMD}_k(\mu, \nu)$ *for all $t \in \mathbb{R}$ and all $\mu, \nu \in \mathcal{P}(\mathbb{R})$.*

- **(S)** *Scale–contraction:* *There exists a nondecreasing $c_k : [0, 1] \to [0, \infty)$ such that*

$$\mathrm{MMD}_k((x \mapsto sx)_{\#}\mu, (x \mapsto sx)_{\#}\nu) \leq c_k(s)\,\mathrm{MMD}_k(\mu, \nu), \quad \forall s \in [0, 1], \ \mu, \nu \in \mathcal{P}(\mathbb{R}).$$

- **($\mathbf{M}_p$)** *Mixture 1–convexity:* *For every probability space $(\Omega, \mathcal{F}, \rho)$ and families $\mu_c, \nu_c \in \mathcal{P}(\mathbb{R})$ for which $\bar{\mu} := \int_\Omega \mu_c\, \rho(dc)$ and $\bar{\nu} := \int_\Omega \nu_c\, \rho(dc)$ are well defined,*

$$\mathrm{MMD}_k\left(\int_\Omega \mu_c\, \rho(dc), \int_\Omega \nu_c\, \rho(dc)\right) \leq \int_\Omega \mathrm{MMD}_k(\mu_c, \nu_c)\, \rho(dc),$$

*and the corresponding max–sliced version (Lemma 14) holds.*

*Assume*

$$\|f_a\|_2 \leq L_{f,a}, \qquad \|\pi_s\|_2 \leq L_\pi.$$

*For the linear blocks in Equation 18,*

$$A(s, a; \varepsilon_f, \varepsilon_\pi) = \gamma \begin{pmatrix} 1 & 0 \\ 0 & f_a^\top \pi_s^\top \end{pmatrix}, \qquad N(s, a; \varepsilon_f) = \gamma \begin{pmatrix} 0 \\ f_a^\top \end{pmatrix}.$$

*Define*

$$\bar{L} := \sup_{s, a, \varepsilon_f, \varepsilon_\pi} \|[\,A \ N\,]\|, \qquad \text{and assume } \bar{L} \leq 1.$$

*For each $(s', a')$, assume there exists a lifting map*

$$\mathcal{J}_{s',a'} : \mathcal{P}(\mathbb{R}^{1+m}) \longrightarrow \mathcal{P}(\mathbb{R}^{1+m+n}),$$

*with first marginal equal to the input law and joint law induced by our distribution parametrization (Appendix B.2), so that the second marginal matches the true law of $\partial_s Z'$. Assume that for some $L_{\mathrm{aug}} \geq 1$,*

$$\mathbf{MS}\mathrm{MMD}_k\big(\mathcal{J}_{s',a'}(\mu),\ \mathcal{J}_{s',a'}(\nu)\big) \ \leq\ L_{\mathrm{aug}}\ \mathbf{MS}\mathrm{MMD}_k(\mu,\nu), \quad \forall\,\mu,\nu.$$

*Define $\overline{\mathbf{MS}\mathrm{MMD}}_k(\eta_1,\eta_2) = \sup_{(s,a)} \mathbf{MS}\mathrm{MMD}_k(\eta_1(s,a),\eta_2(s,a))$. Then, for all return–gradient maps $\eta_1,\eta_2$,*

$$\boxed{\ \overline{\mathbf{MS}\mathrm{MMD}}_k\big(T_\pi^{S_a}\eta_1,\ T_\pi^{S_a}\eta_2\big)\ \leq\ L_{\mathrm{aug}}\ c_k(\bar{L})\ \overline{\mathbf{MS}\mathrm{MMD}}_k(\eta_1,\ \eta_2).\ }$$

*In particular, if $L_{\mathrm{aug}}\, c_k(\bar{L}) < 1$, then $T_\pi^{S_a}$ is a strict contraction.*

*For the multiquadric kernel $k_h(x,y) = -\sqrt{1+h^2\|x-y\|^2}$, Lemma 10 gives $c_k(s) = \sqrt{s}$ for $s \in [0,1]$. Hence, whenever $\bar{L} \leq 1$, a sufficient condition for strict contraction is $L_{\mathrm{aug}}\sqrt{\bar{L}} < 1$.*

*Proof.* Fix $(s,a)$ and a noise draw $C = (\varepsilon_r, \varepsilon_f, \varepsilon_\pi)$. Set

$$s' = f(s,a;\varepsilon_f), \qquad a' = \pi(s';\varepsilon_\pi).$$

We will rewrite the incomplete update as an affine map on an augmented next-step variable that includes the missing state-gradient component.

**Augmented variable.** Let $\tilde{X}_i' = (H_i',\, U_i') \in \mathbb{R}^{(1+m)+n}$ with law $\tilde{\eta}_i = \mathcal{J}_{s',a'}(\eta_i(s',a'))$. Define the affine map on the augmented space

$$\Phi_{s,a}(\tilde{x};C) \ := \ b\ +\ [\,A\ \ N\,]\tilde{x}.$$

Then, conditionally on $C$, we have

$$\langle \theta,\, \Phi_{s,a}(\tilde{X}_i';C)\rangle = \langle\theta,b\rangle + \big\langle[\,A\ \ N\,]^\top\theta,\ \tilde{X}_i'\big\rangle.$$

Moreover, by the same argument as in Lemma 2,

$$(T_\pi^{S_a}\eta_i)(s,a) = \mathrm{Law}\big(\Phi_{s,a}(\tilde{X}_i';C)\big).$$

**Apply max–sliced affine contraction lemma.** Since $\bar{L} \leq 1$, the operator norm of the joint matrix $\|[\,A\ \ N\,]\|$ is in $[0,1]$ for all $C$. We may thus apply Lemma 13 (which relies on **(T)** and **(S)**):

$$\mathbf{MS}\mathrm{MMD}_k\Big(\mathrm{Law}(\Phi_{s,a}(\tilde{X}_1';C)\mid C),\ \mathrm{Law}(\Phi_{s,a}(\tilde{X}_2';C)\mid C)\Big)$$
$$\leq c_k(\|[\,A\ \ N\,]\|)\ \mathbf{MS}\mathrm{MMD}_k(\tilde{\eta}_1,\tilde{\eta}_2).$$

**Lift back to** $(Z, \partial_a Z)$**.** By the lifting assumption,

$$\mathbf{MS}\mathrm{MMD}_k(\tilde{\eta}_1,\tilde{\eta}_2)\ \leq\ L_{\mathrm{aug}}\ \mathbf{MS}\mathrm{MMD}_k(\eta_1(s',a'),\,\eta_2(s',a')).$$

**Average over noise.** By Lemma 14 (mixture 1–convexity), we have

$$\mathbf{MS}\mathrm{MMD}_k\big((T_\pi^{S_a}\eta_1)(s,a),(T_\pi^{S_a}\eta_2)(s,a)\big)$$
$$\leq \int \mathbf{MS}\mathrm{MMD}_k\big(\mathrm{Law}(\Phi_{s,a}(\tilde{X}_1';C)\mid C),\mathrm{Law}(\Phi_{s,a}(\tilde{X}_2';C)\mid C)\big)\,\rho(dC).$$

Combining the affine and lifting bounds derived above, the integrand is bounded pointwise by

$$L_{\mathrm{aug}}\, c_k(\|[\,A\ \ N\,]\|)\, \mathbf{MS}\mathrm{MMD}_k\big(\eta_1(s',a'),\eta_2(s',a')\big).$$

By definition of the supremum metric, $\mathbf{MS}\mathrm{MMD}_k(\eta_1(u,v),\eta_2(u,v)) \leq \overline{\mathbf{MS}\mathrm{MMD}}_k(\eta_1,\eta_2)$ for any state-action pair $(u,v)$. Using this global bound and $\|[\,A\ \ N\,]\| \leq \bar{L}$, the integral satisfies

$$\mathbf{MS}\mathrm{MMD}_k\big((T_\pi^{S_a}\eta_1)(s,a),\ (T_\pi^{S_a}\eta_2)(s,a)\big) \leq L_{\mathrm{aug}}\, c_k(\bar{L})\, \overline{\mathbf{MS}\mathrm{MMD}}_k(\eta_1,\eta_2).$$

Taking the supremum over $(s,a)$ on the left-hand side yields the claim. □

## H  BACKGROUND ON THE WORLD-MODEL

### H.1  CONDITIONAL VARIATIONAL AUTO-ENCODERS

A principled invertible generative model can be obtained from a Variational Auto-Encoder (VAE) (Kingma, 2013). More interestingly for us are conditional VAE Sohn et al. (2015) which we briefly introduce.

A Conditional Variational Autoencoder (cVAE) is a generative model that learns to generate new samples from a distribution conditioned on given input information. In our case, the cVAE models the distribution of next states and rewards conditioned on current states and actions.

Formally, the cVAE consists of two components:

- **Encoder**: The encoder $q_\zeta(\varepsilon \mid s', r; s, a)$ maps the observed next state $s'$ and reward $r$, conditioned on the current state-action pair $(s, a)$, to a latent variable $\varepsilon$, typically modeled as a Gaussian distribution with diagonal covariance matrix:

$$q_\zeta(\varepsilon \mid s', r; s, a) = \mathcal{N}(\varepsilon; \mu_\zeta(s', r, s, a), \sigma_\zeta^2(s', r, s, a) \odot I). \tag{47}$$

- **Decoder**: The decoder $p_\psi(s', r \mid \varepsilon; s, a)$ reconstructs the next state $s'$ and reward $r$ from the latent variable $\varepsilon$, conditioned on the current state-action pair $(s, a)$.

- **Prior**: The prior $p_\upsilon(s', r \mid \varepsilon; s, a)$ allows more flexibility in the latent space than a simple standard Gaussian $\mathcal{N}(0; I)$ It is also parametrized as a multivariate diagonal Gaussian:

$$p_\upsilon(\varepsilon \mid s, a) = \mathcal{N}(\varepsilon; \mu_\upsilon(s, a), \sigma_\upsilon^2(s, a) \odot I). \tag{48}$$

The objective of a cVAE is to maximize the Evidence Lower Bound (ELBO), which balances accurate reconstruction of the input with a regularization term that ensures the learned posterior distribution remains close to the prior distribution. The objective is as follows

$$\begin{aligned}\mathcal{L}_{\text{cVAE}}(\zeta, \psi) = &\ \mathbb{E}_{q_\zeta(\varepsilon \mid s', r; s, a)} \left[ \log p_\psi(s', r \mid \varepsilon; s, a) \right] \\ &- \lambda_{\text{KL}} \times D_{\text{KL}} \left( q_\zeta(\varepsilon \mid s', r; s, a) \parallel p_\upsilon(\varepsilon \mid s, a) \right).\end{aligned} \tag{49}$$

The first term encourages faithful reconstruction of the next state and reward, while the second term regularizes the posterior distribution to remain close to a standard Gaussian prior. Assuming the decoder $p_\psi(s', r \mid \varepsilon; s, a)$ is Gaussian with a fixed variance, the reconstruction term reduces to an L2 loss, which can be estimated using the difference between the reconstructed samples and the true samples.

Assuming the encoder parametrizes a Gaussian with diagonal covariance and that the prior is also Gaussian with identity covariance and zero mean, the KL divergence can be estimated from encoded input samples as

$$D_{\text{KL}}(\zeta) = \mathbb{E}_{(s,a)} \left[ \frac{1}{2} \sum_{j=1}^{d} \left( 1 + \log(\sigma_{\zeta,j}^2(s, a)) - \mu_{\zeta,j}^2(s, a) - \sigma_{\zeta,j}^2(s, a) \right) \right]. \tag{50}$$

### H.2  CONDITIONAL NORMALIZING FLOWS AS WORLD MODELS

Normalizing flows provide an alternative class of generative models based on a sequence of invertible transformations with tractable Jacobians. Let $f_\phi : \mathbb{R}^d \to \mathbb{R}^d$ be an invertible map, and let $z \sim p_Z$ denote a simple base distribution (e.g. standard Gaussian). A flow represents a random variable $x$ through the transformation $x = f_\phi^{-1}(z)$. Its density follows from the change-of-variables formula:

$$p_X(x) = p_Z(f_\phi(x)) \left| \det \nabla_x f_\phi(x) \right|. \tag{51}$$

By designing each transformation so that its Jacobian determinant is efficient to compute, normalizing flows permit exact likelihood evaluation and fast sampling.

**RealNVP.** RealNVP (Dinh et al., 2016) is a widely used normalizing-flow architecture based on *affine coupling layers*. Each coupling layer partitions its input $(x_1, x_2)$, leaves $x_1$ unchanged, and transforms $x_2$ as

$$y_1 = x_1, \qquad y_2 = x_2 \odot \exp\big(s_\phi(x_1)\big) + t_\phi(x_1),$$

where $s_\phi$ and $t_\phi$ are neural networks. Because the transformation of $x_2$ depends only on $x_1$, the Jacobian is triangular, and the log-determinant reduces to a sum over the scaling terms. Stacking multiple coupling layers with alternating partitions yields an expressive model with computationally cheap density evaluation and sampling.

**Conditional flows for one-step dynamics.** To model one-step transitions, the flow is conditioned on the current state–action pair $(s, a)$. The coupling-layer networks $s_\phi(\cdot)$ and $t_\phi(\cdot)$ therefore take $(s, a)$ as an additional input, yielding a conditional distribution

$$p_\phi(s', r \mid s, a).$$

Sampling proceeds by drawing $z \sim p_Z$ and applying the inverse transformation

$$(s', r) = f_\phi^{-1}(z; s, a),$$

which provides a differentiable mapping from $(s, a, z)$ to next-state and reward samples. As with the cVAE model introduced above, this enables efficient sampling together with the pathwise derivatives $\partial(s', r)/\partial(s, a)$ required for Sobolev temporal differences.

# I ALGORITHM DESIGN, PSEUDO-CODE AND BASELINE

In this section we describe the motivation for some design choices of our method. Section I.1 sets the design for the toy reinforcement learning experiment of Section 6.1 and the MuJoCo experiments from Section 6.2. We follow by providing the pseudo-code in Section I.2. Finally in Section I.3 we discuss our implementation of our primary baseline MAGE D'Oro & Jaskowski (2020).

## I.1 DESIGN CHOICES

Firstly, most experiments are conducted in the *data-efficient* setting, where the number of updates per interaction with the environment (UTD ratio) is larger than one, making stability and overestimation bias critical concerns. MAGE adds gradient regularization on top of TD3 (Fujimoto et al., 2018), which itself incorporates several modifications to DDPG (Lillicrap et al., 2016). Below, we describe each modification and whether it was retained in our implementation.

**Target policy smoothing:** this technique applies random noise to the policy's action when estimating the TD learning target. It is the only TD3 (and MAGE) modification we retained, as other changes cluttered the implementation and introduced noise that could interact poorly with distributional modeling. It is worth noting that omitting target policy smoothing in similar settings is not uncommon (Singh et al., 2022; Kuznetsov et al., 2020).

**Delayed policy update:** the policy is updated once for every two critic updates, which helps stabilize learning by preventing premature policy shifts.

**Double estimation:** an ensemble of two critics, $Q_1, Q_2$ or $Z_1, Z_2$, is used, with the bootstrapped target taken as the minimum of the two estimates at the next state (including policy smoothing noise). This modification introduces an underestimation bias to counteract the well-known overestimation issue in value-based methods (Van Hasselt et al., 2016). We observed double estimation to be critical for stable performance in the data-efficient setting: without it, average Q-values quickly diverged. Overestimation bias can also be addressed in the distributional setting, as demonstrated by TQC (Kuznetsov et al., 2020). A straightforward approach to induce underestimation is to truncate the top $p\%$ of values in the critic's target distribution; in our setup, truncating as few as 25% of values proved highly effective. Following TQC (Chen et al., 2021), we employ an ensemble of two distributional critics whose samples are concatenated.

## I.2 PSEUDO-CODES

In this subsection, we present the pseudo-code for the MSMMD estimation procedure, the MMD estimation procedure, and the full algorithm for policy evaluation and improvement.

- Algorithm 1 (Estimation of MSMMD): approximates the supremum over projection directions by gradient ascent on the unit sphere, yielding an empirical max–sliced MMD between two sets of samples.
- Algorithm 2 (MMD Estimation of Sobolev samples): leverages the world model to generate differentiable Sobolev-return samples, and applies truncation to mitigate overestimation bias as in Kuznetsov et al. (2020).
- Algorithm 3 (Full DSDPG algorithm): integrates our Sobolev-MMD components into the DDPG framework Lillicrap et al. (2016).

---

**Algorithm 1** Estimation of MSMMD from empirical samples

---

1: **Require:** Empirical samples $X = \{x_i\}_{i=1}^N \subset \mathbb{R}^d$, $Y = \{y_i\}_{i=1}^N \subset \mathbb{R}^d$
2: **Require:** Kernel $k$ defining the MMD
3: **Require:** Number of gradient steps $T$, step size $\eta$
4: **Output:** $\widehat{\mathrm{MSMMD}}(X, Y)$
5: Draw $w \sim \mathcal{N}(0, I_d)$                                {random unit direction}
6: $\theta \leftarrow w/\|w\|$                                     {random unit direction}
7: **for** $t = 1, \ldots, T$ **do**
8:     **for** $i = 1, \ldots, N$ **do**
9:         $u_i \leftarrow \langle \theta, x_i \rangle$                           {project to 1D along $\theta$}
10:         $v_i \leftarrow \langle \theta, y_i \rangle$                           {project to 1D along $\theta$}
11:     **end for**
12:     $J(\theta) \leftarrow \mathrm{MMD}_k\big(\{u_i\}_{i=1}^N, \{v_i\}_{i=1}^N\big)$         {maximize MMD over $\theta$}
13:     $g \leftarrow \nabla_\theta J(\theta)$                         {gradient w.r.t. direction}
14:     $w \leftarrow w + \eta\, g$             {ascent step in unconstrained space}
15:     $\theta \leftarrow w/\|w\|$           {re-normalize onto the unit sphere}
16: **end for**
17: $\bar{\theta} \leftarrow \mathrm{stop\_grad}(\theta)$          {stop gradient on the final direction}
18: **Return:** $\widehat{\mathrm{MSMMD}}(X, Y) \leftarrow \mathrm{MMD}_k\big(\{\langle \bar{\theta}, x_i \rangle\}_{i=1}^N, \{\langle \bar{\theta}, y_i \rangle\}_{i=1}^N\big)$

---

---

**Algorithm 2** Estimation of $\text{MMD}^2$ loss via imagination of transition samples with Sobolev samples

---

1: **Require:** Number of samples $M$, kernel $k$, discount factor $\gamma \in (0, 1)$
2: **Require:** Truncation percentage $p \in [0, 100]$
3: **Require:** Distributional critic $Z_\phi(s, a, \varepsilon)$
4: **Require:** Policy network $\pi_\theta(s)$
5: **Require:** Conditional VAE (cVAE) with prior $p_\upsilon(\varepsilon)$ and decoder $p_\psi(s', r \mid s, a, \varepsilon)$
6: **Input:** Transition sample $(s, a)$
7: **Input:** Online critic parameter $\phi$, target critic parameters $\phi_1'$ and $\phi_2'$
8: **Input:** Target policy parameter $\theta'$
9: **Input:** Boolean flag `use_action_gradient`
10: **Output:** Gradient estimation of MMD with respect to $\phi$
11: Draw $\varepsilon \sim p_\upsilon(\varepsilon)$        {Sample latent variable from the prior}
12: $(\hat{s}', \hat{r}) \sim p_\psi(s', r \mid s, a, \varepsilon)$        {Generate transition using the decoder}
13: $a' \leftarrow \pi_{\theta'}(\hat{s}')$        {Action from target policy on $\hat{s}'$}
14: Sample $Z_{1:M} \overset{i.i.d.}{\sim} Z_\phi(s, a)$        {Samples from online critic}
15: Sample $Z_{\text{next},1:M}^{(1)} \overset{i.i.d.}{\sim} Z_{\phi_1'}(\hat{s}', a')$,
16: Sample $Z_{\text{next},1:M}^{(2)} \overset{i.i.d.}{\sim} Z_{\phi_2'}(\hat{s}', a')$        {Samples from each target critic}
17: $Z_{\text{next},1:2M} \leftarrow \text{concat}(Z_{\text{next}}^{(1)}, Z_{\text{next}}^{(2)})$        {Concatenate target-critic samples}
18: **if** `use_action_gradient` **then**
19:      **for** each $1 \leq i \leq 2M$ **do**
20:          $Y_i \leftarrow \mathbf{f}_{\hat{r}, \hat{s}', \gamma}^S(Z_{\text{next},i})$        {Bellman target Sobolev samples}
21:          $\nabla_a Y_i \leftarrow \nabla_a \mathbf{f}_{\hat{r}, \hat{s}', \gamma}^S(Z_{\text{next},i})$        {Action gradient of Bellman target}
22:          $Y_i \leftarrow \text{concat}(Y_i, \nabla_a Y_i)$
23:      **end for**
24:      **for** each $1 \leq i \leq M$ **do**
25:          $\nabla_a Z_i \leftarrow \nabla_a Z_\phi(s, a)$        {Action gradient of online critic}
26:          $Z_i \leftarrow \text{concat}(Z_i, \nabla_a Z_i)$
27:      **end for**
28: **else**
29:      **for** each $1 \leq i \leq 2M$ **do**
30:          $Y_i \leftarrow \mathbf{f}_{\hat{r}, \hat{s}', \gamma}^S(Z_{\text{next},i})$
31:      **end for**
32: **end if**
33: Prune top $p\%$ of $\{Y_i\}_{i=1}^{2M}$ by return magnitude        {Low-bias target set}
34: Compute $\text{MMD}^2$:
35: $\text{MMD}^2 \leftarrow \sum_{i \neq j} \left[ k(Z_i, Z_j) - 2k(Z_i, Y_j) + k(Y_i, Y_j) \right]$
36: **Return:** $\text{MMD}^2$

---

---

**Algorithm 3** Distributional Sobolev Deterministic Policy Gradient (DSDPG)

---

1: **Require:** Number of samples $M$, number of policy samples $M_{\text{policy}}$, kernel $k$, discount factor $\gamma \in (0, 1)$, learning rates $\alpha_\theta, \alpha_\phi$
2: **Require:** Two distributional critics $Z_{\phi_1}(s, a)$ and $Z_{\phi_2}(s, a)$ with targets $Z_{\phi'_1}(s, a), Z_{\phi'_2}(s, a)$
3: **Require:** Actor network $\pi_\theta(s)$ with target $\pi_{\theta'}(s)$
4: **Require:** Replay buffer $\mathcal{D}$
5: **Require:** Boolean flag `use_action_gradient`
6: **Require:** Parameters of cVAE: Encoder $(\zeta)$, Decoder $(\psi)$, Prior $(\upsilon)$
7: **Input:** Initial parameters $\theta, \phi_1, \phi_2$ and target parameters $\theta' \leftarrow \theta, \phi'_1 \leftarrow \phi_1, \phi'_2 \leftarrow \phi_2$
8: **Input:** Policy update frequency $d$
9: **for** each episode **do**
10:     Initialize a random process $\mathcal{N}$ for action exploration
11:     Observe initial state $s_0$
12:     **for** each step in the episode **do**
13:         $a_t \leftarrow \pi_\theta(s_t) + \mathcal{N}_t$                              {Select action with exploration noise}
14:         Execute $a_t$, observe $(s_{t+1}, r_t)$
15:         Store $(s_t, a_t, r_t, s_{t+1})$ in replay buffer $\mathcal{D}$
16:         $\zeta, \psi, \upsilon \leftarrow$ **train_world_model**$(s_t, a_t, s_{t+1}, r_t)$         {Train cVAE world model and update parameters}
17:     **end for**
18:     **for** each gradient step **do**
19:         Sample mini-batch $\{(s_i, a_i)\}_{i=1}^N \sim \mathcal{D}$                          {Replay buffer sampling}
20:         Compute Distributional Loss via MMD (cf. Alg. 2):
21:         NB. Samples are concatenated and top $p\%$ are removed as in TQC (Kuznetsov et al., 2020)

22:         $L_{Z_1} \leftarrow \text{MMD\_Sobolev}(Z_{\phi_1}, (Z_{\phi'_1}, Z_{\phi'_2}), s_i, a_i, \pi_{\theta'}, \gamma, \text{use\_action\_gradient})$
23:         $L_{Z_2} \leftarrow \text{MMD\_Sobolev}(Z_{\phi_2}, (Z_{\phi'_1}, Z_{\phi'_2}), s_i, a_i, \pi_{\theta'}, \gamma, \text{use\_action\_gradient})$
24:         Update critics:
25:         $\phi_1 \leftarrow \phi_1 - \alpha_\phi \nabla_{\phi_1} L_{Z_1}$
26:         $\phi_2 \leftarrow \phi_2 - \alpha_\phi \nabla_{\phi_2} L_{Z_2}$
27:         **if** gradient step mod $d = 0$ **then**
28:             Sample $M_{\text{policy}}$ values from critic for each state in the batch:
29:             $Z_{\text{policy},1:M_{\text{policy}}}^{(i)} \overset{i.i.d.}{\sim} Z_{\phi_1}(s_i, \pi_\theta(s_i))$     for each $i \in \{1, \dots, N\}$
30:             Compute actor loss using double summation:
31:             $L_\pi = -\frac{1}{N} \sum_{i=1}^N \frac{1}{M_{\text{policy}}} \sum_{j=1}^{M_{\text{policy}}} Z_{\text{policy},j}^{(i)}$
32:             Update actor: $\theta \leftarrow \theta - \alpha_\theta \nabla_\theta L_\pi$
33:             Update target networks:
34:             $\phi'_1 \leftarrow \tau \phi_1 + (1 - \tau)\phi'_1$
35:             $\phi'_2 \leftarrow \tau \phi_2 + (1 - \tau)\phi'_2$
36:             $\theta' \leftarrow \tau \theta + (1 - \tau)\theta'$
37:         **end if**
38:     **end for**
39: **end for**

---

## I.3   BASELINE - MAGE

The primary baseline of this work is MAGE D'Oro & Jaskowski (2020), as they were the first to propose to use the action gradient to steer policy evaluation. The primary difference with our work is they did so deterministically. Hence our MMD setup collapses to a simple regression setting.

As a primary baseline, we consider the deterministic counterpart of the Sobolev-distributional backup. Define

$$f_{\text{det}}(s,a) = \begin{bmatrix} f_{\text{det}}^{\text{ret}}(s,a) \\ f_{\text{det}}^{\text{act}}(s,a) \end{bmatrix},$$

$$f_{\text{det}}^{\text{ret}}(s,a) = r(s,a) + \gamma\, Q_\phi(s',\pi(s')),$$

$$f_{\text{det}}^{\text{act}}(s,a) = \nabla_a r(s,a) + \gamma\, \nabla_a Q_\phi(s',\pi(s')).$$

$$\mathrm{L}^{S_a}(\phi;\, s,a) = \left| f_{\text{det}}^{\text{ret}}(s,a) - Q_\phi(s,a) \right|^2 \;+\; \lambda_S \left\| f_{\text{det}}^{\text{act}}(s,a) - \nabla_a Q_\phi(s,a) \right\|^2.$$

In practice, $s'$ and $r$ are sampled from the stochastic world model (cVAE) as in DSDPG. Furthermore, the above L2 terms are replaced by Huber losses, with the gradient-term weight set to 5 just as in D'Oro & Jaskowski (2020).

## J  TOY SUPERVISED LEARNING

To motivate our algorithm, we demonstrate its ability to learn the joint distribution over both the output and gradient of a random function in a supervised setup. We compare deterministic Sobolev training Czarnecki et al. (2017) against our Distributional Sobolev training and show that only the latter can capture the full variability of both outputs and gradients.

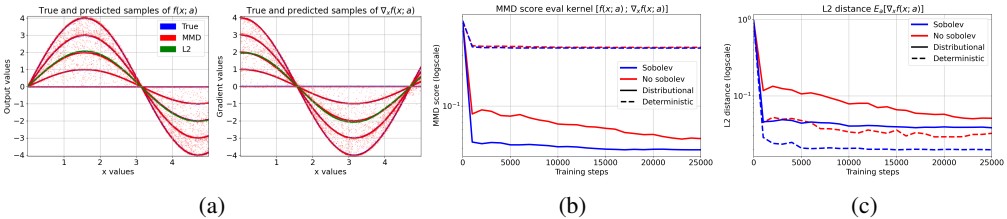

(a)    (b)    (c)

Figure 4: (a) Samples of the marginals of the full Sobolev distribution $[f(x;a); \nabla_x f(x;a)]$: output (left) and gradient (right). Blue: samples from the true distribution; red: samples from the Distributional Sobolev model trained via MMD; green: samples from the deterministic Sobolev baseline Czarnecki et al. (2017). (b) Biased MMD score (lower is better) on the joint variable. (c) $L_2$ error between predicted sample mean and true mean. **NB. Predicted (red and green) and true (blue) samples are highly overlapping.**

The task involves learning a one-dimensional conditional distribution $p(y|x)$, defined as a mixture of sinusoids $f(x;a) = a \times \sin(x)$, where the latent variable $a$ is uniformly drawn from $\{0,1,2,3,4\}$. The distributions over outputs $f(x;a)$ and their gradient $\nabla_x f(x;a)$ are depicted in Figure 4a. It compares an MMD-based model and a regression-based model, trained using stochastic gradient descent with identical architectures. In an unlimited data regime, new pairs of $x$ and $a$ were drawn for each batch, and for each $x$, four $a$ values were sampled with replacement, yielding samples $(x, y_{1:4})$.

As expected, the MMD-based model captures the full joint distribution $[f(x;a);\ \nabla_x f(x;a)]$, whereas the regression baseline collapses to the conditional mean $\mathbb{E}_a[f(x;a); \nabla_x f(x;a)]$. Figure 4b plots the MMD score (lower is better) on this joint variable using an evaluation kernel, and Figure 4c shows the $L_2$ error between the regression model and the sample mean of the Sobolev generator. As a result, the MMD model better matches the entire distribution, the regression model slightly outperforms on the mean, and both methods effectively exploit the gradient signal (blue curves).

The distributional variant via MMD and the regression one via L2 used the same architecture except for some noise of dimension 10 drawn from $\mathcal{N}(0; I)$ concatenated to the input for the distributional generator. For each pair $(x, y_{1:4})$, four samples were drawn from the generator. Both were trained using Rectified Adam Liu et al. (2019) optimizer with a learning rate of $1 \times 10^{-3}$ and $(\beta_0, \beta_1) = (0.5, 0.9)$. Neural network is a simple MLP with 2 hidden layers of 256 neurons and Swish nonlinearities (Ramachandran et al., 2017).

Maximum Mean Discrepancy (MMD) was estimated using a mixture of RBF kernels with bandwidths $\sigma_i$ from the set $\{\sigma_1, \sigma_2, \ldots, \sigma_7\} = \{0.01, 0.05, 0.1, 0.5, 1, 10, 100\}$. We used the biased estimator from Equation 25.

The equation for a mixture of RBF kernels is given by:

$$k^{\text{mix}}(x, y) = \sum_{i=1}^{7} \exp\left(-\frac{\|x - y\|^2}{2\sigma_i^2}\right).$$
(52)

The evaluation kernel we used was the Rational Quadratic $k_\alpha^{\text{RQ}}$ with $\alpha = 1$ with

$$k_\alpha^{\text{RQ}}(x, y) = \left(1 + \frac{\|x - y\|^2}{2\alpha}\right)^{-\alpha}$$
(53)

Regarding the dataset, the $(x, y_{1:4})$ pairs were drawn with $x \sim \mathcal{U}[0; 5]$ and $a$ was drawn from $\{0, 1, 2, 3, 4\}$ with replacement. In the limited data regime, the pairs $(x, y_{1:4})$ were drawn once and stayed fixed. The batch size was thus equal to the number of points in the dataset. In the unlimited data regime 256 new pairs were drawn for each batch. Every experiment was run for 25 000 batch samples and thus the same number of SGD steps.

**Limited data regime** In both supervised and reinforcement learning tasks, the assumption of unlimited data is unrealistic. Here, we explore how the performance of the MMD-based and regression-based models diverges when the amount of available data is restricted. We use the same setup as before, but with a fixed number of $(x, y_{1:4})$ pairs. Several aspects of the learned models can be inspected. In order to assess stability, we report the average norm of the second-order derivative over the input space. For accuracy, we measure the average L2 losses between the true expected gradient and the predicted gradient. Results are shown in Figure 5. As can be seen, the deterministic model tends to overfit rapidly, while the distributional variant (MMD) proves more robust, maintaining better performance even with constrained data. Notably, the second-order derivative for the deterministic model escalates sharply as data becomes constrained, indicating instability in its approximation. In Appendix J.1, we discuss different common tricks to mitigate overfitting and related issues in RL.

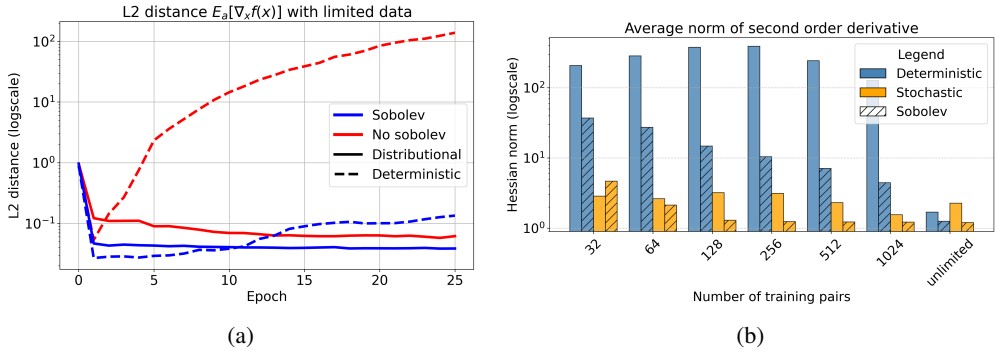

(a)  (b)

Figure 5: Toy supervised learning problem. Comparison between MMD-based or L2 based modeling. Left panel: training curve of L2 loss (logscale) on gradient between true conditional expectation with regression prediction and with empirical mean from MMD-based model. Sobolev (blue) used gradient information to train either using MMD (full line) or L2 regression (dashed line). Right panel: average over the input space of the second order derivative (logscale) of predicted gradient from deterministic model (blue), MMD / stochastic (yellow) and with gradient information / Sobolev (dashed). Metrics averaged over 5 seeds.

## J.1 COMMON TRICKS

Overfitting with limited data is a common issue in regression tasks. Early stopping seems an obvious solution in this case but we emphasize that it requires an evaluation criterion that is not always

available (for example, in policy evaluation). Other solutions include weight regularization Krogh & Hertz (1991), dropout Srivastava et al. (2014), Bayesian neural networks Blundell et al. (2015), ensembling Chua et al. (2018), and spectral normalization Zheng et al. (2023), all of which often reduce network capacity. To address similar issues, Fujimoto et al. (2018) proposed adding noise to the target to match, effectively smoothing the critic. As argued by Ball & Roberts (2021), this method can be seen as indirectly acting like spectral normalization, encouraging smoother gradients and effectively reducing the magnitude of the second derivative. Appendix J.2 shows how noise scale impacts overfitting by inducing bias. On the other hand, we propose avoiding such assumptions by using generative modeling to add latent freedom.

### J.2    ADDING NOISE

Inspired by Fujimoto et al. (2018), we added some independent noise on $x$ for each $(x, y_{1:4})$. Noise scale $\sigma$ was in $\{0.01, 0.1, 0.5\}$. For each new batch, it was sampled from a standard Gaussian $\eta \sim \mathcal{N}(0; \sigma^2)$ and added as $\tilde{x} = x + \eta$

In Figure 6-7, we can see the impact of the various noise scales on the predictions of the deterministic regression. As can be seen, adding noise on $x$ has a positive effect in terms of stabilizing the gradient but it induces a bias that grows with the scale of the noise. Moreover, this noise depends on the application and makes strong assumptions about the function to learn. The stabilizing effect of additive noise can further be seen in Figure 8-9 where both the L2 loss and average second order derivative are displayed as a function of the number of sampled locations.

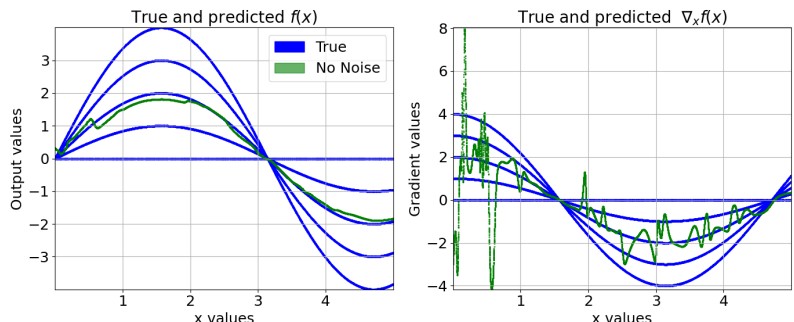

Figure 6: Toy supervised learning problem. Comparison of samples from the true five-mode distribution with predictions made by a deterministic model trained with L2 loss (green). The output space is shown on the left, and the gradient space on the right. Results obtained after 25,000 training steps.

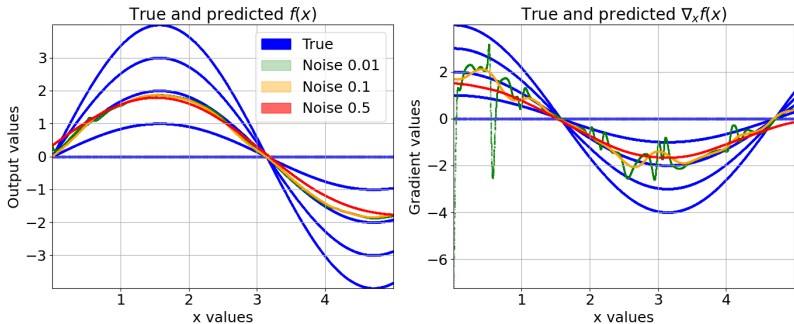

Figure 7: Toy supervised learning problem. Comparison between true samples from the distribution of five modes and deterministic models trained with varying levels of additive noise on their input data. Low level of noise (green), medium level of noise (orange), high level of noise (red). Results obtained after 25,000 training steps.

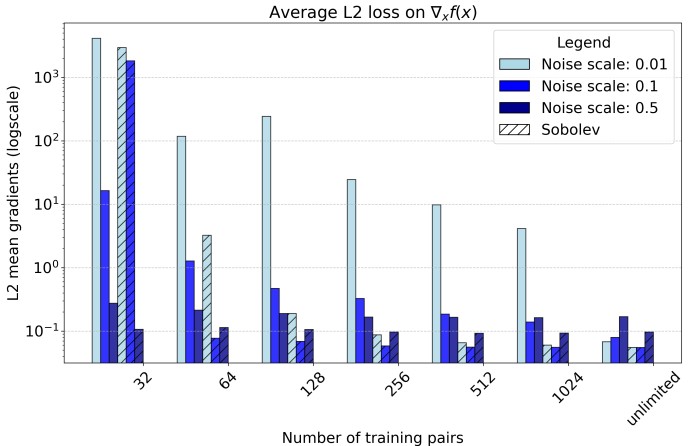

Figure 8: Toy supervised learning problem. Comparison of the L2 loss between the predicted gradient and the conditional expectation of the true distribution. Different scales of additive noise on the input are compared: low noise (light blue), medium noise (medium blue), and high noise (dark blue), alongside Sobolev training (dashed). Results are shown after 25,000 training steps.

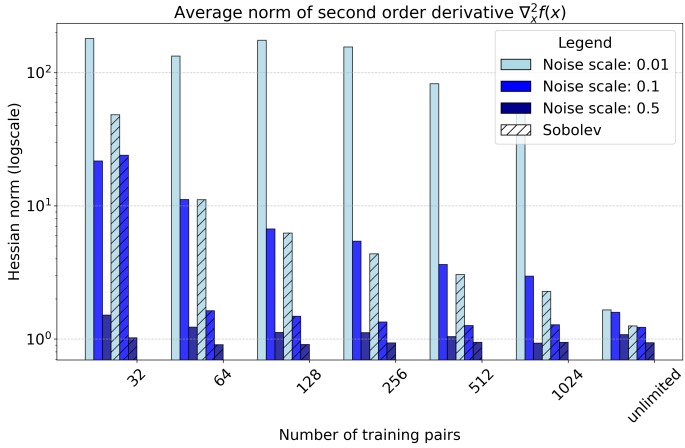

Figure 9: Toy supervised learning problem. Comparison of the average second order derivative norm over the input space. Different scales of additive noise on the input are compared: low noise (light blue), medium noise (medium blue), and high noise (dark blue), alongside Sobolev training (dashed).

## K  TOY REINFORCEMENT LEARNING

### K.1  ENVIRONMENT

In this appendix, we provide full details of our custom toy environment, summarized in Table 1.

**Environment Dynamics.**    A point mass moves in a two-dimensional continuous state space, and the agent controls its acceleration in $[-1, 1]$ per axis (Acceleration range). At each step the mass's velocity is updated via Euler integration with friction (Friction coefficient) and then its position is advanced by the new velocity (Integration time step). The mass has unit mass (Mass of agent) and radius 0.5 (Agent radius), and if its center leaves the square of half-width 3 (Bounding box half-width) the episode ends with no reward (Reward for leaving bounding box).

**Partial Observability via Memory.**    At the start of each episode the mass is placed uniformly in a square of half-width 1 (Initialization area half-width). We then sample one of $M \in \{3, 4, 5, 6\}$ hidden bonus locations arranged radially around the center. Reaching the correct location of radius 0.5 (Bonus radius) yields a terminal reward (Reward for reaching bonus). The agent does not know which location is active, but each visit to a location sets a binary memory flag, making the MDP partially observable and forcing exploration.

**Controlling Distributional Modes via $M$.**    By sweeping $M \in \{3, 4, 5, 6\}$ we directly tune the number of modes in the return distribution across episodes, from concentrated when $M$ is small to highly multimodal when $M$ is large.

| Parameter | Value |
| --- | --- |
| Maximum episode length | 25 steps |
| Reward for reaching bonus | 10 |
| Reward for leaving bounding box | 0 |
| Acceleration range | $[-1, 1]$ per axis |
| Mass of agent | 1 |
| Agent radius | 0.5 |
| Bonus radius | 0.5 |
| Bounding box half-width | 3 |
| Initialization area half-width | 1 |
| Friction coefficient | 0.1 |
| Integration time step | 0.5 |

Table 1: Summary of the toy environment's key parameters.

### K.2  EXPERIMENTAL DETAILS

The settings were similar to Section L apart from the TQC Kuznetsov et al. (2020) truncation parameter $p$ which was set to $5\%$.

## L  REINFORCEMENT LEARNING EXPERIMENTS

Here we describe the architectures, optimizers and other hyperparameters of the full Distributional Sobolev Deterministic Policy Gradient algorithm.

**Policy network**   Policy network is a MLP with 2 hidden layers of 400 neurons. The non-linearity was Swish (Ramachandran et al., 2017). Final activations are mapped to the output space using a linear transformation followed by a $\tanh$ non-linearity. The policy network was optimized using the Rectified Adam Liu et al. (2019) with a learning rate of $1 \times 10^{-4}$.

**Critic network**   Critic network architecture is almost the same as for the policy network. Importantly, it is kept constant for experiments using normal DDPG and DSDPG apart from noise concatenated on the input $[s; a]$. The network is a MLP with 400 neurons, skip connections from the

Table 2: Hyperparameters for the DDPG and DSDPG experiments on MuJoCo environments

| Item | Value |
|------|-------|
| Discount $\gamma$ | 0.99 |
| Polyak averaging $\tau$ | 0.005 |
| Buffer size | $10^6$ |
| Batch size | 256 |
| Exploration noise scale | 0.1 |
| Critic learning rate | $1 \times 10^{-4}$ |
| Policy learning rate | $1 \times 10^{-4}$ |
| cVAE learning rate | $1 \times 10^{-4}$ |
| cVAE weight decay | $1 \times 10^{-4}$ |
| cVAE KL weight | 0.1 |
| cVAE latent dim | $|\mathcal{S}| + 1$ |
| Critic input noise dim | 64 |
| Number of samples dist. | 10 |
| $\%p$ truncation (TQC) | 25% |
| maximum slicing optimization steps | 100 |
| maximum slicing LR | 1e-4 |
| maximum slicing optimizer | Adam (Kingma, 2014) |

input and Swish activations. No non-linearity was applied on the output after the last linear layer. The critic network is optimized using Rectified Adam Liu et al. (2019) with $(\beta_1, \beta_2) = (0.9, 0.999)$ and a learning rate of $1 \times 10^{-4}$.

**Conditional VAE world model**    The encoder, decoder and prior networks are MLPs with 3 hidden layers, each containing 1024 neurons. Skip connections are applied from the input to each hidden layer. The cVAE is optimized using RAdam with weight decay $1 \times 10^{-4}$, $(\beta_1, \beta_2) = (0.9, 0.999)$, and a learning rate of $1 \times 10^{-4}$.

The prior is a learned diagonal multivariate Gaussian $\mathcal{N}(\varepsilon; \mu_v(s, a), \sigma_v^2(s, a) \odot I)$ with a latent dimension equal to the size of the random variable being modeled, which is $|\mathcal{S}| + 1$ for $(s', r)$. Following D'Oro & Jaskowski (2020); Zhu et al. (2024), the cVAE predicts the difference between the current and next observation, $\delta_s = s' - s$ which is then added back to $s$, along with the reward $r$.

**Conditional Generative Moment Matching**    For the distributional critic, noise vectors were concatenated with the state-action pairs $(s, a)$ and passed through the same architecture as the deterministic critic. The noise dimension was set to 64. Noise was transformed by an independent 2 layers small MLP of width 64 with Swish activation before being passed to the critic. For each state-action pair, 10 samples were drawn to update both the critic and the policy. The multiquadric kernel $k_h^{\mathrm{MQ}}(x, y) = -\sqrt{1 + h^2 \|x - y\|_2^2}$ Killingberg & Langseth (2023) was used, with the kernel parameter $h$ set to 100.

## L.1 FULL CURVES AND WALL-CLOCK TIME

The full evaluation curves on the 6 environments are reported in Figure 10. The wall-clock time of the different methods is displayed in Table 3.

## M    ADDITIONAL EXPERIMENTS - RL

In this section we provide further experiments along three axes: kernel bandwidth (Section M.3), noise scale (Section M.4), number of samples in the distributional methods (Section M.5) and capacity of the world model (Section M.6).

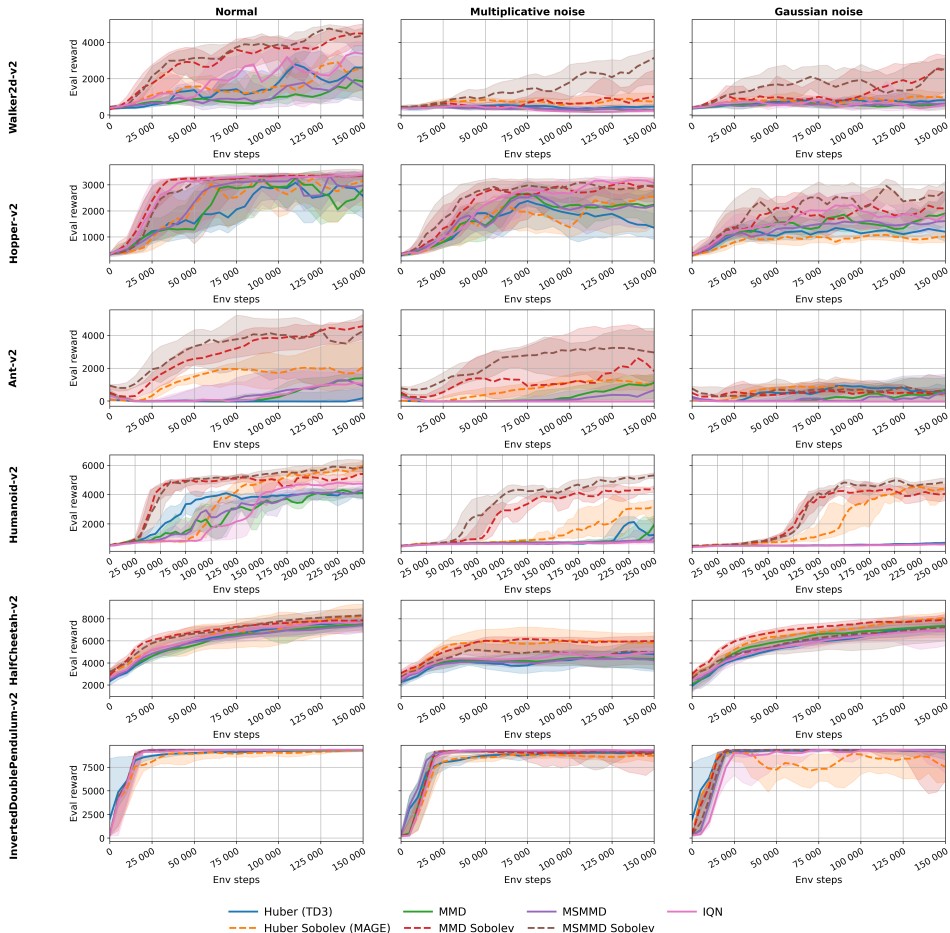

Figure 10: Evaluation of DSDPG (MMD Sobolev), deterministic Sobolev/MAGE D'Oro & Jaskowski (2020), TD3-Huber Fujimoto et al. (2018), IQN Dabney et al. (2018a), and standard MMD Nguyen et al. (2020); Killingberg & Langseth (2023) on six MuJoCo tasks. Results are reported over 10 random seeds. The median is displayed with 25%-75% IQR. Window smoothing with window size 3.

| Method | Time for 1000 iterations (s) |
|---|---|
| MSMMD Sobolev | 62.5 |
| MSMMD | 45.5 |
| MMD Sobolev | 40.0 |
| MMD | 35.7 |
| IQN | 35.7 |
| Huber Sobolev | 31.3 |
| Huber | 30.3 |

Table 3: Wall-clock time to perform 1000 iterations of each method under Humanoid-v2. Experiments were run on a single Nvidia H100 GPU.

### M.1 CONDITIONAL FLOW WORLD MODEL

The conditional flow world model is implemented using a RealNVP architecture (Dinh et al., 2016). Each affine coupling layer is parameterized by a conditioner MLP with two hidden layers of width 512, using SiLU activations (Elfwing et al., 2018) and LayerNorm (Ba et al., 2016) after each hidden layer. The linear transformations within the coupling blocks use the invertible $1 \times 1$ transformation parameterized through an LU decomposition as introduced in Glow (Kingma & Dhariwal, 2018), which enables learned invertible mixing of feature dimensions while keeping the log-determinant of the Jacobian inexpensive to compute. As in the cVAE world model, the flow predicts the difference $\delta_s = s' - s$ along with the reward $r$, and the next state is reconstructed as $s' = s + \delta_s$.

Following the same output parameterization as the cVAE world model, the flow predicts the concatenated vector $(\delta_s, r)$ of dimension $|\mathcal{S}| + 1$, where $\delta_s = s' - s$. The next state is then recovered as $s' = s + \delta_s$.

Training proceeds by maximizing the conditional log-likelihood $\log p_\phi(\delta_s, r \mid s, a)$ using the same optimizer as for the cVAE. Sampling draws $z \sim \mathcal{N}(0, I)$ from the base distribution and applies the inverse flow conditioned on $(s, a)$,

$$(\delta_s, r) = f_\phi^{-1}(z; s, a).$$

Because the model is fully reparameterized, each sample $(\delta_s, r)$ admits pathwise derivatives $\partial(\delta_s, r)/\partial(s, a)$, which are required for computing Sobolev temporal-difference targets.

The empirical performance of the normalizing flow based variant of our method is displayed in Figure 11. As can be seen, the distributional Sobolev variants we propose still outperform the deterministic variant proposed in MAGE (D'Oro & Jaskowski, 2020) under the stochastic settings introduced in Section 6.2.

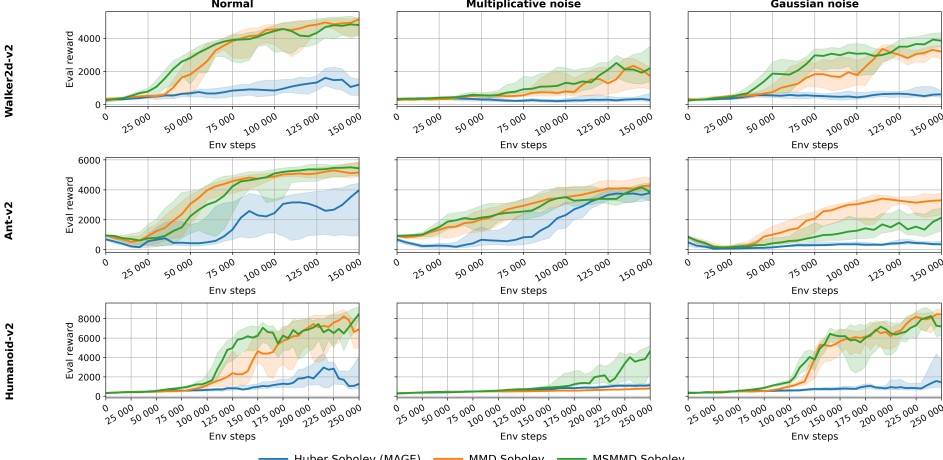

Figure 11: Full curves of DSDPG and MAGE (D'Oro & Jaskowski, 2020) using a RealNVP (Dinh et al., 2016) normalizing flow as differentiable world model.

## M.2    OVERESTIMATION BIAS

In our implementation we rely on the TQC trick (Kuznetsov et al., 2020), which mitigates overestimation by discarding the largest values in the target distribution. This mechanism is introduced and discussed in Section I.1. It is essential for stable learning: in our ablations, removing truncation leads to substantially degraded performance. For completeness, we also remove the double-estimation correction from MAGE (D'Oro & Jaskowski, 2020). These ablations show that controlling overestimation is critical for reliable training. The corresponding results are reported in Figures 12a and 12b, which present the final evaluation performance and the normalized Area Under the Curve with and without the overestimation–bias countermeasure.

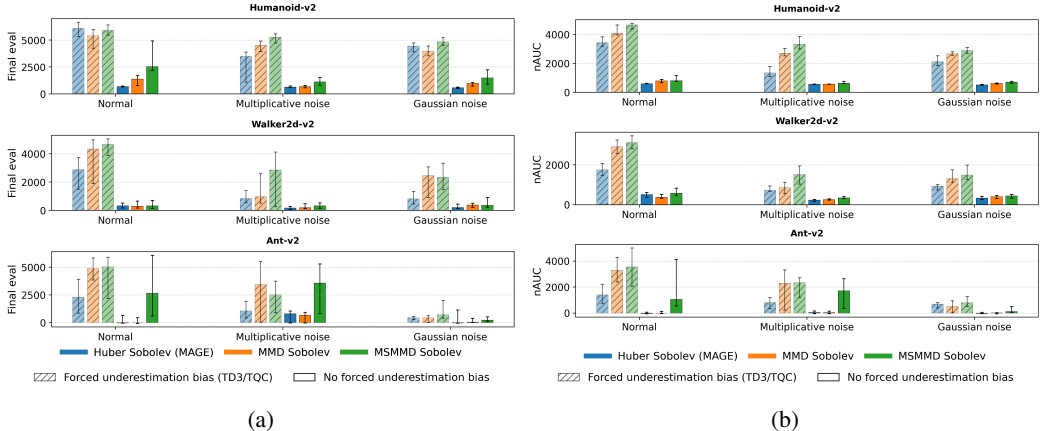

Figure 12: Comparison of MAGE (D'Oro & Jaskowski, 2020) and DSDPG with and without overestimation bias measures. We removed the double estimation of MAGE borrowed from (Fujimoto et al., 2018) and the TQC trick (Kuznetsov et al., 2020) of our DSDPG method.

## M.3    MULTIQUADRIC KERNEL BANDWIDTH

We recall the multiquadric kernel

$$k_h^{\mathrm{MQ}}(x, y) = -\sqrt{1 + h^2 \|x - y\|_2^2}.$$

As shown in Killingberg & Langseth (2023), selecting a proper value for $h$ is critical as it affects both the expressiveness of the kernel and the numerical stability of its MMD estimator: a too–small $h$ yields a nearly constant kernel, reducing its discrimination power, while a too–large $h$ increases the gradient magnitude, scaling on the order of $h$, which can lead to exploding gradients and training oscillations unless mitigated by techniques such as kernel rescaling or gradient clipping.

In Section 6.2, the default value for $h$ was 100 which worked reliably. Here we test the values $h = 10$ and $h = 250$. The results are depicted in Figure 14 and 13 where we display the final evaluation performance and the evaluation curves. The comparison is made on the environments with and without multiplicative noise (as in Section 6.2). As can be seen, overall, the performance of MMD and DSDPG (MMD Sobolev) seems to be quite insensitive to the kernel bandwidth. Our choice of $h = 100$ seems to be robust.

## M.4    NOISE SCALE

Since the noise scale we exposed in Section 6.2, which we called *medium*, was already sufficient to observe a gap between DSDPG and the baseline, we tried to increase the noise scale. We moved from $n \sim \mathcal{U}[0.8, 1.2]$ to $n \sim \mathcal{U}[0.7, 1.3]$. These changes already made the tasks harder. The results are displayed in Figure 15. As can be seen, the good performance of DSDPG (MMD Sobolev) depicted in Section 6.2 can be extended to the larger noise scale, as our method maintains a consistent gap against the baselines (especially MAGE).

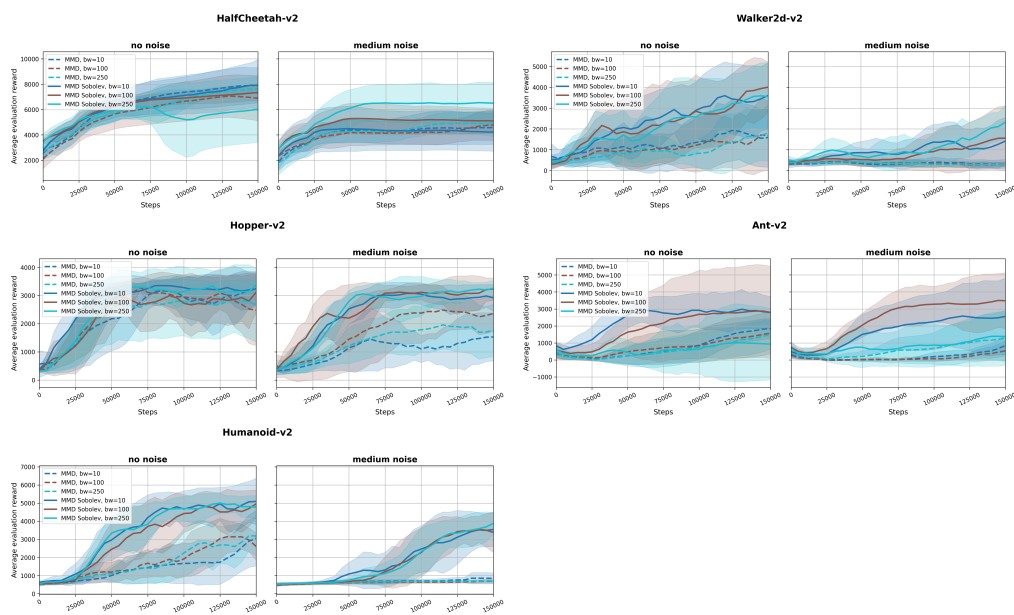

Figure 13: Full curves of DSDPG against baselines on 5 MuJoCo environments.

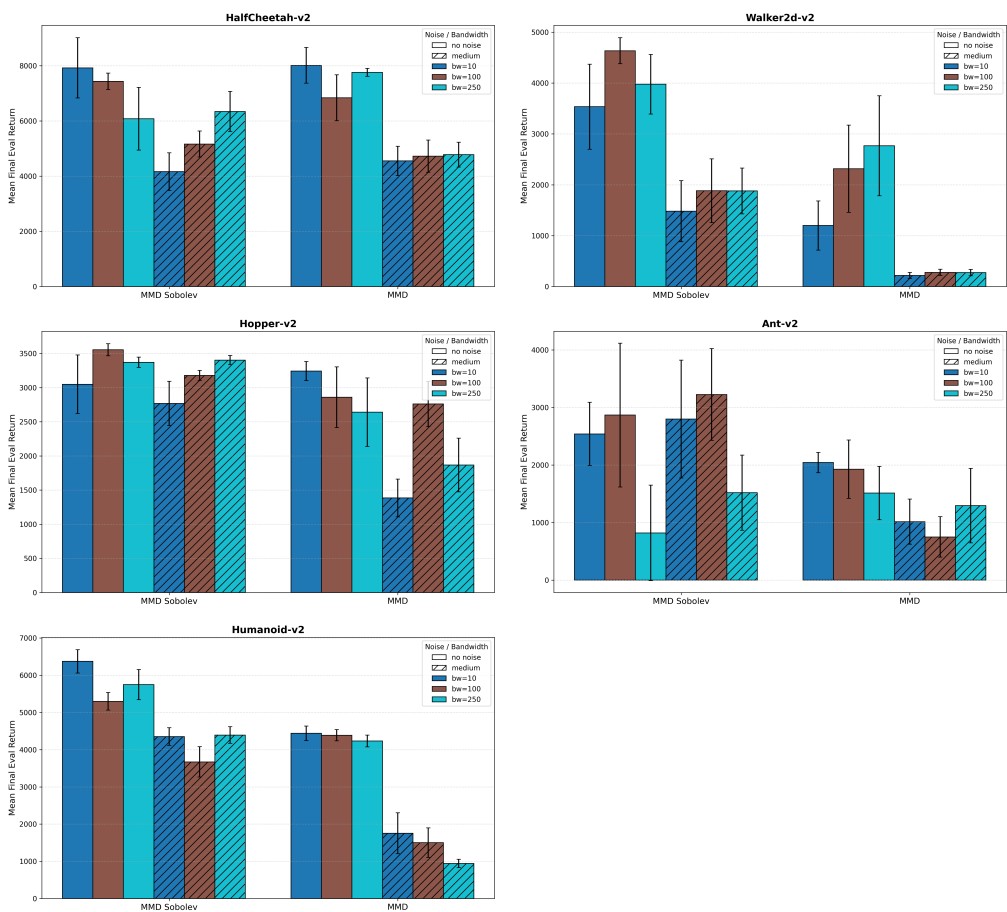

Figure 14: Comparison of bandwidths under the multiquadric kernel for MMD distributional RL and Sobolev MMD distributional RL. Bar plots of the final averaged evaluation reward. Mean over 5 seeds.

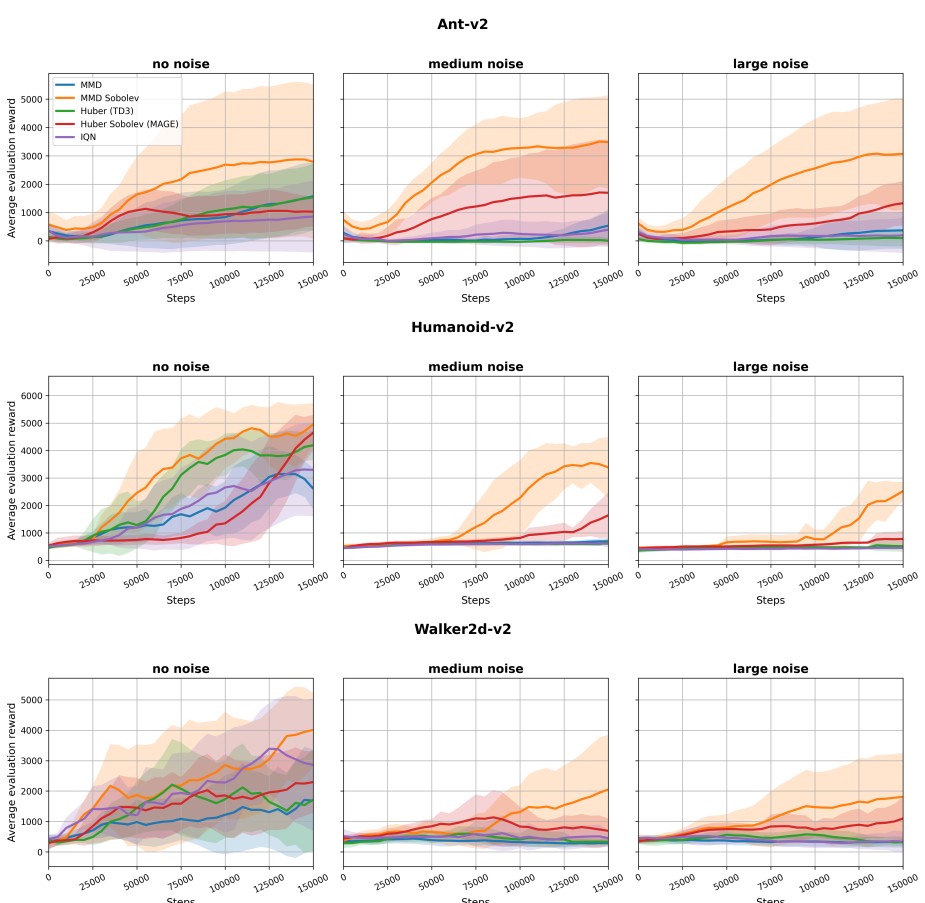

Figure 15: Comparison of multiplicative noise scales. Averaged evaluation sum of rewards over 5 seeds.

## M.5 NUMBER OF SAMPLES

Sensitivity to the number of samples is an important design question as the cost of generating samples will scale linearly and the cost of estimating MMD will scale quadratically. It is also interesting to verify how sensitive our method is to that parameter. The results displayed in Section 6.2 used 10 samples to model the Sobolev distributions. Here we additionally test with 5 and 25 samples. In Figure 17 and 16 we display the final evaluation returns and evaluation curves while varying the number of samples. As can be seen, there is no clear trend to be found. The Ant-v2 environment seems to be particularly sensitive to this parameter. Apart from this example we observe our method is robust to the number of samples as at least two different values maintained performance.

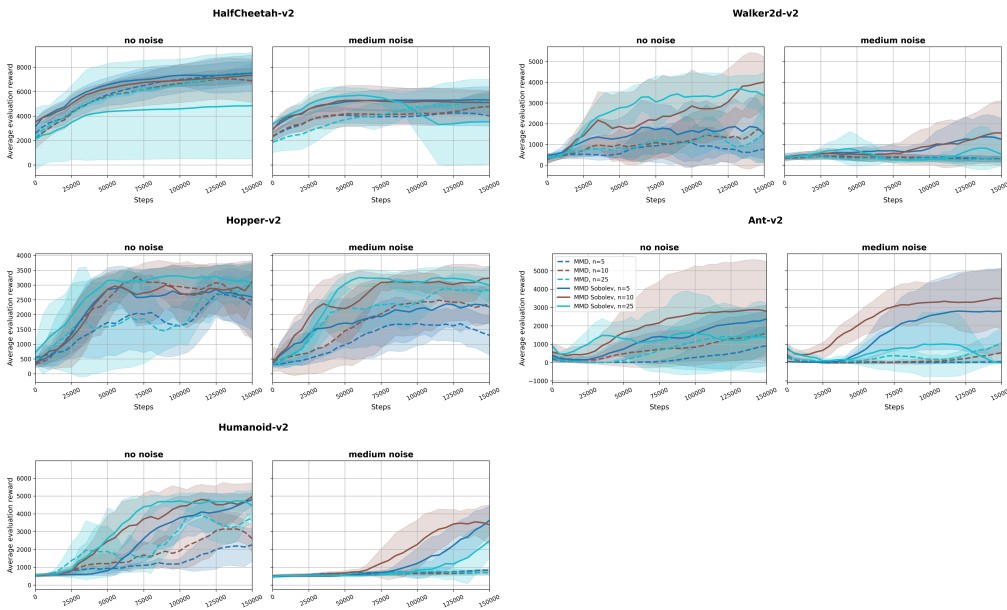

Figure 16: Comparison of number of samples used in the MMD methods (usual MMD and DSDPG denoted as MMD Sobolev). Evaluation curves are mean over 5 seeds.

## M.6 WORLD MODEL CAPACITY

The world model size is, along the number of samples, one of the primary parameter driving the overall cost of our method. As for MAGE D'Oro & Jaskowski (2020), we backpropagate through the world model to infer the gradients of the target. Here we change the width of each neural network in the cVAE world model from 1024 to 256. We observe in Figure 18, on the single Walker2d-v2 environment we tested, rather than being insensitive, it seems the choice of 1024 was sub-optimal for this environment. This suggests our method is robust to this design choice. This should be evaluated on more demanding (higher dimensional) environments (Ant-v2, Humanoid-v2).

## N LLM USAGE

We used an LLM-based assistant to support the preparation of this paper. In particular, it was employed to (i) rephrase draft paragraphs for clarity and suggest alternative framings of related work, (ii) format proofs, explore directions, and verify intermediate steps, (iii) assist in debugging code, (iv) suggest LaTeX equation formatting, and (v) help identify relevant theoretical results in preceding works. All core research contributions, including the development of theoretical results, algorithms, and experiments, were carried out by the authors.

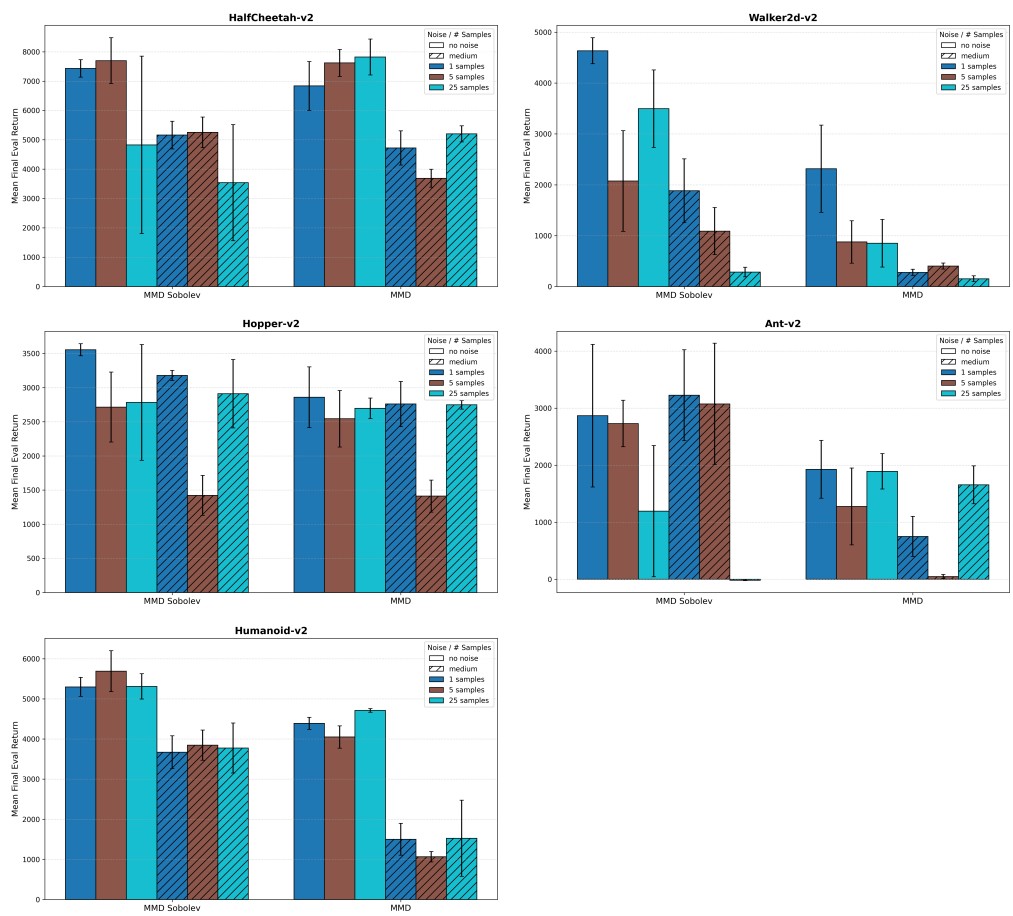

Figure 17: Comparison of number of samples used in the MMD methods (usual MMD and DSDPG denoted as MMD Sobolev). Final evaluation performance as mean over 5 seeds.

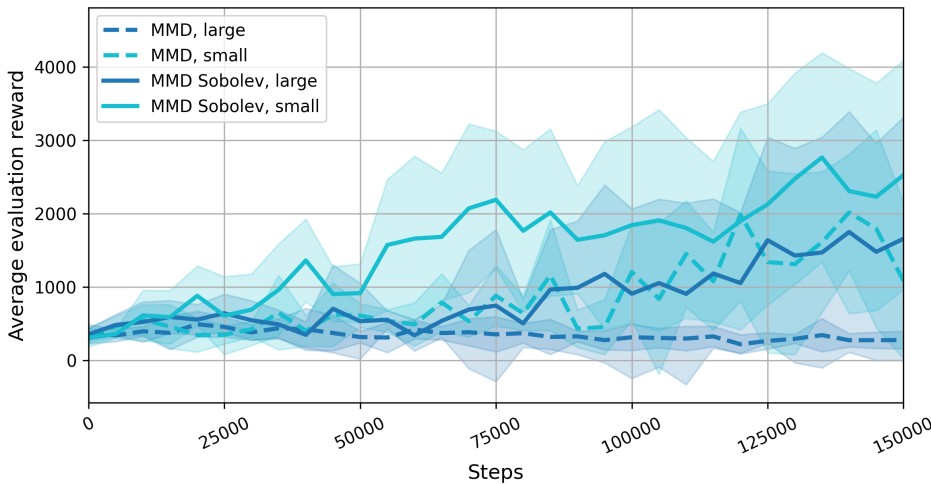

Figure 18: Comparison of world model size for MMD distributional RL and Sobolev MMD distributional RL. Bar plots of the final averaged evaluation reward. Mean over 5 seeds.

