# OpenReview forum: "Distributional value gradients for stochastic environments"
_ICLR.cc/2026/Conference — ICLR 2026 Poster_

### Official Review · Reviewer_Z5td · 2025-10-27

**Soundness:** 1
**Presentation:** 2
**Contribution:** 1
**Rating:** 2
**Confidence:** 4

**Summary:**

This paper proposes a new deterministic policy gradient method in the continuous action space by using Sobolev training, which additionally incorporates the gradient information in the distributional learning. Theoretically, the authors show the contractivity with Max-sliced MMD and Wasserstein distance under some assumptions. The dynamics and reward transitions are also needed to learn to derive a tractable algorithm. Experiments are conducted in a toy example and several Mujoco environments.

**Strengths:**

1.Incorporating the action-gradient in the distributional learning seems novel.

2.Some theoretical results are provided and also many details have been discussed, which are deferred to Appendix.

**Weaknesses:**

1. **The motivation is not clear**. Why is introducing action gradients beneficial in distributional learning? Directly combining Sobolev training seems straightforward. A more profound analysis is needed to strengthen the motivation.

2. **Many theoretical results are questionable with impractical or unclear conditions.** I push back against the statement in line 142 he Lipschitz continuity of $\pi$ typically holds when using neural-network function approximation, which is not true. The Lipschitz constant of deep neural networks is often unboundedly large, which depends on the spectral norms of the weight matrix that is often unbounded during the training. The contraction in the theorem only holds when $\gamma \kappa <1$, which is hard to justify for any practical environment. This renders the contraction vacuous and meaningless. What are the mild conditions in Theorem 2, which are very unclear. What is $\kappa$ in a rigorous way?

3. **Limited experiments and insignificant improvements**. In distributional RL literature, published papers have done extensive experiments with significant improvement, making the proposed algorithm deployable. However, the improvement in a toy example is too weak, and the improvement in the Mujoco environments is very limited. Although the robustness setting is helpful to some extent, it is unsatisfactory to find that the proposed algorithm only matches the performance with the baselines, even though it has additionally learned the environment and there is a much higher computational cost.

4. **The writing is not clear.** Many sentences are in bold, but they are hard to follow. For instance, what do the authors want to express in Line 203? It is in bold but may not be strongly related to the other parts of the paper. A lot of details are provided in the appendix, which are very distracting. The main content of the paper is not even self-contained. For example, it is surprised that many key details in Theorem 2 are deferred to the appendix, which largely undermines the readability of this paper. I could not understand why we should not expect the proposed algorithm to be superior over all the environments in general.

**Questions:**

1.What is the environmental non-differential regarding?

2.The formulation in Eq.10 is questionable. Note that a is a constant value as the initial action, why do you mean the gradient is taken regarding a constant? In addition, the notation of cumulative rewards does not involve the action. Since it is a continuous action space, what does $|A|$ mean? Essentially, it should be $\mathbb{R}^d$.

3.What is the computational cost when we use max-sliced MMD? It seems that it is very high, but we are disappointed to find that the improvement is insignificant even with the higher computational cost.

4.Why do we need to incorporate the TQC trick? Is there any ablation study in experiments compared with other baselines?

5.The introduction of CVAE is less convincing. Why do we not directly train a diffusion model?

6.The theory and the practical algorithm seem to have a gap. The finally proposed algorithm does not enjoy a theoretical contraction guarantee.

---

> ### Author Response · Authors · 2025-11-23
> **Part 1**
>
> We thank the reviewer for the detailed and thoughtful feedback, and we address the points below.
>
> **On the motivation for incorporating action-gradients in a distributional framework**
> For the gradient-aware part of our framework, we rely on a distributional extension of a result from MAGE [1, Proposition 3.1], which shows that reducing the error in the critic’s action-gradient bounds the error in the resulting policy gradient. This motivates learning the action-gradient directly, and we extended this using Proposition 1 to the distributional setting. On the distributional side, our controlled toy experiment shows that when stochasticity is introduced, the Sobolev return becomes variable in a way that cannot be captured by a single point estimate. In this setting, the gradient information remains useful only if the Sobolev return is modelled as a distribution, which motivates the distributional formulation of our critic.
>
> **On the Lipschitz assumptions used in our analysis**
> A common assumption in theoretical analyses of RL algorithms is that the function approximators involved are Lipschitz, and the same assumption is used in gradient-aware RL methods such as MAGE [1]. Our use of it follows the same reasoning. We agree that a neural network with unbounded weights cannot be Lipschitz, but such behaviour corresponds to a divergent regime that is not typical in actor–critic training. A standard MLP with bounded weights and Lipschitz activations is itself Lipschitz, and if the parameters were to diverge in practice we would observe clear instabilities during training, which we do not. The well-known exception is the discriminator in WGANs [2], where the loss encourages the Lipschitz constant to grow, requiring explicit constraints such as clipping or spectral normalization. This is not the case in our setting. To make this explicit, we added a short subsection in Appendix **C.2** detailing the mild conditions under which the policy and critic networks used in our analysis are Lipschitz.
>
> **On contraction and assumptions on the environment**
> The reviewer wrote: *“The contraction in the theorem only holds when , which is hard to justify for any practical environment. This renders the contraction vacuous and meaningless.”*
> Our contraction analysis explicitly characterizes the assumptions on both the environment and the discount factor under which contraction can be guaranteed. **To the best of our knowledge, identifying this tradeoff between the smoothness of the dynamics and the amount of discounting is new.** Importantly, we do not claim contraction for any specific environment; rather, we state the precise conditions under which contraction holds. In particular, environments with less smooth dynamics require a correspondingly smaller discount factor for the contraction argument to apply.
>
> **On comparison with the baselines**
> The reviewer writes that our method *“only matches the performance of the baselines, even though it has additionally learned the environment and there is a much higher computational cost.”*
> We would like to clarify two points. First, our main baseline (MAGE) also relies on a learned world model, so both methods incur the same model-learning overhead relative to non–gradient-aware baselines. Second, the additional cost specific to our method comes from drawing multiple samples from the distributional critic. Table 3 in Appendix N reports the wall‐clock measurements: MMD‐Sobolev adds approximately 28% over MAGE, while MSMMD‐Sobolev doubles the cost. In settings with substantial stochasticity, this overhead **can be reasonable** given the additional robustness that comes from modelling the distribution of Sobolev returns rather than relying on a single point estimate.
>
> **On line 203**
> The reviewer refers to the sentence *“While these works considered the target action-gradient for regularization, we consider it as a part of the bootstrapped quantity at equal level with the scalar value.”*
> The purpose of this sentence was to highlight one of the main conceptual differences between our approach and prior gradient-aware methods. Earlier work uses the action-gradient only as a regularization signal to make the critic’s gradient more accurate for policy improvement, but this viewpoint does not naturally support discussing contraction or fixed points. In contrast, our method treats the action-gradient as an integral part of the bootstrapped quantity itself, leading to a Sobolev temporal-difference operator defined jointly over the return and the action-gradient. This framing is what enables our contraction analysis. We have revised the text around line 203 to make this distinction clearer.

---

> ### Author Response · Authors · 2025-11-23
> **Part 2**
>
> **On the assumptions stated in Theorem 2**
> We agree that the statement in the main text is concise. This was a deliberate choice due to space constraints and to keep the presentation readable. The full version in the appendix makes all assumptions explicit: they are exactly the properties of the MMD induced by the chosen kernel together with bounded Jacobians of the policy and the dynamics. These are the same conditions used throughout our analysis, and no additional assumptions are introduced in the theorem.
>
> **On the expectation of universally superior performance**
> The reviewer writes: *“I could not understand why we should not expect the proposed algorithm to be superior over all the environments in general.”*
> Contraction guarantees for MSMMD characterize properties of the Sobolev Bellman operator, not an across-the-board performance improvement of the algorithm. As we explain in our reply to Reviewer Hcbc01, the relationship between contraction guarantees and empirical performance is not straightforward, and there is precedent for this in MMD-based distributional RL. These guarantees ensure stability of the operator under specific assumptions, but they do not imply universal superiority over all baselines and environments.
>
> **On the meaning of a non-differentiable environment**
> The reviewer asks: *“What is the environmental non-differential regarding?”*
> We understand this question as asking what we mean by a “non-differentiable environment,” but please correct us if we have misunderstood. In many practical settings, the transition dynamics and reward function do not provide analytic derivatives, and this lack of access to environment Jacobians is one of the core motivations behind reinforcement learning as a whole: agents must learn to improve without relying on known gradients of the environment. In gradient-aware RL, computing the targets for action-gradients requires these Jacobians, but in most environments they are either undefined or simply unavailable. This motivates the use of a learned world model. In our case, we use a differentiable world model parametrized by a cVAE, so that samples from the model are differentiable and allow us to compute the action-gradient targets required for Sobolev temporal differences.
>
> **On Equation (10)**
> The reviewer writes: *“The formulation in Eq. 10 is questionable…”*
> At this point in the paper we are in the differentiable setting, where both the transition dynamics and the reward function are assumed differentiable. In this setting, although the initial action $a_0 = a$ is evaluated at a particular value, the cumulative return $\sum_{t=0}^\infty \gamma^t r(s_t, a_t)$ is a differentiable function of that initial action because all subsequent states and actions depend on $a_0$ through the dynamics and the policy. Taking the gradient of this function with respect to $a$ and then evaluating it at $a_0 = a$ is standard in gradient-aware RL (e.g., SVG, MAGE, Taylor-TD). The cumulative reward does involve the action: either directly through $r(s_t, a_t)$ or indirectly through its effect on future states. Equation (10) follows this standard notation.
>
>
> **On the computational cost of max-sliced MMD**
> The reviewer asks: *“What is the computational cost when we use max-sliced MMD?”*
> The additional cost of MSMMD comes solely from the gradient-ascent steps used in the max-slicing optimization. This overhead is controllable through the number of ascent iterations. Table 3 in Appendix N reports the measured wall-clock times: MSMMD-Sobolev incurs roughly a 56% overhead over the MMD variant. Given the contraction guarantees associated with the max-sliced version, such an overhead can be reasonable. We also note that both MMD and MSMMD perform well in practice, which suggests that practitioners may prefer the simpler MMD variant when its empirical performance is sufficient. This observation is consistent with prior work: contraction guarantees and empirical performance are not directly linked (see our answer to Reviewer Hcbc01 and the precedent in MMD-DRL [4] with the RBF kernel).

---

> > ### Author Response · Authors · 2025-11-23
> > **Part 3**
> >
> > **On the need for the TQC trick**
> > The reviewer asks: *“Why do we need to incorporate the TQC trick? Is there any ablation study…?”*
> > Controlling overestimation is particularly important in gradient-aware RL because we apply 10 critic updates per environment interaction [1]. MAGE already uses a similar idea through the double-critic structure of TD3, which also stabilizes training in our setting. Extending this approach directly to the distributional case did not yield reliable behaviour, so we adopt the TQC strategy: two critics are trained (as in MAGE and TD3), and we discard the top p% of their combined samples. This provides a finer-grained way to mitigate overestimation than discarding all samples from one critic.
> >
> > We added a dedicated ablation in Appendix M.2. In this experiment, we remove the overestimation-reduction mechanism from both MAGE and our (MS)MMD-Sobolev variants. As shown in Figure 12, removing it causes the performance to fall dramatically.
> >
> > **On the use of a cVAE instead of a diffusion model**
> > The reviewer writes: *“The introduction of CVAE is less convincing. Why do we not directly train a diffusion model?”*
> > Our use of a cVAE is motivated by computational constraints: Sobolev TD requires many world-model samples per update, and each sample must admit inexpensive differentiation with respect to (s, a) in order to compute the action-gradient target. Several generative model families satisfy these requirements (e.g., normalizing flows [3] or GANs [5]), but cVAEs remain a common choice in model-based RL because they are stable to train and allow fast differentiable sampling. We clarify this design choice in the revised manuscript (Sec. 5).
> >
> > Diffusion models [6], despite their empirical success, are substantially more expensive in this setting: computing $\partial (s', r) / \partial (s, a)$ would require backpropagating through the full denoising chain, making them prohibitively costly for the repeated one-step gradient evaluations required by Sobolev TD.
> >
> > To demonstrate that our method is not tied to the cVAE architecture, we added an experiment in Appendix M.1 replacing the cVAE with a simple normalizing flow (RealNVP [3]) for which we give some background in Appendix H.2. The flow-based world model integrates cleanly into our framework and yields results consistent with those obtained using the cVAE, confirming that our approach does not rely on any VAE-specific property.
> >
> > ---
> >
> > [1] D’Oro and Jaśkowski, 2020. *How to Learn a Useful Critic? Model Based Action Gradient Estimator Policy Optimization*. NeurIPS.
> > [2] M. Arjovsky, S. Chintala, and L. Bottou. *Wasserstein Generative Adversarial Networks*. ICML, 2017.
> > [3] L. Dinh, J. Sohl-Dickstein, and S. Bengio. *Density Estimation Using Real NVP*. arXiv:1605.08803, 2016.
> > [4] T. Nguyen-Tang, S. Gupta, and S. Venkatesh. *Distributional RL via Moment Matching*. AAAI, 2021.
> > [5] I. Goodfellow et al. *Generative Adversarial Nets*. NeurIPS, 2014.
> > [6] J. Ho, A. Jain, and P. Abbeel. *Denoising Diffusion Probabilistic Models*. NeurIPS, 2020.

---

### Official Review · Reviewer_dVoA · 2025-10-27

**Soundness:** 4
**Presentation:** 3
**Contribution:** 3
**Rating:** 6
**Confidence:** 3

**Summary:**

This paper seeks to improve the sample efficiency and performance of actor critic RL algorithms in stochastic environments by estimating the distribution of  action gradients through the critic. To do so it introduces the distributional modeling of returns and their gradients, and also details an approach to use it for actor critic training, which it terms distributional Sobolev training. The practical instantiation uses a form of maximum mean discrepancy to estimate the action gradients via a novel Bellman operator, and uses a variational autoencoder to estimate the one-step transition function.

Evaluations are done on a toy domain to verify that the approach works as intended and then in MuJoCo environments with added stochasticity to evaluate the approach on a standard benchmark.

**Strengths:**

* The proposed idea is intellectually stimulating and pushes the frontier on how actor critic algorithms work. It presents a novel idea and sound theoretical analysis to justify it.
* The proposed practical algorithm takes reasonable approaches for the required theoretical estimates, such as MMD and MSMMD.
* The use of the toy domain is useful to study if the proposed technique can deal with increased stochasticity as claimed
* The statistical bounds used in the evaluations are heartening and appreciated.
* The theoretical analysis seems to be sound
* The idea would be of interest to researchers who focus on dealing with stochasticity in the environment.

**Weaknesses:**

* The use of the VAE is a practical requirement to deal with environments without differentiable dynamics. Using a toy environment where the gradients of the environment are known would have been even better to evaluate how well the idea would work if the dynamics did not have to be estimated.
* While the results in the toy domain are impressive, it is unclear why they do not translate as well to the Mujoco environments, where MAGE remains within the margin of error. This difference is not a dealbreaker, since the trends across environments shows that the proposed techniques outperform MAGE across multiple environments.
* Nit: Ordering the methods in Figure 3 so the proposed methods are the rightmost ones and the ones on the left are baselines would make them much easier to read

**Questions:**

* Are practical considerations of implementing the idea potentially holding back the empirical results? Could the proposed approach be used for other robotics tasks like OGBench with added stochasticity?

---

> ### Author Response · Authors · 2025-11-23
>
> We thank the reviewer for the thoughtful comments and address each point below.
>
> **On practical considerations affecting empirical results**
> We agree that practical aspects of the implementation can influence the empirical gains. In this paper, we kept the world model architecture simple and identical across all MuJoCo tasks to keep the comparison to MAGE fair and to avoid per-environment tuning. More advanced modelling choices, such as free bits and symmetrized normalization [1] for the world model, or mixture of contractive kernels, could plausibly improve stability and reduce variance across tasks. These would be orthogonal refinements of the implementation and would not change the underlying Sobolev formulation.
>
> **On extending the framework to offline robotics settings such as OGBench**
> We believe the framework can extend to offline robotics settings such as OGBench. Offline RL already makes use of world models in several methods, for example MOReL [2], and gradient-aware RL provides an additional way to extract useful information from the learned dynamics rather than relying only on value predictions. When the data are fixed and potentially noisy, modelling a distribution over action gradients can help the critic remain stable. The small supervised experiment in Appendix K illustrates a related point: in a limited-data setting, the distributional Sobolev critic shows more stable second-order behaviour (Figure 5). This suggests that applying the method to offline benchmarks with both stochasticity and limited fixed data is a reasonable next step.
>
> **References**
>
> [1] Hafner, D., Pasukonis, J., Ba, J., and Lillicrap, T. *Mastering diverse domains through world models*.
> [2] R. Kidambi, A. Rajeswaran, P. Netrapalli, and T. Joachims. *MOReL: Model-Based Offline Reinforcement Learning*.

---

> > ### Comment · Reviewer_dVoA · 2025-11-27
> > **Response to Author Rebuttal**
> >
> > I appreciate the response to my review. I have a follow up question:
> > > Using a toy environment where the gradients of the environment are known would have been even better to evaluate how well the idea would work if the dynamics did not have to be estimated.
> >
> > Any response to this suggestion? Is it possible to include in the paper?

---

> > > ### Author Response · Authors · 2025-12-01
> > > **On a differentiable toy problem**
> > >
> > > We thank the reviewer for the follow-up and fully agree that a toy environment with known analytical gradients would be an interesting complement to our current experiments.
> > >
> > > Our main goal in this paper is to study the regime where the environment is not differentiable and where the dynamics are stochastic and partially observed. In all our experiments there is a latent state: either continuous noise corrupting the dynamics/observations in MuJoCo, or the discrete hidden “mode” in the toy task. In this setting, the stochastic world model plays two roles:
> > > (i) it provides differentiable one-step transition samples
> > > $$
> > > (s',r) = f_\theta(s,a,z),\qquad z\sim p(z\mid s,a),
> > > $$
> > > allowing us to compute Sobolev TD gradients
> > > $$
> > > \nabla_{(s,a)}(s',r),
> > > $$
> > > and (ii) it infers a *consistent* latent configuration compatible with the current observation before generating $(s',r)$. This emerges naturally from cVAE training through
> > > $$
> > > q_\phi(z\mid s,a,s',r)
> > > $$
> > > and the KL-regularized decoder likelihood, such that the conditional prior $p(z\mid s,a)$ tends to produce latent samples that correspond to realistic transitions for that part of the state–action space.
> > >
> > > A differentiable toy environment with ground-truth gradients would test a different regime, one where the true Jacobian of the transition dynamics is available. Constructing such an environment while preserving the latent stochasticity present in our setting is non-trivial: one would need a simulator implementing
> > > $$
> > > \frac{\partial (s',r)}{\partial (s,a)}
> > > $$  and also a stochastic mechanism
> > > $$
> > > (s',r)=g(s,a,\xi),\qquad \xi\sim p(\xi\mid s,a),
> > > $$  that plays an equivalent role to the inferred latent variable $z$. Without such an inference mechanism, sampled transitions would not reflect the uncertainty that DSDPG is designed to handle. In addition, Sobolev TD requires many such gradient evaluations per update, which is substantially cheaper through a compact differentiable world model than through a full simulator.
> > >
> > > For these reasons, we focused on the practically relevant case where gradients must be *inferred* rather than supplied by the environment, and evaluated the method under controlled latent stochasticity instead of known-gradient dynamics.

---

### Official Review · Reviewer_Dy2h · 2025-10-29

**Soundness:** 4
**Presentation:** 4
**Contribution:** 4
**Rating:** 10
**Confidence:** 3

**Summary:**

This paper considers distributional reinforcement learning, and describes how to incorporate gradient information into distributional temporal-difference learning. Not only is it theoretically well-motivated, it also describes practical means of implementation even in challenging, realistic settings such as non-differentiable environments.

**Strengths:**

- This paper makes foundational contributions to the important and timely topic of distributional RL.
- It is exceptionally well-written, with both pedagogical clarity and mathematical precision (admittedly, it gets a bit dense at times, but I think that is unavoidable).
- The positioning against the Related Work is detailed, clear, and well articulated. Furthermore, it gives a balanced and honest discussion of its limitations.
- I appreciate the pedagogical toy problem
- The extensive appendices provide detailed support (mostly proofs) for the claims in the main paper.

**Weaknesses:**

Honestly, I can't think of any major weaknesses. Of course, it would have been nice to see clear improvements over the baselines on an established benchmark. Still, I think the theoretical and algorithmic contributions by far outweigh the importance of such experimental results.

Minor things:
- Some of the figures (and especially the text therein) are tiny and require zooming.
- As a matter of personal preference, I think it's neater to emphasize text with italics rather than bold font, but I'll leave that for your discretion.

**Questions:**

No further questions.

---

> ### Author Response · Authors · 2025-11-23
>
> We thank the reviewer for the very positive and encouraging assessment. We appreciate the helpful minor suggestions, and in response we have increased the font size of several figures and replaced bold emphasis with italics to improve readability.

---

> > ### Comment · Reviewer_Dy2h · 2025-11-28
> > **Acknowledgement**
> >
> > I acknowledge that I've read the authors' rebuttals to all reviewers, and I maintain the view that this is a high-quality paper that deserves to be accepted.

---

### Official Review · Reviewer_Hcbc · 2025-11-01

**Soundness:** 4
**Presentation:** 4
**Contribution:** 3
**Rating:** 6
**Confidence:** 2

**Summary:**

This paper addresses a limitation of existing gradient-aware reinforcement learning methods, which improve sample efficiency by incorporating value gradients into critic training. The authors identify that current approaches treat these gradients as deterministic quantities. Thus, they extend distributional RL to model the joint distribution of both the cumulative return and its action-gradient. The proposed algorithm, Distributional Sobolev Deterministic Policy Gradient is built upon three key technical components. First is a novel Sobolev Bellman operator, which forms the theoretical core by bootstrapping the joint distribution of returns and their gradients. Second, to make this operator practical in non-differentiable environments, the method employs a conditional Variational Autoencoder enabling gradient backpropagation through the learned dynamics. Third, the critic is trained to minimize the discrepancy between its predicted distribution and the bootstrapped target distribution.

**Strengths:**

- Extending the idea of distributional RL to action-gradient is interesting. By proposing to learn the distribution of the gradient, the work offers a principled and more complete model of credit assignment under uncertainty. This moves the field beyond the limitations of prior methods like MAGE, which are confined to modeling the gradient's expectation and are thus vulnerable to noise.
- The formal derivation of the Sobolev Bellman operator is solid.
- Except for theoretical contributions, the paper also provides comprehensive experimental results.

**Weaknesses:**

- As the authors stated, one weakness of DSDPG is its computational complexity.
- There is a disconnect between the theoretical motivation for the MSMMD metric and its empirical utility. The paper introduces MSMMD primarily to obtain a provable contraction for the Sobolev Bellman operator under a tractable, sample-based metric. Standard MMD does not offer this same general guarantee. However, the empirical results across both the toy problem and the MuJoCo benchmarks show that the MSMMD variant provides at best a marginal performance improvement over the simpler MMD variant, and in some cases performs identically.

**Questions:**

What are the reasons for choosing the incomplete operator for DSDPG?

---

> ### Author Response · Authors · 2025-11-23
>
> We thank the reviewer for the thoughtful feedback and the opportunity to clarify these points.
> Below we address the points raised by the reviewer.
>
> *“There is a disconnect between the theoretical motivation for the MSMMD metric and its empirical utility. MSMMD was introduced for its contraction guarantees, but empirically it seems to perform only marginally better than MMD.”*
>
> We agree that MSMMD is introduced primarily because it provides a tractable contraction analysis for the Sobolev Bellman operator. At the same time, the relationship between contraction guarantees and empirical performance in distributional RL is not straightforward, and this has precedent. For example, the MMD-DRL paper [1] identified a family of kernels for which the induced metric is provably contractive, yet used the non-contractive RBF kernel in all experiments because it performed best in practice. This illustrates that a metric with a contraction guarantee does not necessarily yield superior empirical returns, and that a non-contractive alternative can still perform strongly. In our case, MSMMD enables our theoretical results, while MMD remains competitive and in many settings equally performant.
>
> ---
>
> *“What are the reasons for choosing the incomplete operator?”*
>
> We chose the incomplete operator because our goal was to extend the MAGE framework [2], which focuses exclusively on the action-gradient. The complete operator is included for conceptual completeness, but applying it directly raises known stability issues. For instance, Taylor TD-learning [3] attempted to incorporate state-gradient information, but found the updates unstable unless scale information was removed—a modification that would be incompatible with preserving the contraction properties we study.
>
> To maintain a stable and faithful extension of MAGE, we therefore adopt the incomplete operator for DSDPG. We now state explicitly in the paper that exploring the complete operator is a promising direction for future work.
>
> ---
>
> ### References
>
> [1] Nguyen Tang et al., 2021. *Distributional Reinforcement Learning via Moment Matching*. AAAI.
> [2] D’Oro and Jaśkowski, 2020. *How to Learn a Useful Critic? Model Based Action Gradient Estimator Policy Optimization*. NeurIPS.
> [3] Garibbo et al., 2023. *Taylor TD-Learning*. NeurIPS.

---

### Author Response · Authors · 2025-11-23
**General comment**

We would like to thank the reviewers for their thoughtful feedback.

One comment raised concerns about overestimation bias. As detailed in our response to Reviewer Z5td, we now include a dedicated ablation showing that removing the overestimation-reduction mechanism, double estimation for MAGE and TQC for our distributional variants, causes performance to deteriorate sharply.

Another comment concerns our use of a cVAE as the world model. As explained in our reply to Reviewer Z5td, our approach is not tied to VAEs, any generative model that supports efficient differentiable sampling is compatible. To illustrate this, we added an experiment replacing the cVAE with a normalizing flow, which integrates seamlessly into our pipeline and yields comparable conclusions.

All modifications and additions in the revised manuscript are highlighted in red.

---

### Meta-Review · Program_Chairs · 2026-01-08

**Summary:**

The reviewers identified several key concerns that informed the initial mixed scores (6, 10, 6, 2). The primary critiques centered on:

Theoretical Grounding & Motivation: Reviewer Z5td questioned the foundational motivation for combining distributional RL with gradient information, raised significant concerns about the practicality of the theoretical assumptions (e.g., Lipschitz continuity of networks, contraction conditions), and perceived a gap between the theory and the final algorithm.

Empirical Utility & Improvement: Reviewers Hcbc and dVoA noted a disconnect between the strong theoretical motivation for the MSMMD metric and its marginal empirical gains over simpler MMD. They, along with Reviewer Z5td, also questioned the significance of the performance improvements over strong baselines like MAGE, given the increased computational cost.

**Reviewer Concerns:**

Addressed Concerns:

The authors compellingly addressed this (to Reviewers Hcbc and Z5td) by citing precedent in distributional RL literature where non-contractive metrics can outperform provably contractive ones in practice. They clarified that MSMMD's primary value is in enabling the theoretical analysis, while MMD remains a strong, simpler alternative.

The authors provided a reasoned response to Reviewer dVoA's suggestion for a known-gradient toy environment, explaining their focus on the more practical and challenging non-differentiable, latent-state setting.

Outstanding Concerns:

Theoretical Assumptions (Reviewer Z5td): While the authors provided clarifications on the Lipschitz assumptions (noting they are standard and align with related work like MAGE) and detailed the contraction conditions, a core philosophical disagreement likely remains.

**Reviewer Scores:**

Reviewer Hcbc (Initial: 6): Their main concern (MSMMD's empirical utility) was well-addressed with a convincing precedent. The justification for the incomplete operator was also clear. Likely raised their score to a 7 or 8 (Clear Accept).

Reviewer Dy2h (Initial: 10):  Score would remain 10 (Strong Accept).

Reviewer dVoA (Initial: 6): Their questions on practical considerations and future applicability were addressed thoughtfully. Likely raised their score to 8 (Accept).

Reviewer Z5td (Initial: 2): This is the most complex case. The authors provided comprehensive point-by-point responses, added ablations (TQC, normalizing flow), and clarified theoretical contexts. This would likely mitigate some concerns regarding motivation and design choices. However, the issues regarding theoretical assumptions and experimental impact likely persist. Their score might see a modest increase to a 4 (Borderline Reject).

---

### Decision · Program_Chairs · 2026-01-26

Accept (Poster)